# Comparison of basin-scale *in situ* and meteoric $^{10}$Be erosion and denudation rates in felsic lithologies across an elevation gradient at the George River, northeast Tasmania, Australia

Leah A. VanLandingham[1], Eric W. Portenga[1], Edward C. Lefroy[2], Amanda H. Schmidt[3], Paul R. Bierman[4], Alan J. Hidy[5]

[1]Geography and Geology Department, Eastern Michigan University, Ypsilanti, MI 48197, United States
[2]Tasmanian Institute of Agriculture, University of Tasmania, Private Bag 98, Hobart 7001, Australia
[3] Geology Department, Oberlin College and Conservatory, Oberlin, OH 44074, United States
[4]Rubenstein School for Natural Resources and the Environment, University of Vermont, Burlington, VT 05405, United States
[5]Center for Accelerator Mass Spectrometry, Lawrence Livermore National Laboratory, Livermore, CA 94550, United States

*Correspondence to*: Eric W. Portenga (eric.portenga@emich.edu)

**Abstract.** Long-term erosion rates in Tasmania, at the southern end of Australia's Great Dividing Range, are poorly known; yet, this knowledge is critical for making informed land-use decisions and improving the ecological health of coastal ecosystems. Here, we present quantitative, geologically-relevant estimates of erosion rates for the George River basin, in northeast Tasmania, based on *in-situ* produced $^{10}$Be ($^{10}$Be$_i$) measured from stream sand at two trunk channel sites and seven tributaries (mean $24.1 \pm 1.4$ Mg km$^{-2}$ y$^{-1}$; $1\sigma$). These new $^{10}$Be$_i$-based erosion rates are strongly related to elevation, which appears to control mean annual precipitation and temperature, suggesting that elevation-dependent surface processes influence rates of erosion in northeast Tasmania. Erosion rates are not corelated with slope in contrast to erosion rates along the mainland portions of Australia's Great Dividing Range. We also extracted and measured meteoric $^{10}$Be ($^{10}$Be$_m$) from grain coatings of sand-sized stream sediment at each site, which we normalize to measured concentrations of $^9$Be and use to estimate $^{10}$Be$_m$-based denudation rates for the George River. $^{10}$Be$_m$/$^9$Be$_{reac}$ denudation rates replicate $^{10}$Be$_i$ erosion rates within a factor of three but are highly sensitive to the value of $^9$Be that is found in bedrock ($^9$Be$_{parent}$), which was unmeasured in this study. $^{10}$Be$_m$/$^9$Be$_{reac}$ denudation rates seem sensitive to recent mining, forestry, and agricultural land use, all of which resulted in widespread topsoil disturbance. Our findings suggest that $^{10}$Be$_m$/$^9$Be$_{reac}$ denudation metrics will be most useful in drainage basins that are geologically homogeneous, where recent disturbances to topsoil profiles are minimal, and where $^9$Be$_{parent}$ is well constrained.

## 1 Introduction and the Importance of the George River, Tasmania

Erosion rates of river basins derived from measurements of the *in-situ* produced cosmogenic isotope, $^{10}$Be$_i$, have been used to infer topographic, tectonic, and climatic drivers of landscape evolution for thousands of individual river basins (Codilean et al., 2018; Harel et al., 2016; Mishra et al., 2019; Portenga and Bierman, 2011; Wittmann et al., 2020) and to contextualize

the effects of land use on erosion and sediment dynamics (Portenga et al., 2019; Schmidt et al., 2018). Sufficient data now exist that erosion rates from individual studies have been compiled and analysed at the scale of entire continental orogens to

demonstrate primary and secondary controls on erosion across thousands to tens of thousands of years (Aguilar et al., 2014; Carretier et al., 2018; Codilean et al., 2021; Delunel et al., 2020; Starke et al., 2020). For example, Delunel et al. (2020) find that $^{10}Be_i$ erosion rates across the European Alps are strongly linked to mean basin slope and influenced by uplift and glaciation. A number of north-south latitudinal studies from the South American Andes show that erosion in some segments of the range is driven by uplift (Carretier et al., 2015; Starke et al., 2017) and slope (Carretier et al., 2018) but not necessarily

by rainfall unless one considers the effects of vegetation in driving soil weathering rates (Carretier et al., 2015; Starke et al., 2020). A new compilation and analysis of $^{10}Be_i$ erosion rates across the Great Dividing Range of eastern Australia is the first to analyse landscape dynamics across a continent-spanning, passive, post-orogenic rift margin and finds that basin slope is most closely related to erosion at all spatial scales, more so than any other potential driver of erosion (Codilean et al., 2021). While Codilean et al.'s (2021) analysis comprises erosion rates from the western and eastern flanks of the Great Dividing

Range—from tropical rainforests in northern Queensland to temperate southeast Victoria—it is restricted to mainland Australia.

Despite the widespread measurement of $^{10}Be_i$ to elucidate erosion rates globally, erosion rate data do not exist for many areas of Earth's surface. Understanding of drivers of erosion will be improved by measuring erosion rates in these understudied

areas. In this study, we supplement Codilean et al.'s (2021) erosion compilation with the first $^{10}Be_i$-based erosion rates from the southernmost end of the eastern Australian passive margin on the island-state of Tasmania, specifically the George River basin (Fig. 1). Data in this study are also the first erosion rates measured in temperate rainforests of the Southern Hemisphere (cf. Adams and Ehlers, 2017; Belmont et al., 2007). Quantitative erosion rate data for Tasmania and many of its fluvial systems are currently lacking (Jerie et al., 2003; Koehnken, 2001); data, such as we provide here, are useful information for

land managers and for estuary restoration efforts.

The George River empties into Georges Bay (with an 's'), which is known for its oyster stocks (Mitchell et al., 2000) but has been degraded by a history of timber production, tin mining, and agriculture. Historical land-use practices in the catchment have supplied $>10^6$ m$^3$ of sediment to Georges Bay since the late 19$^{th}$ century (Knighton, 1991) and continue to release

pollutants to the Bay (Bleaney et al., 2015; Crawford and White, 2005). The success of efforts to rehabilitate Georges Bay relies in part on reducing sediment delivery from the George River to Georges Bay to pre-disturbance levels (Batley et al., 2010; Crawford and White, 2005; Kragt and Newham, 2009; McKenny and Shepherd, 1999; Mount et al., 2005), but no pre-disturbance erosion data exist for the George River, nor do any geologically-relevant erosion rates exist for any part of Tasmania. Measuring erosion rates for the George River contributes to the growing geomorphological understanding of the

drivers of erosion in Tasmania, across Australia, and in similar geological settings elsewhere.

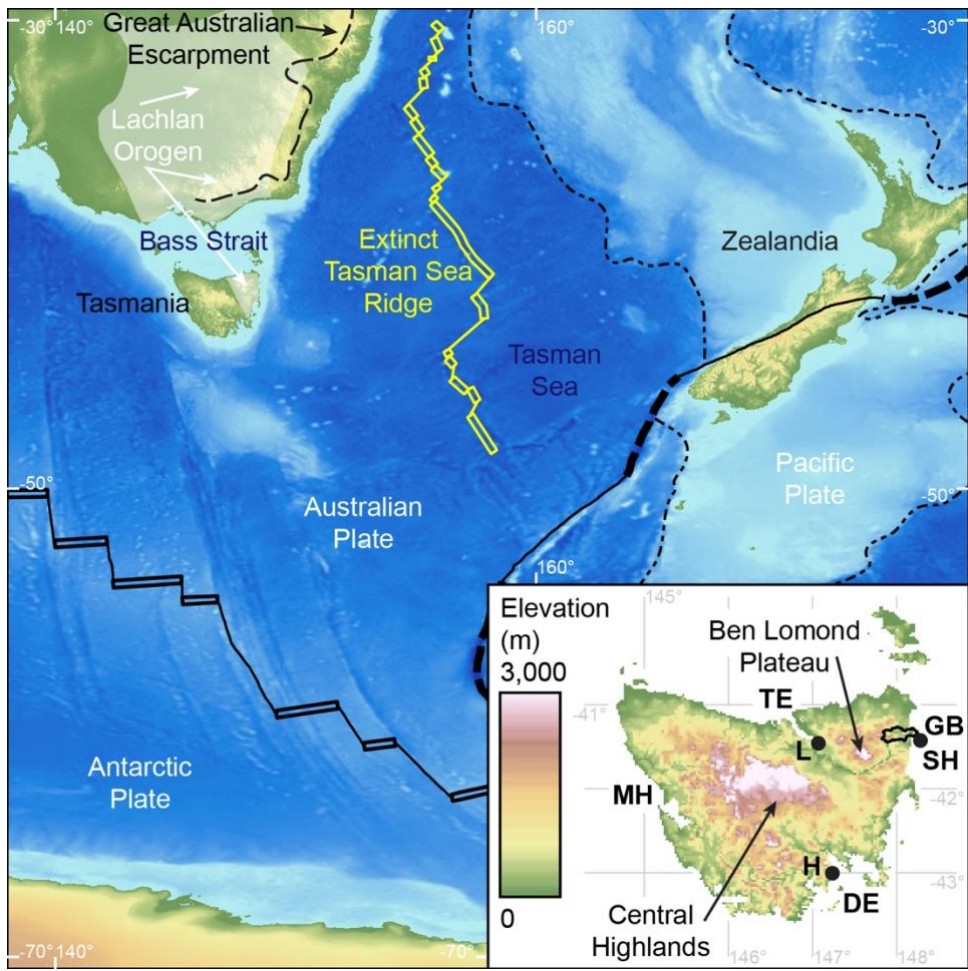

**Figure 1: Generalized tectonic map of the eastern Southern Ocean/southwest Pacific Ocean, surrounding Tasmania, including large-scale geologic structures in southeast Australia and Tasmania: double-black lines = active mid-ocean ridges; bold dashed black line = convergent plate boundaries; thin solid black lines = transform boundaries. Inset shows detailed topography of Tasmania. The main George River basin is shown outlined in black. Major estuaries of other Tasmanian river systems are indicated for reference: Derwent Estuary (DE), Macquarie Harbour (MH), Tamar Estuary (TE), Georges Bay (GB). Cities are shown with black dots for reference: Hobart (H), Launceston (L), St. Helens (SH).**

## 1.1 Quantifying landscape dynamics with *in situ* and meteoric $^{10}$Be

The primary goal of this study is to provide background rates (over millennia) of landscape change in the George River basin using the *in situ* cosmogenic isotope beryllium-10 ($^{10}$Be$_i$) in fluvial sediment (Bierman and Steig, 1996; Brown et al., 1995; Granger et al., 1996). $^{10}$Be$_i$ production decreases exponentially with depth in rock and sediment near Earth's surface such that $^{10}$Be$_i$ concentrations at depths >2 m are much lower compared to those measured closer to Earth's surface (Gosse and Phillips, 2001; Lal, 1991). $^{10}$Be$_i$ produced by muons dominates at depths >2 m (Braucher et al., 2003; Gosse and Phillips, 2001; Heisinger et al., 1997), but muogenic $^{10}$Be$_i$ production is negligible when compared to near-surface spallogenic $^{10}$Be$_i$ production, except in rapidly eroding landscapes or landscapes with steep terrain (e.g., Dethier et al., 2014; Fellin et al.,

2017; Rosenkranz et al., 2018; Scherler et al., 2014; Siame et al., 2011). Bioturbation homogenizes $^{10}Be_i$ concentrations in soils, in many places to depths of at least ~1 m (Brown et al., 1995; Schaller et al., 2018), and thus $^{10}Be_i$ erosion rates are largely insensitive to widespread shallow erosion. This insensitivity allows $^{10}Be_i$ erosion rates to be a useful gauge of pre-disturbance rates of landscape change (Ferrier et al., 2005; Portenga et al., 2019; Schmidt et al., 2018; Vanacker et al., 2007), except where human land use is intensive (i.e., Schmidt et al., 2016) or the effects of human land use are exacerbated by climate extremes (i.e., Rosenkranz et al., 2018). Pre-disturbance $^{10}Be_i$ erosion data can thus inform approaches to reducing sediment delivery from the George River and support efforts to improve the ecological health of the Georges Bay estuary and possibly other watersheds in northeast Tasmania that share similar bedrock and topographic characteristics by providing a benchmark against which to compare modern sediment loads.

In addition to $^{10}Be_i$, which is produced in rock and sediment, $^{10}Be$ is also produced via spallation of oxygen and nitrogen in the atmosphere; this $^{10}Be$ rains out or falls to Earth's surface (meteoric $^{10}Be$; $^{10}Be_m$; Heikkilä and von Blanckenburg, 2015; Monaghan et al., 1986; Reusser et al., 2010) where it is readily adsorbed into sediment grain coatings. $^{10}Be_m$ has traditionally been used to trace sediment through landscapes (Brown et al., 1988; Helz et al., 1992; Portenga et al., 2017; Reusser and Bierman, 2010; Valette-Silver et al., 1986), but recently derived equations (along with a series of assumptions) now allow denudation rates to be calculated from measurements of $^{10}Be_m$ that are normalized to non-cosmogenic, stable $^{9}Be$, which weathers out of mineral grains ($^{9}Be_{reac}$; von Blanckenburg et al., 2012). $^{10}Be_m/^{9}Be_{reac}$ denudation rates have been used to quantify landscape evolution over a variety of spatial scales for different river basins (Dannhaus et al., 2018; Deng et al., 2020; Portenga et al., 2019; Rahaman et al., 2017; Wittmann et al., 2012, 2015; in some cases $^{10}Be_m$ is referred to as the reactive phase of $^{10}Be_m$ [$^{10}Be_{reac}$] and denudation rates may be referred to as $^{10}Be_{reac}/^{9}Be_{reac}$ denudation rates) and have shown promise in quantifying landscape dynamics in quartz-poor landscapes (Deng et al., 2020; Rahaman et al., 2017).

In this study, we use both $^{10}Be_i$ and $^{10}Be_m/^{9}Be_{reac}$ to measure the rates at which mass is lost from the George River basin's slopes. Over timescales sufficiently long that the assumption of steady state is approached, all of this mass will transported to the Georges River estuary. Such mass loss from the George basin slopes is both chemical (dissolved load) and physical (sediment transport). The partitioning between these phases differs dramatically around the world depending on rock type, topography, and weathering regime and likely differs within the study basin. The assumptions underlying these two methods ($^{10}Be_i$ and $^{10}Be_m/^{9}Be_{reac}$) differ; thus, results from each method may not be the same. The concentration of $^{10}Be_i$ is biased towards mass loss within the upper meters of Earth's surface where rates of neutron spallation are high. Both chemical and physical mass losses within this surface layer of regolith are reflected by $^{10}Be_i$ concentrations. $^{10}Be_m/^{9}Be_{reac}$, if the assumptions of the analytical model are met, reflects both physical and chemical mass loss throughout the regolith, regardless of depth.

The terms "erosion" and "denudation" have been used without precision in the literature, often as a replacement for one another. Erosion is applied more often to rates calculated using the concentration of $^{10}Be_i$; rates calculated using $^{10}Be_m$ /$^9Be_{reac}$ are more frequently referred to as denudation. We follow that convention in this paper. Because we have dissolved and suspended load data as well as river flow over time from the mouth of the George River, we attempt to provide a full

discussion of what the rates we measure mean for landscape dynamics within the George River Basin.

The small size and relatively uniform bedrock geology of the George River basin provide an ideal location to compare $^{10}Be_i$ erosion rates with $^{10}Be_m$/$^9Be_{reac}$ denudation rates (von Blanckenburg et al., 2012). $^{10}Be_m$ can be desorbed from sediment grain coatings under low pH conditions (Aldahan et al., 1999; You et al., 1989), but $^{10}Be_m$ loss from soil profiles in solution is

likely minimal in the George River basin because measured soil pH values in the catchment range from 4.0–5.5 (Kidd et al., 2015) and long-term monitoring of stream water pH at two gauging stations—one in Ransom Creek and the other at the George River in St. Helens—shows that stream pH is consistently >5 and mostly >6 (DPIPWE, 2021a,b). The George River basin is a landscape of relative geological homogeneity in comparison to more geologically-diverse landscapes with similar data sets (i.e., Deng et al., 2020; Portenga et al., 2019; Rahaman et al., 2017). Although the George River has a simple

bedrock geology, it also has a long history of forestry and lode and placer tin mining that has, in the past, disturbed the hillslopes and fluvial systems (Knighton, 1991; Preston, 2012). Given that land use is speculated to affect the results of $^{10}Be_m$/$^9Be_{reac}$-derived denudation rate calculations elsewhere (Portenga et al., 2019), we also explore how land use in the George River affects our interpretations of $^{10}Be$-based erosion and denudation calculations in this study.

**2 Field Area**

Tasmania separated from mainland Australia during Cretaceous rifting of Antarctica and Australia and sits at the southern end of the Great Australian Escarpment—a steep arch-type escarpment that formed during the separation of Zealandia from mainland Australia in the Mid- to Late-Cretaceous (Fig. 1; Codilean et al., 2021; Crowder et al., 2019; Etheridge et al., 1987; Gaina et al., 1998; Griffiths, 1971; Gunn, 1975; Hayes and Ringis, 1973; Lanyon et al., 1993; Matmon et al., 2002;

McDougall and van der Lingen, 1974; Mortimer et al., 2017; Persano et al., 2002; Sutherland et al., 2001; Weissel and Hayes, 1977). Bedrock of the George River basin is granodiorite and granite associated with the Blue Tier Batholith (S-type granites), which was emplaced into sediments of the Mathinna Supergroup in the Devonian (Fig. 2; Foster et al., 2000; Gee and Groves, 1971; Gray and Foster, 2004; Higgins et al., 1985; McCarthy and Groves, 1979; Seymour et al., 2006). Siluro-Devonian sedimentary rocks and Neogene basalts underlie small areas, primarily along drainage divides in the central and

the western George River basin (Seymour et al., 2006).

The George River basin, located in northeastern Tasmania, is of modest size (557 km$^2$) with low elevation (mean = 386 m) and gentle hillslopes (mean = 10°). It drains the eastern slopes of the Rattler Range, which currently has a warm, temperate climate (Kottek et al., 2006). Despite eastern Tasmania being in the rain shadow of the central Tasmanian Highlands and

western coast ranges, measurements from rainfall gauging stations and temperature data loggers within and near the George River basin show that the local topography of the Ben Lomond Plateau induces strong relationships across the basin between elevation, mean annual precipitation, and mean annual temperature (Fig. 3; Table 1; BoM, 2021; Webb et al., 2018, 2020).

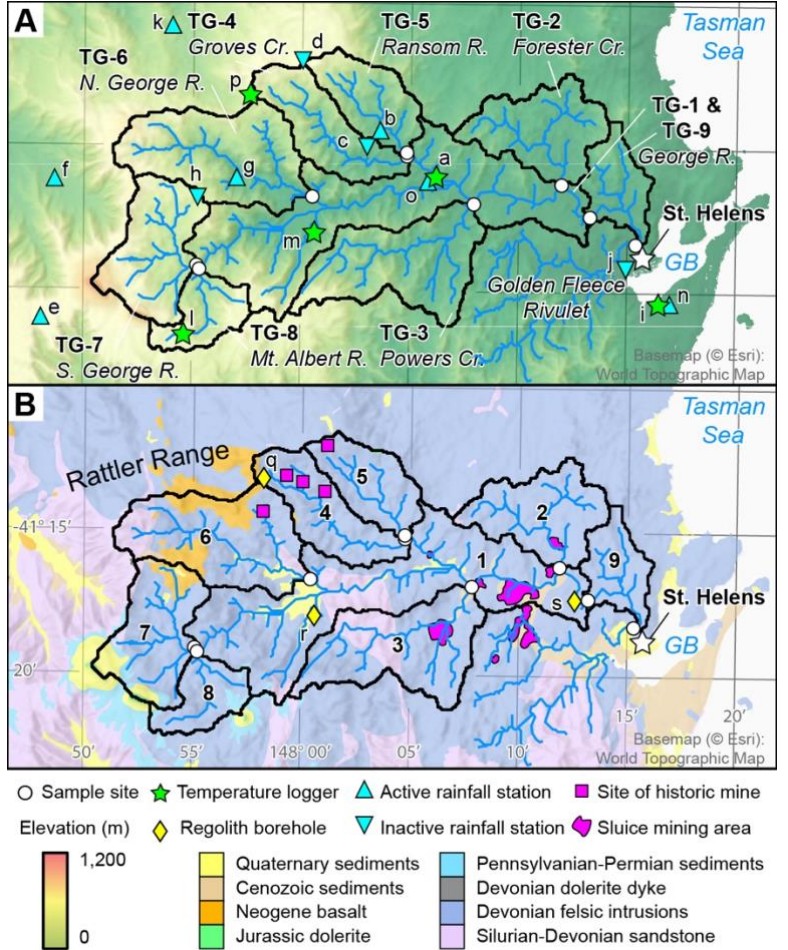

**Figure 2: A. Elevation map of the topography of the George River basin. Sample collection sites (white circles), active and inactive**
**Australian Bureau of Meteorology rainfall gauging stations (upright and inverted cyan triangles, respectively), and temperature logger locations (green stars) are shown (Webb et al., 2018, 2020). B. Bedrock geology map of George River shows the widespread occurrence of Devonian felsic intrusions of the Blue Tier Batholith, which underlies the vast majority of the field area. Note that basins TG-2, TG-4, TG-5, and TG-8 are almost entirely underlain by Devonian felsic intrusions. Areas of historic mining are shown (pink squares and polygons; Knighton, 1991), the action of which delivered >10$^6$ m$^3$ to the George River delta in Georges**
**Bay (GB). Locations of boreholes, that strike bedrock are shown by yellow diamonds (BoM, 2015). Maps are projected in UTM Zone 55S; coordinates shown in Panel B are the same for Panel A. Topographic relief basemaps come from the ArcGIS Online World Topographic Map (Esri, 2012).**

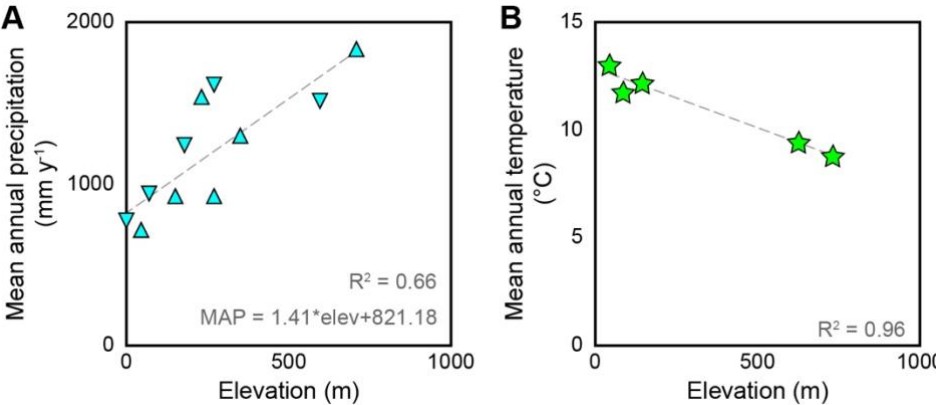

Figure 3: A. Mean annual precipitation from active (cyan triangles) and inactive (inverted cyan triangles) Australian Bureau of Meteorology rainfall gauging stations across George River basin that have at least 1 full year of recorded data exhibiting a strong correlation with station elevation. B. Mean annual temperature (green stars) taken from temperature loggers with >2 years of nearly-daily data showing a strong inverse correlation with elevation. Precipitation and temperature data shown in Table 1.

Human land use in Tasmania begins >35 ka, when Aboriginal Australians crossed to the island from the Australian mainland (Cosgrove, 1995; Cosgrove et al., 1990), possibly corresponding to subaerial exposure of the Bass Strait ~56–40 ka (MacIntosh et al., 2006) and localized ice advances in the central Tasmanian highlands (Barrows et al., 2001, 2002; Colhoun, 2002; MacIntosh et al., 2006). Ecological habitat suitability models, based on characteristics and locations of thousands of archaeological sites across Tasmania indicate that Aboriginal communities were located close to freshwater sources and coastal resources, such as the landscapes around Georges Bay and the lower elevations within tributaries to the George River (Jones et al., 2019). Human arrival in Tasmania has been linked to widespread erosion events in mid-elevation landscapes (McIntosh et al., 2009).

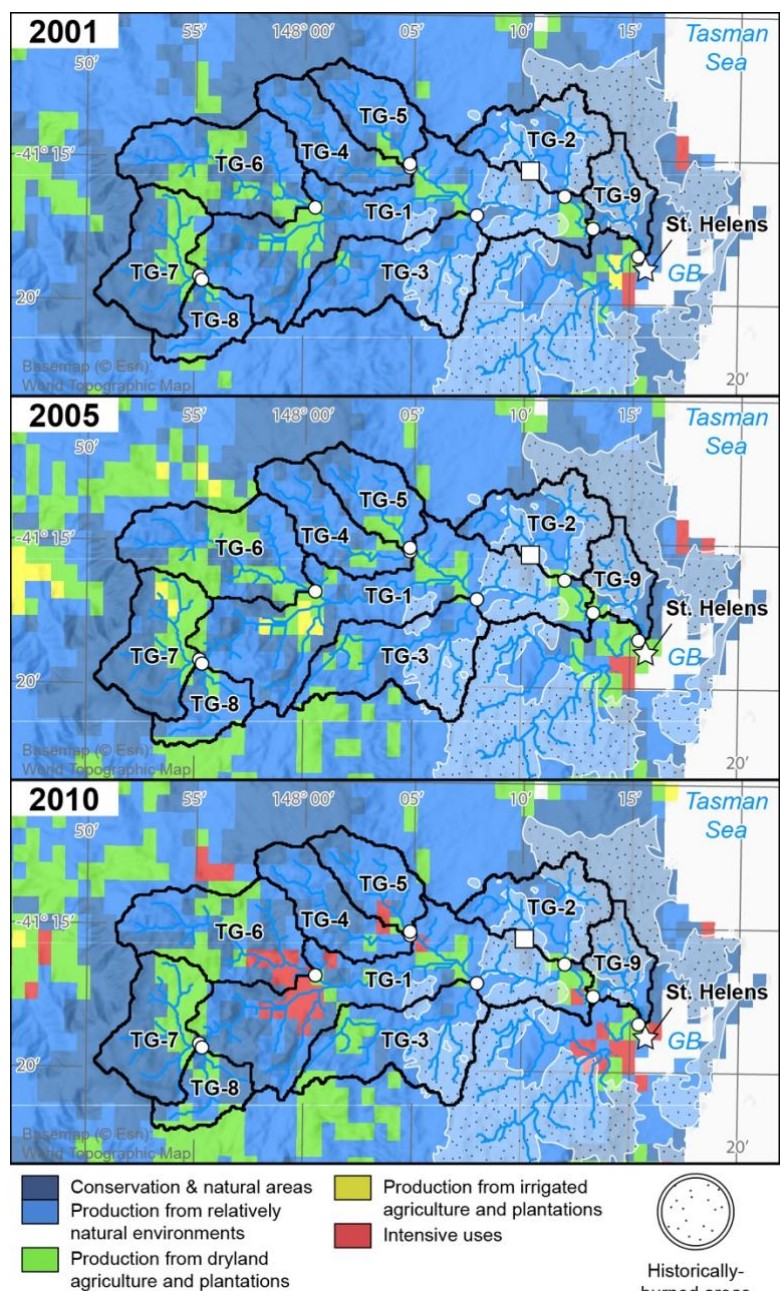

**Figure 4:** Land cover for each sampled tributary catchment in the George River basin from 2001 (top), 2005 (center), and 2010 (bottom) – the period of leading up to and immediately following sample collection in 2008. The Australian Land Use and Management Classification system groups land use into five primary Classes based on their potential to impact the natural environment (ABARES, 2016). White square denotes location of the Gentle Annie experimental catchment (Wilson, 1999). Stippled areas outlined in white are areas that have been affected by forest fires or prescribed burns in the past (Land Tasmania, 2020). Maps are projected in UTM Zone 55S. Topographic relief basemaps come from the ArcGIS Online World Topographic Map (Esri, 2012).

**Table 1. Meteorological and bedrock data for the George River basin**

| Bureau of Meteorology Rainfall Station Name | Figure 2A Map ID | Bur. Of Met. Station ID | Latitude (°) | Longitude (°) | Station Elevation (m) | Data Range[a] | Years of Record | Active? | Mean Annual Precipitation (mm y$^{-1}$) |
|---|---|---|---|---|---|---|---|---|---|
| Goshen (Post Office) | a | 92065 | -41.27 | 148.10 | 76 | 1965–1970, 1972–1973 | 8 | No | 934 |
| Goulds Country | b | 92131 | -41.24 | 148.06 | 237 | 2005, 2016, 2018, 2020 | 4 | Yes | 1503 |
| Goulds Country Post Office | c | 92016 | -41.25 | 148.05 | 183 | 1885–1895, 1897–1963 | 78 | No | 1228 |
| Lottah | d | 92022 | -41.20 | 148.00 | 274 | 1902–1916, 1918–1935, 1943–1950 | 41 | No | 1611 |
| Mt. Victoria (Una Plains) | e | 91194 | -41.35 | 147.80 | 710 | 1958, 1960, 1962–1964, 1966–1967, 1969, 1971–1974, 2011–2016, 2018–2020 | 21 | Yes | 1836 |
| New River (New River Road) | f | 91300 | -41.27 | 147.81 | 274 | 1997, 2015, 2019–2020 | 4 | Yes | 901 |
| Pyengana (Forest Lodge Road) | g | 92051 | -41.27 | 147.95 | 155 | 1963–1999, 2002, 2005, 2007–2008, 2010–2015, 2017–2020 | 51 | Yes | 904 |
| Pyengana (Sea View) | h | 92103 | -41.28 | 147.92 | 598 | 1988–1992, 1994–2000, 2002, 2005–2006 | 15 | No | 1512 |
| St Helens Aerodrome | i | 92120 | -41.34 | 148.28 | 48 | 2001, 2003–2010, 2012, 2014–2020 | 16 | Yes | 681 |
| St Helens Post Office | j | 92033 | -41.32 | 148.25 | 5 | 1890–1904, 1906–1993, 1995–1999 | 108 | No | 777 |
| Weldborough | k | 92126 | -41.18 | 147.90 | 355 | 2004–2011, 2013–2014, 2016 | 11 | Yes | 1265 |

| Temperature Logger Location ID[b] | Figure 2A Map ID | Latitude (°) | Longitude (°) | Logger Elevation (m) | Data Range[b] | Years of Record | Active? | Mean Annual Temperature (°C) |
|---|---|---|---|---|---|---|---|---|
| 1619552 | l | -41.36 | 147.91 | 732 | 2013–2017 | 5 | No | 8.8 |
| 1620197 | m | -41.30 | 148.01 | 145 | 2013–2017 | 5 | No | 12.2 |
| 1621107 | n | -41.34 | 148.28 | 44 | 2013–2017 | 5 | No | 13.0 |
| 1621175 | o | -41.27 | 148.10 | 86 | 2013–2015 | 3 | No | 11.8 |
| 2623239 | p | -41.22 | 147.96 | 627 | 2016–2017 | 2 | No | 9.5 |

| Depth to Regolith Borehold ID[c] | Figure 2B Map ID | Latitude (°) | Longitude (°) | Elev. of Top of Bore (m) | Depth to Bedrock through Regolith (m) |
|---|---|---|---|---|---|
| 17640 | q | -41.22 | 147.97 | 627.8 | 18.3 |
| 40783 | r | -41.29 | 148.21 | 81.1 | 51.8 |
| 41615 | s | -41.30 | 148.01 | 162.0 | 54.0 |

[a] Years listed in data ranges are the first and last years for which 12 months of data are available

[b] Temperature logger data sourced from the State of Tasmania Air Temperature Logger Recording Database, used by Webb et al. (2018, 2020). Each year has temperature recorded for at least 30% of days (average = 71%)

[c] Depth to regoltih measured in boreholes (BoM, 2015)

**Table 1: Locations and data used from rainfall gauges, air temperature loggers, and depth of regolith geological borehole (Fig. 2).**

190

195

**Table 2. Sample locations and topographical basin data**

| Sample ID | Sample Location | | | Basin-average elevation (m)[a] | Basin Area (km²)[a] | Mean local relief (m)[a] | Mean Basin slope (°)[a] | Mean Annual | % of Tributary with |
| | River name | Latitude (°) | Longitude (°) | | | | | Precipitation (mm y⁻¹)[b] | >"High" Erosivity[c] |
|---|---|---|---|---|---|---|---|---|---|
| TG-1 | George River | -41.29017 | 148.22217 | 346 | 397.25 | 218.0 | 10 | 1,310 | |
| TG-2 | Forester Creek | -41.27183 | 148.19925 | 141 | 40.21 | 120.0 | 6 | 1,020 | 9.2 |
| TG-3 | Powers Creek | -41.28286 | 148.13247 | 265 | 55.56 | 214.8 | 10 | 1,195 | 38 |
| TG-4 | Groves Creek | -41.25514 | 148.08317 | 364 | 34.39 | 238.0 | 11 | 1.336 | 49.5 |
| TG-5 | Ransom River | -41.25364 | 148.08239 | 347 | 27.71 | 226.8 | 10 | 1.312 | 48.8 |
| TG-6 | North George River | -41.28067 | 148.00697 | 439 | 65.84 | 275.5 | 12 | 1,442 | 49.3 |
| TG-7 | South George River | -41.32208 | 147.92172 | 652 | 42.53 | 211.5 | 9 | 1,743 | 26.9 |
| TG-8 | Mt. Albert Rivulet | -41.32178 | 147.92592 | 596 | 20.42 | 227.8 | 10 | 1,663 | 40.4 |
| TG-9 | George River @ St. Helens | -41.31350 | 148.26531 | 331 | 426.88 | 213.5 | 10 | 1,289 | |

[a] Based or derived from Satellite Radar Topography Mission data, 90 m resolution (Gallant et al., 2011). Mean local relief calculated using a 10-cell (~900 m) circular moving window.

[b] Used in the calculation of the meteoric $^{10}$Be delivery rate, $^{10Be}F_{met}$, for each catchment (Graly et al., 2011). Calculated using the basin average elevation and using the regression equation between elevation and mean annual precipitation at Australian Bureau of Meteorology stations (Figures 2, 3; Table 1).

[c] Erosivity ratings from Kidd et al. (2014, 2015).

**Table 2: Locations of sample collection sites and upstream stream basin characteristics.**

| In situ Sample ID | UVM Batch No. | Quartz mass (g) | Carrier mass (μg) | LLNL Sample ID | $^{10}Be_i/^9Be_{carr}$ | $\pm 1\sigma$ | $^{10}Be_i$ (atoms g$^{-1}$) | $\pm 1\sigma$ |
|---|---|---|---|---|---|---|---|---|
| TG-1 | 432[a] | 20.099 | 250.791 | BE28820 | 4.37E-13 | 7.83E-15 | 3.64E+05 | 6.52E+03 |
| TG-2 | 438[b] | 20.100 | 249.506 | BE29129 | 6.83E-13 | 9.31E-15 | 5.66E+05 | 7.72E+03 |
| TG-3 | 438 | 22.423 | 249.704 | BE29130 | 4.79E-13 | 1.41E-14 | 3.97E+05 | 1.17E+04 |
| TG-4 | 438 | 19.288 | 248.814 | BE29131 | 3.10E-13 | 8.41E-15 | 2.56E+05 | 6.95E+03 |
| TG-5 | 438 | 20.702 | 250.296 | BE29133 | 4.37E-13 | 1.02E-14 | 3.63E+05 | 8.48E+03 |
| TG-6 | 446[c] | 20.532 | 249.209 | BE29303 | 2.81E-13 | 6.11E-15 | 2.33E+05 | 5.05E+03 |
| TG-7 | 446 | 20.156 | 249.111 | BE29304 | 2.28E-13 | 6.76E-15 | 1.88E+05 | 5.60E+03 |
| TG-8 | 446 | 20.747 | 249.704 | BE29305 | 2.99E-13 | 7.35E-15 | 2.48E+05 | 6.10E+03 |
| TG-9 | 446 | 20.169 | 250.791 | BE29306 | 4.94E-13 | 1.19E-14 | 4.11E+05 | 9.92E+03 |

| Meteoric Sample ID | UVM Batch No. | Sample mass (g) | Carrier mass (μg) | LLNL Sample ID | $^{10}Be_m/^9Be_{carr}$ | $\pm 1\sigma$ | $^{10}BeF_{met}$ (atoms cm$^{-2}$ y$^{-1}$) | $^{10}Be_m$ (atoms g$^{-1}$) | $\pm 1\sigma$ | $^9Be_{min}$ (atoms g$^{-1}$) | $^9Be_{reac}$ (atoms g$^{-1}$) |
|---|---|---|---|---|---|---|---|---|---|---|---|
| TG-2 | MB-15[d] | 0.463 | 328.71 | BE27783 | 1.51E-12 | 2.07E-14 | 8.55E+05 | 7.16E+07 | 9.83E+05 | 2.51E+16 | 1.32E+16 |
| TG-3 | MB-15 | 0.497 | 298.02 | BE27784 | 1.50E-12 | 2.26E-14 | 1.00E+06 | 5.99E+07 | 9.05E+05 | 3.19E+16 | 1.06E+16 |
| TG-4 | MB-15 | 0.457 | 296.04 | BE27785 | 1.12E-12 | 1.55E-14 | 1.12E+06 | 4.84E+07 | 6.69E+05 | 3.29E+16 | 1.08E+16 |
| TG-5 | MB-15 | 0.491 | 300.00 | BE27786 | 1.05E-12 | 1.46E-14 | 1.10E+06 | 4.29E+07 | 5.95E+05 | 2.84E+16 | 1.09E+16 |
| TG-6 | MB-15 | 0.466 | 300.99 | BE27787 | 4.30E-12 | 5.79E-14 | 1.21E+06 | 1.85E+08 | 2.50E+06 | 4.54E+16 | 4.06E+16 |
| TG-7 | MB-15 | 0.487 | 299.01 | BE27788 | 5.60E-12 | 6.09E-14 | 1.46E+06 | 2.30E+08 | 2.50E+06 | 3.09E+16 | 5.82E+16 |
| TG-8 | MB-15 | 0.487 | 300.00 | BE27789 | 5.35E-12 | 5.83E-14 | 1.39E+06 | 2.20E+08 | 2.40E+06 | 2.71E+16 | 5.54E+16 |
| TG-9 | MB-15 | 0.541 | 299.01 | BE27790 | 1.19E-12 | 1.64E-14 | 1.08E+06 | 4.39E+07 | 6.06E+05 | 1.53E+16 | 1.08E+16 |

[a] In situ Batch 432 Blank $^{10}Be_i/^9Be_{carr}$ ratio = 1.25 x 10$^{-14}$ ± 5.87 x 10$^{-16}$

[b] In situ Batch 438 Blank $^{10}Be_i/^9Be_{carr}$ ratio = 1.22 x 10$^{-14}$ ± 1.82 x 10$^{-15}$

[c] In situ Batch 446 Blank $^{10}Be_i/^9Be_{carr}$ ratio = 1.27 x 10$^{-14}$ ± 6.70 x 10$^{-16}$

[d] Meteoric Batch MB-15 Blank $^{10}Be_m/^9Be_{carr}$ ratio = 1.65 x 10$^{-14}$ ± 1.72 x 10$^{-15}$

200

**Table 3: In situ and meteoric beryllium isotope data used to derive $^{10}Be_i$-based erosion rates and $^{10}Be_m/^9Be_{reac}$-based denudation rates for this study.**

More recently, decades of intensive tin lode mining in isolated headwaters of some tributaries and pockets of hydraulic
sluice mining for tin in lowland floodplains introduced $>10^6$ m$^3$ of tailings to the George River and its tributaries (Fig. 2a).
Knighton (1991) notes that the pre-mining average grain-size of alluvium for the George River was 30–50 mm, and that this
was reduced to 1–2 mm during the mining era; however, it is not clear whether the 30–50 mm average grain size was
specific to one sample site, or for the George River as a whole. Knighton (1991) notes that bedload characteristics have since
returned to their pre-disturbance values following widespread alluvium storage in floodplains and aggradation at the George
River delta in Georges Bay (Cheetham and Martin, 2018; Martin and Cheetham, 2018). Despite the George River's return to
pre-disturbance channel and bedload characteristics, a study from an experimental forest in the Gentle Annie tributary to the
George River shows that sediment yields from logged plots continue to be elevated relative to sediment yields from
unlogged plots (Wilson, 1999). More recently, land use within the George River basin in 2008, at the time of sample
collection, consisted primarily of forestry production from relatively natural environments and secondarily of conservation
land (Fig. 4); intensive land use (i.e., built structures, permanent land alteration) and agricultural production from unirrigated
land occur in equal proportion, though much less than the primary and secondary land uses. Only a small percentage of the
George River basin is used for agricultural production from irrigated lands (ABARES, 2016).

## 3 Methods

### 3.1 Sample collection and measurement

Sediment samples for this study were collected in 2008 from several locations along the trunk ($n = 2$) and tributaries ($n = 7$)
of the George River (Fig. 2; Table 2), upstream of which channels are generally concave-up and therefore in geomorphic
steady-state (Fig. 5). At each site, sediment was collected from the streambed and/or in-channel bars to ensure active fluvial
transport and mixing. Samples were sieved in the field to the 250–850 μm grain-size fraction. Although this grain-size is
finer than the mean natural grain size (30–50 mm; Knighton, 1991), previous studies show that $^{10}Be_i$ grain-size bias is
minimal or not present in small, low-elevation, low-relief, temperate landscapes where landslides are uncommon (van
Dongen et al., 2019); thus, $^{10}Be_i$ measured from the 250–850 μm grain-size fraction at George River can be interpreted as a
geological erosion rate.

$^{10}Be_m$ and the weathered reactive and silicate-bound *in situ* phases of $^9Be$ ($^9Be_{reac}$, $^9Be_{min}$, respectively) were measured only
from the 250–850 μm grain-size fraction from all seven tributary sites (TG-2 through TG-8) and one of the trunk channel
sites (TG-9). When $^{10}Be_m$ is normalized to $^9Be_{reac}$ following von Blanckenburg et al.'s (2012) denudation rate equation,
grain-size biases in resulting $^{10}Be_m/^9Be_{reac}$-based denudation rates are diminished (Wittmann et al., 2012). Singleton et al.
(2016) also showed the diminishment of grain-size bias in stream sand for $^{10}Be_m$ measurements when normalized to $^9Be_{reac}$.
Although it is possible to calculate erosion rates from $^{10}Be_m$ alone (Brown et al., 1988; Harrison et al., 2021; Willenbring and
von Blanckenburg, 2010), this method does not include any normalization to $^9Be_{reac}$, and $^{10}Be_m$ erosion rates are thus
susceptible to grain-size bias, especially if the full grain-size distribution is not known and/or has not been analysed. As our

samples are of one grain-size fraction and were collected and sieved in the field prior to $^{10}Be_m$ erosion rate derivations, we only present $^{10}Be_m/^{9}Be_{reac}$-based denudation rates in this study.

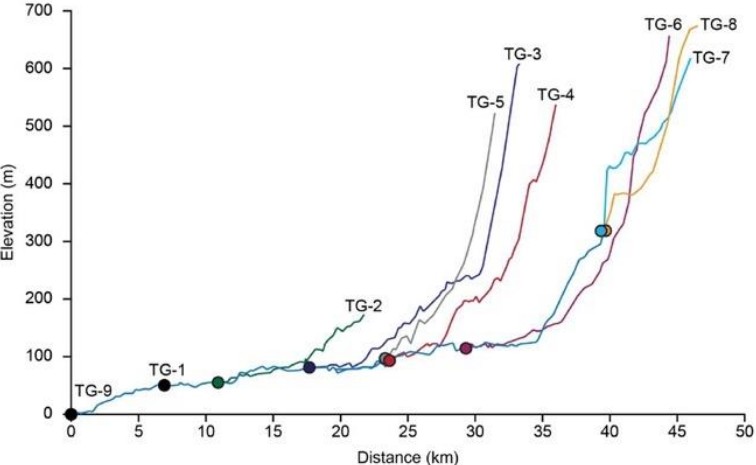

**Figure 5: Stream profiles of sampled sites along the George River trunk channel (TG-1, TG-9) and its tributaries (TG-2 through TG-8).**

$^{10}Be_i$ was extracted at the University of Vermont from quartz from each sample following standard methods, during which a known amount of a $^{9}Be$ carrier ($^{9}Be_{carr}$) was added to each sample (Kohl and Nishiizumi, 1992; Corbett et al., 2016); relative to the amount of $^{9}Be_{carr}$, no significant native Be was found in quartz concentrates from any sample, which can otherwise lead to significant overestimates of $^{10}Be_i$-based erosion rates (Portenga et al., 2015). $^{10}Be_i/^{9}Be_{carr}$ ratios were measured by accelerator mass spectrometry at the Lawrence Livermore National Laboratory CAMS facility (Table 3); $^{10}Be_i$ measurements were blank-corrected (the average ratio of three blanks was subtracted from the ratio of each unknown sample) and normalized to the 07KNSTD3110 AMS $^{10}Be$ standard material, which has a nominal $^{10}Be/^{9}Be$ ratio of 2.85 x $10^{-12}$ (Nishiizumi et al., 2007). $^{10}Be_i$ production was averaged across all sampled basins to a single point following Portenga and Bierman (2011), and the online erosion rate calculator described by Balco et al. (2008), which has been subsequently updated, was used to derive $^{10}Be_i$ erosion rates following the Lal (1991) and Stone (2000) scaling schemes ($\varepsilon$, Tables 4, 5). Here, $\varepsilon$ is presented in units of Mg km$^{-2}$ ky$^{-1}$ (Table 5) allowing us to compare measures of $\varepsilon$ directly with $^{10}Be_m/^{9}Be_{reac}$-based denudation rates ($D_m$; see below). Muogenic production of $^{10}Be_i$ is incorporated into $\varepsilon$; however, muogenic $^{10}Be_i$ is negligible relative to spallogenic $^{10}Be_i$ production given the George River's post-orogenic, low-elevation, low-relief setting.

$^{10}Be_m$ was extracted following Stone's (1998) fusion method and a $^{9}Be$ carrier solution was added to each sample. Through this process, some amount of $^{10}Be_i$ from bulk sediment is incorporated into the $^{10}Be_m$ sample; however, the amount of $^{10}Be_i$ is negligible, consistently two orders of magnitude less than $^{10}Be_m$ measurements (Table 3). $^{10}Be_m/^{9}Be_{carr}$ ratios of these fusion extracts were measured at the Lawrence Livermore National Laboratory CAMS facility, blank-corrected (ratio of one blank was subtracted from ratio of each unknown sample; Table 3) and normalized to the 07KNSTD3110 standard material

(Nishiizumi et al., 2007). Sample material used to calculate $^9Be_{reac}$ was first subject to 6N HCl acid leaching to remove sediment grain coatings (Greene, 2016; Portenga et al., 2019 supplement); it was then fully digested in HF and $^9Be_{min}$ was

measured in that solution. Both $^9Be_{reac}$ from sediment grain coatings and $^9Be_{min}$ from the remaining mineral material were measured by inductively coupled plasma-optical emission spectrometry (ICP-OES) at the University of Vermont. Together, these data were used to derive denudation rates following von Blanckenburg et al. (2012; Table 4); two variables required to calculate denudation rates that we did not, or were not able to measure, are the deposition rate of meteoric $^{10}Be$ ($^{10Be}F_{met}$) and the amount of $^9Be$ that is naturally occurring in bedrock ($^9Be_{parent}$). We use estimated values of $^{10Be}F_{met}$ based on global

deposition models presented in Graly et al. (2011) because it provides an estimate of $^{10Be}F_{met}$ that is specific for each sampled basin (8.55 x $10^5$ atoms $cm^{-2}$ $y^{-1}$ to 1.46 x $10^6$ atoms $cm^{-2}$ $y^{-1}$). At the time of sample collection (2008), the equations to calculate $D_m$ had not been published, and bedrock samples from the field area were not collected. We therefore use a value of 4.1 ppm for the amount of $^9Be_{parent}$ in our samples because the George River basin is underlain by biotite granites, and the average $^9Be_{parent}$ value of biotite granites comprising a subset from over 200 felsic intrusions measured across China and the

Soviet Union in the mid-1900s was reported to be 4.1 ppm (Beus, 1962; additionally reported in Sainsbury, 1964). We discuss the use of Graly et al.'s (2011) $^{10Be}F_{met}$ estimates and Beus's (1962) average $^9Be_{parent}$ for biotite granites in the Discussion section. In this study, $D_m$ is presented in units of Mg $km^{-2}$ $y^{-1}$ (Table 5).

### 3.3 Topographic, climatic, and anthropogenic characterization of the George River basin

We compare $\varepsilon$ and $D_m$ to various topographic and land-use factors to assess possible processes driving or related to background landscape evolution in the George River (Tables 1, 2). Topographic data are derived from the SRTM 90-m resolution global dataset (Gallant et al., 2011). Mean local relief was calculated over a moving 10-cell (~900 m) circular window. We do not compare $\varepsilon$ or $D_m$ to climate data from global gridded datasets for mean annual temperature and mean annual precipitation, although such data are available. This is because the gridded datasets are all models based on limited

measurements and include a strong elevation component in their interpolation scheme (e.g. WorldClim, Fick and Hijmans, 2017) or have spatial resolutions that do not provide sufficient detail for the small size of the George River basin (e.g. TRMM, Huffman, 2021). These characteristics of gridded climate datasets makes it difficult to attribute erosion to climatic drivers independent of their self-correlation with elevation. Thus, we rely on observed relationships between elevation and precipitation and temperature data from precipitation gauges ($n = 10$, each with >4 years of daily data; Table 1, Figs. 2, 3)

and temperature loggers ($n = 5$, each with >2 years of hourly data from at least 30% of days reporting [average = 70% of days reporting]; Table 1, Figs. 2, 3). Although the spatial coverage of rainfall gauges and temperature loggers is small relative to the coverage of interpolated, modelled, gridded data, they provide us an opportunity to work with measured, basin-specific data.

| Equation | Variable | Description | Unit |
|---|---|---|---|
| | $\varepsilon$ | $^{10}Be_i$ erosion rate | cm y$^{-1}$ |
| *$^{10}Be_i$ Erosion Rate* | $\Lambda$ | Attenuation length for cosmic-ray penetration[a] | 160 g cm$^{-2}$ |
| | | Production rate of $^{10}Be_i$ at Earth's surface[b] | |
| $$\varepsilon = \Lambda\left(\frac{P_0}{N} - \lambda\right)$$ | $P_0$ | | atoms g$^{-1}$ y$^{-1}$ |
| | $N$ | Measured concentration of *in-situ* produced $^{10}Be_i$ | atoms g$^{-1}$ |
| | $\lambda$ | $^{10}Be$ decay constant[c] | y$^{-1}$ |
| | $^{10Be}F_{met}$ | Atmospheric $^{10}Be_m$ delivery rate | atoms cm$^{-2}$ y$^{-1}$ |
| *$^{10}Be_m/^9Be_{reac.}$ Denudation Rate[d]* | $^{10}Be_m$ | Measured concentration of $^{10}Be_m$ extracted from sediment grain coatings | atoms g$^{-1}$ |
| $$D_m = \frac{^{10Be}F_{met}\left(\frac{^9Be_{min}}{^9Be_{reac}}+1\right)}{\left(\frac{^{10}Be_m}{^9Be_{reac}}\right)\,^9Be_{parent}}$$ | $D_m$ | $^{10}Be_m/^9Be_{reac}$-based denudation rate | g cm$^{-2}$ y$^{-1}$ |
| | $^9Be_{min}$ | Measured concentration of $^9Be$ still within mineral grains | atoms g$^{-1}$ |
| | $^9Be_{reac}$ | Measured concentration of $^9Be$ extracted from sediment grain coatings | atoms g$^{-1}$ |
| | $^9Be_{parent}$ | Assumed concentration of $^9Be$ in crustal bedrock[e] | 1.671 x 10$^{17}$ atoms g$^{-1}$ |

[a] Balco et al. (2008), Gosse and Phillips (2001)

[b] Scaled for each basin following Lal (1991) and Stone (2000)

[c] Half-life of $^{10}Be$ = 1.36 My

**Table 4: Equations and variables used in this study to derive $^{10}Be_i$-based erosion rates and $^{10}Be_m/^9Be_{reac}$-based denudation rates from beryllium data.**

**Table 5.** *In Situ* [10]Be Erosion Rates and Meteoric [10]Be Denudation Rates

| Sample ID | [10]Be$_i$ erosion, $\varepsilon^a$ (Mg km$^{-2}$ y$^{-1}$) | $\pm 1\sigma^b$ | Integration duration (ky) | [10]Be$_m$/[9]Be$_{reac}$ denudation rate, D$_m$ (Mg km$^{-2}$ y$^{-1}$) | $\pm 1\sigma$ |
|---|---|---|---|---|---|
| TG-1 | 25.9 | 2.2 | 61.8 | | |
| TG-2 | 13.1 | 1.1 | 122.5 | 16.7 | 0.2 |
| TG-3 | 21.7 | 1.9 | 73.7 | 25.9 | 0.4 |
| TG-4 | 38.1 | 3.2 | 42.1 | 36.9 | 0.5 |
| TG-5 | 25.8 | 2.2 | 62.0 | 36.8 | 0.5 |
| TG-6 | 45.1 | 3.8 | 35.5 | 20.5 | 0.3 |
| TG-7 | 66.2 | 5.7 | 24.2 | 20.7 | 0.2 |
| TG-8 | 47.5 | 4.0 | 33.7 | 19.1 | 0.2 |
| TG-9 | 22.4 | 1.9 | 71.5 | 23.4 | 0.3 |

$^a$ [10]Be$_i$ erosion rates calculated using the CRONUS erosion rate calculator version 3.0,

wrapper version 3.0, erates version 3.0, muons version 3.1 (Balco et al., 2008).

**Table 5: New [10]Be$_i$-based erosion rates and [10]Be$_m$/[9]Be$_{reac}$-based denudation rates presented in this study.**

Proper interpretation of [10]Be$_m$ derived denudation rates requires an understanding of the potential for beryllium weathering and desorption from sediment grain coatings and mobility through regolith (von Blanckenburg et al., 2012). To this end, information on (1) the depth of regolith and (2) chemical weathering data across the George River basin is needed. A potentially relevant dataset available for Tasmania is an interpolated gridded map of depth of regolith (Wilford et al., 2016). However, like the WorldClim precipitation and temperature datasets, the gridded regolith dataset was created by interpolating measured data from around Australia using a model and has an implicit dependence on elevation that does not reflect measured depths to bedrock in the George River basin. Only three boreholes exist in the George River basin that clearly go through regolith to bedrock, from which we extracted regolith depth (BoM, 2015; Fig. 2A; Table 1). They do not match the model results. These three boreholes, and others in the study area, have some units that could be alluvium or regolith; this differentiation is not clear and therefore the depth of regolith could be overestimated if alluvium is marked as regolith. Thus, we do not know with certainty the depth of regolith across our field area and we therefore cannot draw any clear conclusions about beryllium mobility in deep, weathered soils from the borehole data alone and do not explore it further.

Qualitative ratings of soil erosivity have previously been determined for Tasmania (Kidd et al., 2014, 2015) based on modelled soil loss should substantial vegetation and ground cover be removed; these ratings are strongly tied to hillslope angle within the George River basin (Fig. 6). Additionally, slope and erosion are strongly linked across the Great Dividing

Range on the Australian mainland (Codilean et al., 2021). Thus, we compare erosion and denudation metrics against basin slope metrics, which enables us to compare our measures of $\varepsilon$ and $D_m$ to basin slope to assess how Kidd et al.'s (2014, 2015) metrics for of hillslope erodibility and erosion in the George River are related and to compare these new $^{10}Be_i$ erosion rates to those presented by Codilean et al. (2021) for the Australian mainland.

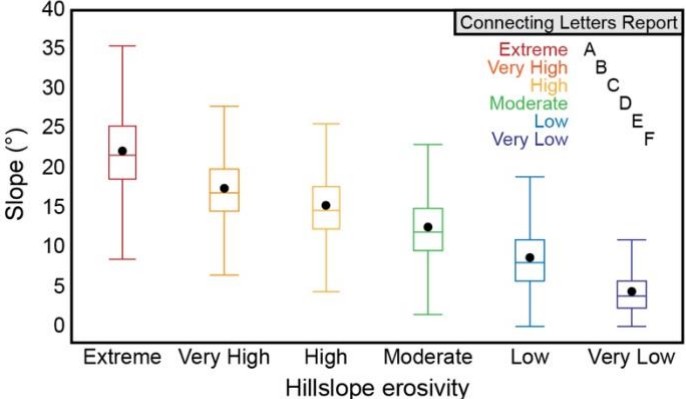

**Figure 6: Analysis of variance, showing hillslope angles associated with categories of landscape erosivity (Kidd et al., 2014, 2015) at George River. Box-and-whiskers cover ±1.5x the interquartile range; black dot is the mean slope for the erosivity category. The mean slope for each erosivity category is significantly different from every other category, illustrated by the Connecting Letters Report (if the mean slope in any two erosivity categories were statistically indistinguishable, they would otherwise share a letter in**
**the report). We therefore use hillslope angle as a quantitative proxy for erosivity in the George River basin.**

### 3.4 Calculating the dissolved and suspended loads of George River at St. Helens

There is one long-term water quality and stream gauging station in the George River basin at the inlet to the local water treatment plant drawing water from the trunk channel of the George River in the town of St. Helens (Fig. 2). Thus, we can
only estimate chemical weathering for the entire George River basin, not individual tributaries. Chemical weathering rates for the George River at St. Helens were calculated using these water quality data (i.e., dissolved major and trace element data) and discharge data (J. Fawcett, TasWater, pers. comm. 2021). Discharge measurements were taken at intervals ranging from 4 to 96 times per day from 1968 to 2021; 26 complete years of discharge data were available. Water quality measurements have been conducted since 2015 and we used the data from July 2015 to September 2021 in our derivation of
chemical weathering for the George River basin.

We matched water quality measurements with the nearest discharge measurement in time; when times did not line up exactly, we used the average of the nearest two discharge measurements. We then explored the relationship between discharge and each water quality parameter. For parameters that are invariant with discharge (iron, potassium, sulphate,
silica), we calculated the mean concentration of the parameter (Table 6). For parameters that scale with discharge (calcium, magnesium), we used a rating curve to determine how discharge relates to each water quality parameter; we then applied the mean measured values and rating curves, as appropriate, to every discharge measurement for years with complete discharge records (Table 6). Sodium and chlorine were balanced (suggesting a sea salt contribution) and were thus omitted from the

calculation. Carbonate that balanced the calcium and magnesium present was included; the rest was assumed to be from

atmospheric sources (Table 6). Silica concentrations were measured independently, once annually from 1974 to 1981 (J. Fawcett, TasWater, pers. comm. 2021), and we used all eight of those measurements; measurements in individual years were taken in March, June, August, October, and November (Table 6). We report total dissolved solids (TDS) measurements that are the sum of potassium, sulphate, silica, calcium, magnesium, and carbonate concentrations following West et al. (2005)'s chemical weathering rate calculation. We used a similar method to calculate the total suspended sediment (TSS) for each year of complete discharge data; TSS scales with discharge and so we applied a rating curve (Table 6).

**Table 6. Water Quality Data for the George River at St. Helens**

| Parameter | Number of Datapoints | Calculation technique | Equation used | Mean value [ppm] |
|---|---|---|---|---|
| Iron | 25 | Mean value | | 0.45 |
| Potassium | 24 | Mean value | | 1.03 |
| Sulphate | 24 | Mean value | | 2.03 |
| Silica | 8 | Mean value | | 9.90 |
| Calcium | 24 | Rating curve | -0.06 * Discharge + 0.90 | |
| Magnesium | 24 | Rating curve | -0.045 * Discharge + 0.55 | |
| Carbonate | | Required to balance Ca and Mg | 1.5*Ca+2.5*Mg | |
| Total suspended solids | 25 | Rating curve | 0.66 * Discharge + 0.25 | |

**Table 6: Water quality data from St. Helens used to calculate the dissolved load for the George River basin.**

## 4 Results

### 4.1 $^{10}Be_i$ erosion rates, $\varepsilon$

Erosion rates, $\varepsilon$, based on measured concentrations of $^{10}Be_i$ range from 13.1 to 66.2 Mg km$^{-2}$ y$^{-1}$. They integrate landscape dynamics in the George River basin since ~24–122 ka (Table 4). The average $\varepsilon$ from tributaries (36.8 ± 1.3 Mg km$^{-2}$ y$^{-1}$) is greater than from either of the trunk channel samples (TG-1 = 25.9 ± 2.2 Mg km$^{-2}$ y$^{-1}$; TG-9 = 22.4 ± 1.9 Mg km$^{-2}$ y$^{-1}$). Tributary values for $\varepsilon$ are greater in the high-elevation, western headwaters of the George River basin and decrease systematically, eastwards towards the lower-elevation coast (Fig. 6; $R^2$ = 0.91, p < 0.001). Relationships between $\varepsilon$ in tributary catchments and mean local relief, mean basin slope, and the percent of each basin that is categorized as being greater than or equal to "High" Erosivity are weak and not significant ($R^2$ = 0.28, $R^2$ = 0.17, $R^2$ = 0.05, respectively, p ≥ 0.13). Taking the product of $\varepsilon$ and basin area provides us with the average annual mass loss for each catchment. Making the assumption of steady state and no change in storage over time, we can then compare mass export rates across the catchment. Following this approach, we find that a similar mass exited sampled tributaries (10,511 ± 394 Mg y$^{-1}$) as the mass that passes

through the trunk channel sites (TG-1 = 10,286 ± 859 Mg y$^{-1}$; TG-9 = 9,555 ± 817 Mg y$^{-1}$). This comparison suggests little to no contribution of mass from the lowland, mainstem George River valley below the tributaries and above the basin outlet sampling sites.

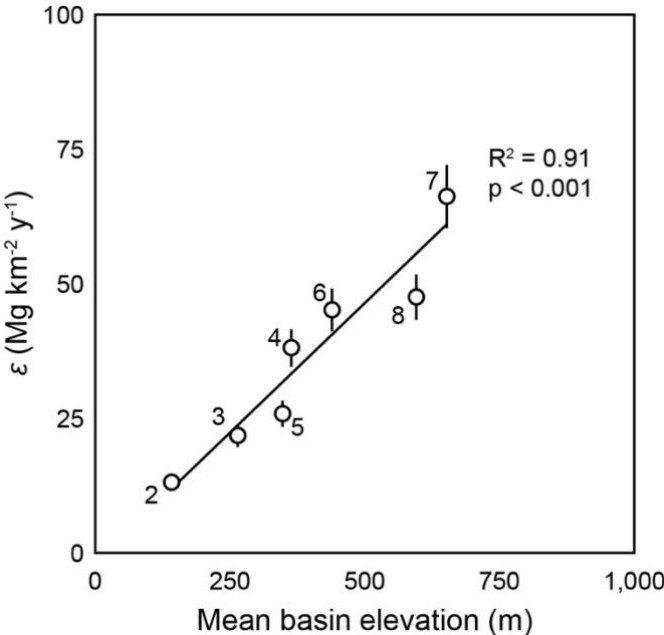

Figure 7: A strong correlation between $^{10}$Be$_i$ based erosion rates ($\varepsilon$) and mean basin elevation for the seven tributary samples collected in this study. We do not include trunk-channel sites because erosion rates here also incorporate erosion occurring in tributary catchments.

### 4.2 $^{10}$Be$_m$ denudation rates, $D_m$

Based on an assumed $^9$Be$_{parent}$ value of 4.1 ppm (Beus, 1962), $^{10}$Be$_m$/$^9$Be$_{reac}$-based denudation rates, $D_m$, range from 16.7 to 36.9 Mg km$^{-2}$ y$^{-1}$. Most values for $D_m$ in tributaries do not replicate well the $^{10}$Be$_i$-derived erosion rates, $\varepsilon$, with the exception of TG-4 (Fig. 8). The $^{10}$Be$_m$/$^9$Be$_{reac}$-based denudation rate at the trunk channel site, TG-9 (23.4 ± 0.3 Mg km$^{-2}$ y$^{-1}$), replicates the $^{10}$Be$_i$ erosion rate (22.4 ± 1.9 Mg km$^{-2}$ y$^{-1}$). In general, $^{10}$Be$_m$-based measures $D_m$ of tributaries are not significantly related to any topographic or basin metric such as mean basin elevation, mean local relief, or mean basin slope ($R^2$ = 0.12, $R^2$ = 0.06, $R^2$ = 0.11, respectively; p > 0.44). $^{10}$Be$_m$-based measures $D_m$ of tributaries appear to be moderately related to the percentage of each basin that Kidd et al. (2014, 2015) categorizes with a land use of "High" to "Extreme" erosivity, though we note this relationship is not significant at high-confidence levels ($R^2$ = 0.42; p = 0.18; Fig. 9).

### 4.3 Dissolved load and suspended sediment fluxes

The total dissolved sediment load in the George River at St. Helens for the 26 years between 1969 and 2020 is between 1,820 and 10,770 Mg y$^{-1}$ (mean = 4,400 ± 2,230 Mg y$^{-1}$, 1σ) and the total suspended sediment load ranges from 280 to

10,560 Mg y$^{-1}$ (mean = 1,830 ± 2,180 Mg y$^{-1}$, 1σ). The water treatment plant from which the dissolved load data were obtained is close to site TG-9, and data from this site allow us to place $^{10}$Be-inferred erosion and denudation rates for the whole George River basin in context. These data show that the dissolved load export rate averages to about 10.3 Mg km$^{-1}$ y$^{-1}$, which is <50% of ε (22.4 Mg km$^{-2}$ y$^{-1}$), based on decades of flow records and five years of discontinuous water sampling at the same sampling location. The suspended sediment export rate out of the George River basin is less, 4.3 Mg km$^{-2}$ y$^{-1}$.

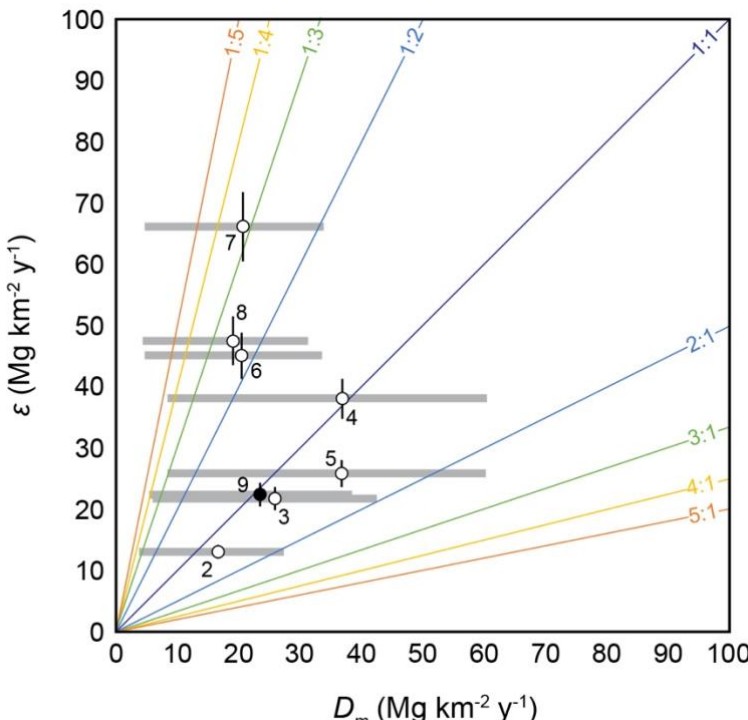

Figure 8: $^{10}$Be$_i$ based erosion rates (ε) compared $^{10}$Be$_m$/$^9$Be$_{reac}$-based denudation rates ($D_m$) for tributary basins (open circles) and the trunk channel site, TG-9 (closed circle). Measures of ε and $D_m$ are different at each site, but similar within a factor of three. $D_m$ is calculated using a $^9$Be$_{parent}$ value of 4.1 ppm, taken from an average values of a suite of biotite granites across the former Soviet Union and China (Beus, 1962; also reported in Sainsbury, 1964); horizontal grey bars, however, shows the range of $D_m$ values calculated using low estimates of crustal $^9$Be$_{parent}$ (2.5 ppm; high-end of $D_m$ values; von Blanckenburg et al., 2012) and the average $^9$Be$_{parent}$ value measured on S-type granites and those that are tin-bearing measured globally (18 ppm; n = 11; low-end of $D_m$ values; London and Evanson, 2018).

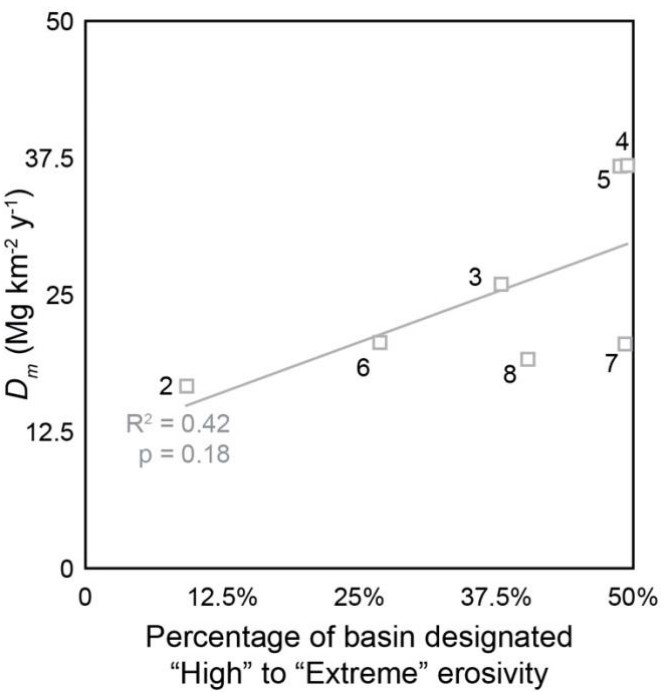

**Figure 9:** $^{10}Be_m/^9Be_{reac}$-based denudation rates, $D_m$, (gray squares) from tributary basins measured at George River are related to the percentage of the basin that is classified as "High," "Very High," or "Extreme" Erosivity (Kidd et al., 2014, 2015), though this relationship is not significant (p = 0.18). The basins with the highest denudation rates are those with histories of intensive mining and/or recent forestry, both of which disturb topsoils.

**5 Discussion**

The multi-methodological approach we employ in this study provides four new datasets, all of which quantify some component of landscape change at different spatial scales: (1) mass loss rates inferred $^{10}Be_i$ at seven tributary and two trunk channel sites, (2) denudation rates from $^{10}Be_m/^9Be_{reac}$ from seven tributary sites and one trunk channel site, (3) suspended

sediment export at the mouth of the George River, and (4) the dissolved load of the George River from the water quality and flow data at the mouth of the catchment. Comparing and interpreting these new datasets improves our understanding of the rate of landscape change over time in the George River basin. Given that the only location for which we have data from all four of datasets is at the mouth of the George River in St. Helens, we explore what the different rates presented in this study might mean for landscape change across the whole river basin, recognizing that without more data, we cannot be more

specific in our interpretation of $\varepsilon$ or $D_m$ at tributary sites beyond the traditional meanings of erosion or denudation, respectively.

**5.1 Relationships between $\varepsilon$, elevation-dependent climate conditions, and land use**

Erosion rates in the George River basin are strongly related to mean basin elevation, which varies greatly across the

catchment as the study area extends east from the Rattler Range and Mt. Victoria (1,213 m) to the coast at sea-level (Fig. 2).

In contrast, we find no evidence to suggest that $\varepsilon$ is related to slope in the George River over millennial timescales. This result differs from many studies, which show strong correlations between $\varepsilon$ and mean basin slope at a global scale (Portenga and Bierman, 2011) and at regional scales across the Great Dividing Range on Australia's mainland (Fig. 10; Codilean et al., 2021; Nichols et al., 2014). Our results also differ from prior assessments of the George River basin using measured climate data, bedrock structure, topographic analysis, water quality models, and geographical landscape characterization that suggest slope imparts a large control over erosion and sediment generation in the catchment on human timescales (Jerie et al., 2003; Kragt and Newham, 2009).

Any process-based explanation for the correlation of erosion rates with elevation requires that we consider how relevant geomorphic and geochemical processes vary across the George River basin. Climatic data collected from stations in and near the George River basin indicate that both mean annual temperature and mean annual precipitation are strongly correlated with elevation (Fig. 3). At higher elevations, rocks are experiencing lower temperatures more frequently and receive more precipitation than lower elevations, increasing the potential for both mechanical (frost cracking) and chemical weathering (dissolution). Frost cracking rates are greatest in rocks where mean annual temperature is above freezing (which is the case for all of the George River basin) but temperatures go below freezing both long and frequently enough to crack rocks, which is also the case across much of the basin (Delunel et al., 2010; Hales and Roering, 2007). In the George River basin, the only temperature-related metric that correlates with elevation is mean annual temperature, in contrast to, for example, the time spent below freezing, likely because temperature inversions, with cold air drainage to lower elevation valleys, are common (Webb et al., 2018, 2020). Additionally, the underlying mechanics that lead to rock fracturing in the first place have been demonstrated to be strongly linked to climate and the availability of water (Eppes and Keanini, 2017; Eppes et al., 2018). While water is plentiful across the George River basin, we see that $\varepsilon$ is greater at higher elevations where rainfall is also greater, facilitating faster breakdown of rock.

Mean annual precipitation at meteorology stations in the George River basin varies less (2.7-fold) from low to high elevations (681–1,836 mm y$^{-1}$; Fig. 3) than $\varepsilon$ (4.8–24.5 mm ky$^{-1}$; a 5.1-fold difference; Table 4). The elevation-induced precipitation and erosion rate gradients we observe are consistent with suggestions made at regional and global scales that the relationship between slope and erosion becomes secondary to precipitation in low-slope, low-elevation, post-tectonic settings (Henck et al., 2011; Mishra et al., 2018). We note that Mishra et al. (2018) also suggest that at the global scale, the erosional effects of increased of precipitation may be balanced by increased vegetation cover, which serves to stymie erosion. However, the George River basin is densely vegetated throughout, and forests are no more prevalent at higher than lower elevations in our field area. We propose that in the George River basin, $\varepsilon$ is related to elevation in large part because precipitation is strongly correlated with elevation. This interpretation seems to hold true for bedrock outcrops, the erosion rates of which are most-closely correlated to mean annual rainfall in aseismic landscapes, globally; however, basin-wide erosion rates in aseismic areas globally remain more strongly correlated to mean basin slope and subsequently to elevation

and climate-related processes (Portenga and Bierman, 2011), which stands in contrast to the relationship we observe here between elevation and $\varepsilon$.

The concave-up geometry of sampled streams (Fig. 5) demonstrates that values of $\varepsilon$ presented here come from streams that are in steady state. Thus, the very strong relationship between elevation, climate (both mean annual rainfall and

temperature), and $\varepsilon$ would likely not have emerged had our $^{10}Be_i$ samples been affected by clast attrition (Carretier et al., 2009), deep-seated landslides (Aguilar et al., 2014; Gonzalez et al., 2016; Puchol et al., 2014), or intensive erosion associated with mining, forestry, or agriculture (Barreto et al., 2014; Neilson et al., 2017). Even intensive tin mining, which supplied $>10^6$ m$^3$ to the George River over the last two centuries (Knighton, 1991) seems not to have had a long-lasting diluting effect on $^{10}Be_i$ in sampled stream sediment. It is possible that mining efforts, especially sluice mining, did not lead to

$^{10}Be_i$ dilution because of the homogenizing effect of $^{10}Be_i$ in bioturbated soils (Brown et al., 1995; Schaller et al., 2018) or because the size of the George River basin is large enough to buffer the effects of mining efforts in a similar way that large catchments may buffer the effects of landslide material (Niemi et al., 2005; Yanites et al., 2009). It is also possible that mining activity did lead to $^{10}Be_i$ dilution, but concentrations have returned to pre-disturbance levels in the same way that bedload characteristics returned to pre-disturbance levels (Knighton, 1991) and similar to the rapid, two-year recovery of

$^{10}Be_i$ concentrations following storm-triggered landslides in Puerto Rico (Grande et al., 2021).

Overall, the close relationship between $^{10}Be_i$ erosion rates and climate across the George River basin demonstrates that $^{10}Be_i$ erosion rates reflect background, geologically-meaningful rates of landscape evolution on millennial timescales, even in areas with long histories of intensive human land-use (e.g., Barreto et al., 2014; Rosenkranz et al., 2018; Vanacker et al.,

2007). Secondarily, that higher values of $\varepsilon$ are observed where there is more rainfall and are colder temperatures suggests that more sediment is being generated per unit area in the western portion of the catchment. There, larger volumes of rainfall and colder temperatures facilitate the generation, erosion, entrainment, and delivery of more sediment to trunk channels than in the eastern portion of the catchment.

Since pre-disturbance stream flow and bedload conditions were re-established by the 1990s (Knighton, 1991), it appears the greatest risk of enhanced sediment flux from the George River to Georges Bay in the future comes from land-use changes involving the widespread disturbance of surficial soils, such as through forestry (Wilson, 1999). The percentage of land used for production forestry in native environments has been decreasing throughout the 21$^{st}$ century (Fig. 4). Although some land previously used for production forestry in native environments is being supplanted by conservation and protected native land

cover, which could buffer the effects of widespread erosion, much is being replaced by grazing and agriculture, which would likely increase erosion, particularly in the headwater catchments where geological erosion rates are naturally higher (Fig. 4). Given recent land-use trends, the $^{10}Be_i$ erosion rates presented here provide a useful benchmark level of sediment delivery to

the George River, Georges Bay, and other fluvial systems in northeast Tasmania that share topographic and geologic characteristics similar to those of the George River basin.


### 5.2 Considerations of $\varepsilon$ for trunk channel versus tributary sites

Taking the product of $\varepsilon$ and basin area calculates the annual mass exported from sampled basins. The mass leaving the tributaries (mean = 10,510 ± 390 Mg y$^{-1}$) is about the same as the mass passing through TG-1 (10,290 ± 860 Mg y$^{-1}$) and the mass of sediment leaving TG-9 (9,560 ± 820 Mg y$^{-1}$). We infer from these data that the $^{10}$Be$_i$ measured at TG-1 and TG-9

trunk channel locations is dominated by mass produced in the higher-elevation tributary basins with minimal sediment input from the George River valley bottoms. Similar interpretations have been made elsewhere, albeit in much larger river basins (i.e., Wittmann et al. 2009, 2011, 2016). Given these similarities, we average $\varepsilon$ from the two trunk channel sites to produce a nominal average erosion rate for the George River basin as a whole (24.1 ± 1.4 Mg km$^{-2}$ y$^{-1}$; or 8.9 ± 0.5 mm ky$^{-1}$ when dividing $\varepsilon$ by rock density, $\rho = 2.7$ g cm$^{-3}$), which is of the same magnitude to the average erosion rate of catchments

draining the eastern flanks of the Great Dividing Range along the southeastern passive margin of mainland Australia (11.6 mm ky$^{-1}$; Fig. 10; Codilean et al., 2021). Average $\varepsilon$ from the George River basin is most consistent with erosion of basins across the Bass Strait, which share similar topographic characteristics and geological histories as the George River basin (Codilean et al., 2021). The similarity between the geology, topography, and climate of newly-sampled basins and derived $^{10}$Be$_i$ erosion rates in Tasmania from this study and those from southeast mainland Australia supports the notion that

evolution of landscapes that share similar climatic, topographic, and geologic characteristics is similar.

### 5.3 Comparing $^{10}$Be$_i$-based erosion rates and $^{10}$Be$_m$-based denudation rates

Once delivered to Earth's surface in temperate regions, $^{10}$Be$_m$ concentrates in uppermost soil horizons (Graly et al., 2010; Willenbring and von Blanckenburg, 2010). This behaviour differs from that of $^{10}$Be$_i$, the concentration of which remains

homogenous in well-mixed, bioturbated soils for millennia (Jungers et al., 2009). Thus, any disturbance of large volumes of topsoil (i.e., agriculture, forestry, wildfire erosion, or mining activities) strips material with the highest concentrations of $^{10}$Be$_m$ and introduces that material into streams, a process similar to that identified following early land-use changes and deforestation in the Chesapeake Bay and San Francisco Bay (Portenga et al., 2019; Valette-Silver et al., 1986; van Geen et al., 1999). In contrast, the strong relationship between $^{10}$Be$_i$ erosion rates and elevation, and thus both precipitation and

temperature, across the George River basin (Fig. 7) suggests that $^{10}$Be$_i$ erosion rates, $\varepsilon$, are unaffected by land use.

Assuming a $^9$Be$_{parent}$ value of 4.1 ppm (Beus, 1962), calculated values of $D_m$ do not consistently replicate $\varepsilon$ (Fig. 8), nor does $D_m$ replicate the spatial patterns or yield the same relationships with topographic parameters that we observe with $\varepsilon$ in the small, geologically-homogeneous landscape of the George River basin (e.g., Fig. 7). In fact, we calculate similar $D_m$ values

at TG-2 and TG-7, which have the lowest and highest calculated values for $\varepsilon$ (Fig. 8). We know that decades-old historical mining activities and historical bushfires in the George River were restricted to lower catchment areas and tributaries where

measurements of $D_m$ are highest (Figs. 4, 9). Additionally, we infer from the moderate correlation observed between $D_m$ and the percent of tributary basins classified as "High" to "Extreme" Erosivity ($R^2 = 0.42$; Fig. 9; Kidd et al., 2014, 2015) that $^{10}Be_m/^9Be_{reac}$-derived denudation rates appear to be sensitive to recent land-use practices that disturb soils. The highest
denudation rates, $D_m$, we measured are those from basins with past histories of intense surface disruption through mining and forestry (i.e., TG-4, TG-5).

The similarity of $D_m$ and $\varepsilon$ for the entire George River basin, however—within a factor of 3 (Fig. 8)—provides general support for the hypothesis that $^{10}Be_m/^9Be_{reac}$-based denudation rates more-closely resemble $^{10}Be_i$-based erosion rates in small
river basins where geological heterogeneity is minimized. This observed similarity between $D_m$ and $\varepsilon$ supports the continued exploration and application of $^{10}Be_m/^9Be_{reac}$ denudation rates in geomorphological studies (i.e. Dannhaus et al., 2017; Deng et al., 2020; Portenga et al., 2019; Rahaman et al., 2017; Wittmann et al., 2012, 2015). However, data presented here suggest that this method should be used with caution in landscapes with recent soil disturbance.

**5.4 Sensitivity analysis of $^9Be_{parent}$ and $^{10Be}F_{met}$**

The values of $D_m$ we present in this study are calculated with assumed values for the amount of $^9Be$ naturally occurring in bedrock in the field area ($^9Be_{parent}$) and the rate at which meteoric $^{10}Be$ is delivered from the atmosphere to Earth's surface ($^{10Be}F_{met}$) because we did not measure these values specifically for the field area. Thus, we carry out a sensitivity analysis of both variables to assess how much $D_m$ responds to changes in these values:


Grew (2002) suggests that Earth's crustal average concentration of $^9Be_{parent}$ is 3 ppm, though it is not unheard of for $^9Be_{parent}$ to be <1 ppm in (ultra)mafic lithologies and that $^9Be_{parent}$ can range 10-fold within the same igneous complex. Von Blanckenburg et al. (2012), the study that first presents calculations for $D_m$, cite a slightly lower crustal average for $^9Be_{parent}$ of 2.5 ppm. London and Evensen (2002) present $^9Be_{parent}$ concentrations measured from felsic granites, which range from
1.6–160 ppm; for S-type granites or those that are tin-bearing – the same as the Blue Tier batholith in our field area (Higgins, 1985) – $^9Be_{parent}$ ranges from 2.3–130 ppm ($n = 11$, mean = 18 ppm). Additionally, Sainsbury (1964) presents data from a tin-bearing biotite granite in Alaska, showing that $^9Be_{parent}$ concentrations range from 2–26 ppm ($n = 5$, mean = 16.6 ppm). Thus, it seems a reasonable range of values for $^9Be_{parent}$ that might apply to bedrock in this study are as low as crustal averages (2.5 ppm) or as high as tin-bearing biotite granites elsewhere (>100 ppm). We choose to calculate and analyse $D_m$ from a more
modest estimate of 4.1 ppm (Beus, 1962) because single-digit concentrations of Be are most common for felsic igneous intrusions (London and Evensen, 2002). At lower $^9Be_{parent}$ concentrations (2.5 ppm), $D_m$ values across the George River basin increase such that $D_m$ values replicate $\varepsilon$ within a factor of two. However, when conservative, but higher $^9Be_{parent}$ concentrations are used (18 ppm; the average of values presented for S-type and tin-bearing granites presented by London and Evensen [2002]), $D_m$ values decrease across the field area such that all $D_m$ values are lower than $\varepsilon$ by at least a factor of
three (Fig. 8). The results of this sensitivity analysis highlights the importance of collecting representative bedrock samples

throughout a field area to ascertain appropriate measures of $^9Be_{parent}$ when using von Blanckenburg et al.'s (2012) $^{10}Be_m/^9Be_{reac}$ denudation rate method because of the highly-sensitive dependency of $D_m$ on $^9Be_{parent}$.

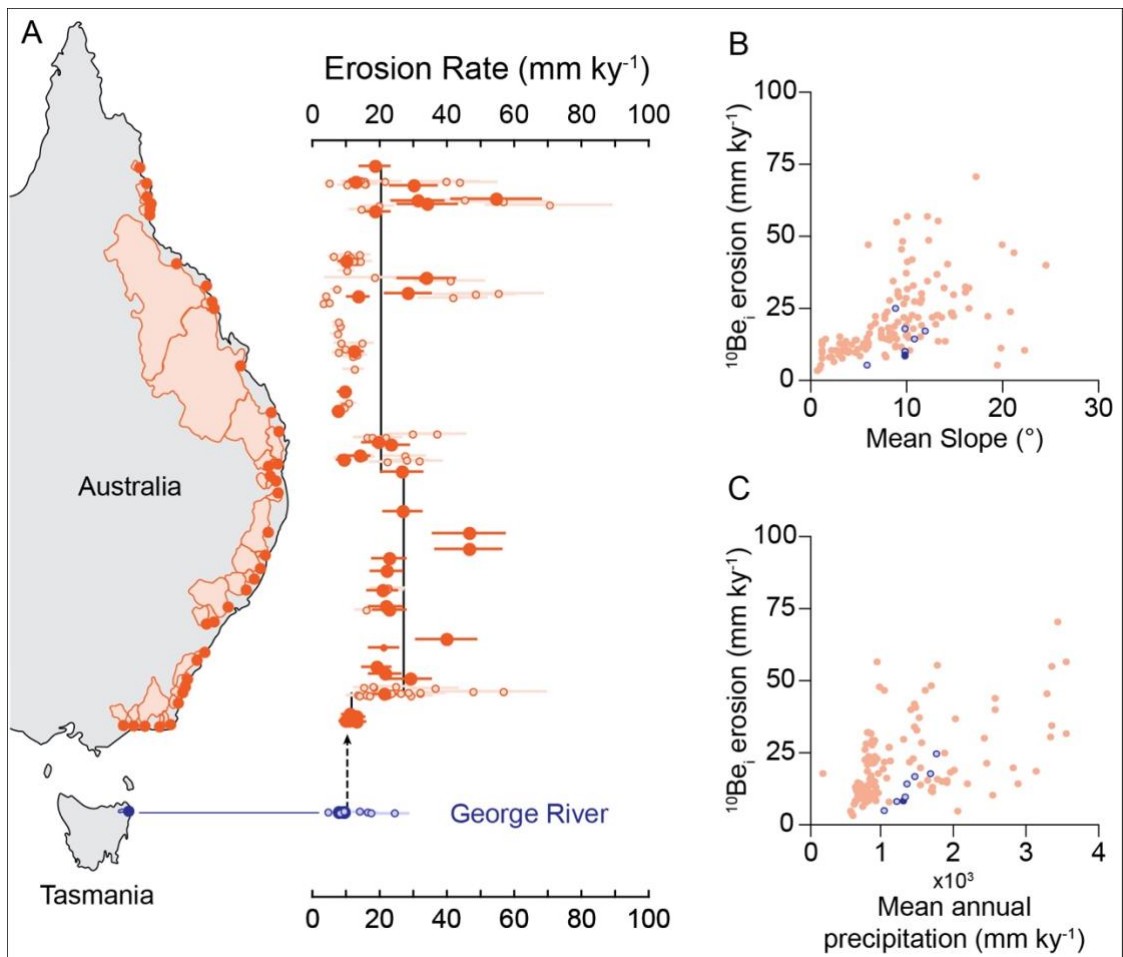

Figure 10: A. Map of river basins draining east off the Great Australian Escarpment, where $^{10}Be_i$ erosion rate data are available; figure adapted from Codilean et al. (2021). Filled circles are trunk streams and open circles are tributaries. Orange data include previously-published data (Codilean et al., 2021; Croke et al., 2015; Fülöp et al., 2020; Godard et al., 2019; Nichols et al., 2014; Tomkins et al., 2007). Blue data are new data presented in this study from the George River basin, Tasmania. The average $^{10}Be_i$ erosion rates from the George River (8.9 mm ky$^{-1}$) is consistent with erosion rates from southeast mainland Australia (average 11.6 mm ky$^{-1}$; Codilean et al., 2021). B. Comparison of $^{10}Be_i$ erosion rates from the George River basin (blue circles; tributaries) and the eastern flanks of the Great Australian Escarpment (orange circles) to basin average slope. C. Comparison of $^{10}Be_i$ erosion rates from the George River basin (blue circles) and the eastern flanks of the Great Australian Escarpment (orange circles) to mean annual precipitation; in this comparison, mean annual precipitation for George River samples comes from the elevation scaling for measured rainfall at meteorological gauging stations (Figs. 2, 3, Table 1) whereas Codilean et al. (2021) summarize precipitation data for mainland basins from the WorldClim database (Fick and Hijamans, 2017).

Values for the rate at which $^{10}Be_m$ is delivered from the atmosphere to Earth's surface ($^{10Be}F_{met}$) have been measured and modelled in various ways at both local and global scales, each with its own strengths. In the South Pacific region, for instance, Reusser et al. (2010a) directly measured $^{10Be}F_{met}$ in a dated New Zealand paleosol (1.68 to 1.72 x 10$^6$ atoms cm$^{-2}$ y$^{-}$

[1]) and Graham et al. (2003) report $^{10Be}F_{met}$ values measured from rainfall across New Zealand, finding a wider range of $^{10}Be_m$

deposition rates (1.7 to 2.9 x 10$^6$ atoms cm$^{-2}$ y$^{-1}$). In the absence of direct measurement, $^{10Be}F_{met}$ must be estimated or modelled. Heikkilä and von Blanckenburg (2015) integrate $^{10Be}F_{met}$ through the Holocene while others integrate $^{10Be}F_{met}$ for total atmospheric thickness, all at a global scale (Masarik and Beer, 2009; Willenbring and von Blanckenburg, 2010), but the resolution of these models is coarse, relative to the small spatial scale of this study, and $^{10Be}F_{met}$ would be the same for each sampled basin (1.0–1.5 x 10$^6$ atoms cm$^{-2}$ y$^{-1}$ for Holocene integrated or ~7 x 10$^5$ atoms cm$^{-2}$ y$^{-1}$ for atmospheric depth-

integrated $^{10Be}F_{met}$). Graly et al. (2011), however, present an equation that estimates $^{10Be}F_{met}$ from a location's mean annual precipitation and latitude, which provides a more specific value for $^{10Be}F_{met}$ for a given study site. We choose to use $^{10Be}F_{met}$ modelled from Graly et al.'s (2011) equation because of its ability to provide basin-specific values of $^{10Be}F_{met}$, but we present $D_m$ calculations for all basins using other $^{10Be}F_{met}$ values to assess the sensitivity of $D_m$ to $^{10Be}F_{met}$ (Fig. 11). In doing so, we find that $D_m$ calculated from Reusser et al.'s (2010a) and Graham et al.'s (2003) values of $^{10Be}F_{met}$ are consistently higher

than using the Graly et al.'s (2011) model, likely owing to precipitation rate differences between northeast Tasmania and New Zealand, thousands of kilometres away. $D_m$ calculated using $^{10Be}F_{met}$ values integrated through total atmospheric thickness (Masarik and Beer, 2009; Willenbring and von Blanckenburg, 2010), are consistently lower than those calculated using Graly et al.'s (2011) model, but those using $^{10Be}F_{met}$ values averaged through the Holocene (Heikkilä and von Blanckenburg, 2015) are remarkably consistent with results from the Graly et al. (2011) model. We suggest that the

consistency of $D_m$ modelled using the Graly et al. (2011) $^{10Be}F_{met}$ values and $D_m$ calculated using Heikkilä and von Blanckenburg's (2015) $^{10Be}F_{met}$ values provides support for our decision to use Graly et al.'s model. Additionally, we suggest our use of Graly et al.'s (2011) estimates of $^{10Be}F_{met}$ is reasonable because $D_m$ values using Graly et al.'s $^{10Be}F_{met}$ values plot between $D_m$ values calculated using $^{10Be}F_{met}$ values from both global climate models (Heikkilä and von Blanckenburg, 2015; Masarik and Beer, 2009; Willenbring and von Blanckenburg, 2010), at least for northeast Tasmania.


## 5.4 Where does the dissolved load originate in the George River basin?

If chemical weathering occurs primarily in the uppermost meters of the landscape, where most $^{10}Be_i$ is produced, then the erosion rate, $\varepsilon$, we calculate represents total landscape mass loss over time—a combination of physical and chemical mass loss. We could then partition $\varepsilon$ along the trunk channel at the mouth of the George River basin (TG-9 = 22.4 Mg km$^{-2}$ y$^{-1}$)

into mass flux removed in the measured dissolved load (10.3 Mg km$^{-2}$ y$^{-1}$) and the remainder, mass flux removed as solid sediment (12.1 Mg km$^{-2}$ y$^{-1}$). Of the sediment mass flux, it appears that 4.3 Mg km$^{-2}$ y$^{-1}$ is transported as suspended load (measured from water quality data) and the difference of 7.8 Mg km$^{-2}$ y$^{-1}$ is bedload (i.e., 12.1 Mg km$^{-2}$ y$^{-1}$ minus 4.3 Mg km$^{-2}$ y$^{-1}$). Our measure of $\varepsilon$ at TG-9 (22.4 Mg km$^{-2}$ y$^{-1}$) is similar to the $^{10}Be_m/^9Be_{reac}$ measure of denudation at this site (23.4 Mg km$^{-2}$ y$^{-1}$), which, if the assumptions of the method are met, represents total physical and chemical mass loss. Taken at face

value, $D_m$ assuming $^9Be_{parent}$ = 4.1 ppm is an accurate measure of total mass loss from the George basin at TG-9. However, given the wide range of $D_m$ possible using other reasonable values for $^9Be_{parent}$, it is difficult to know how the two measures of landscape change, $\varepsilon$ and $D_m$, truly compare.

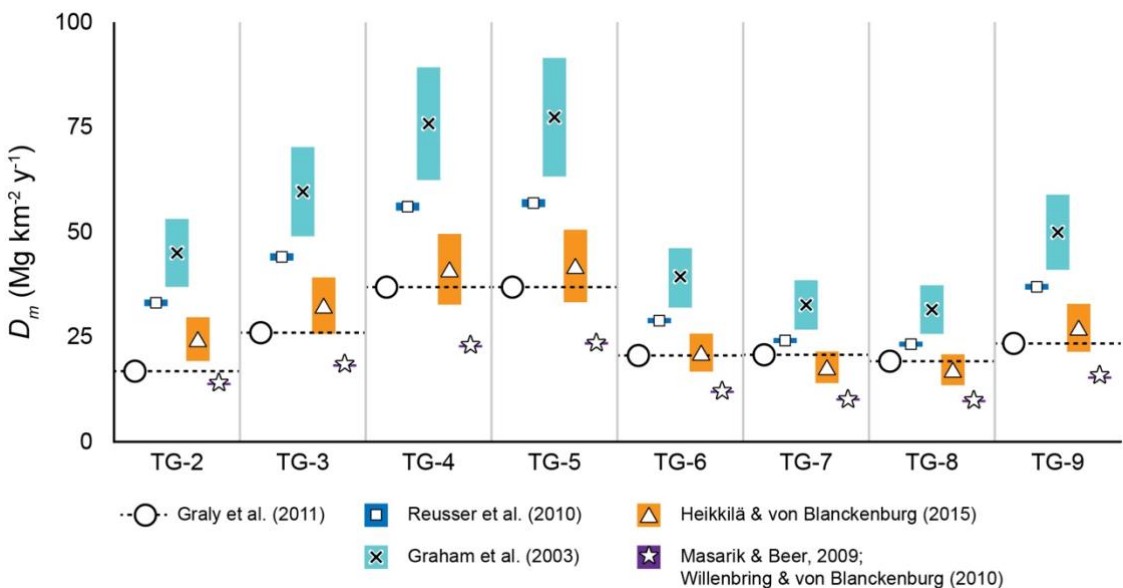

**Figure 11: Comparison of denudation rates, $D_m$, using the von Blanckenburg et al. (2012) method, a bedrock beryllium concentration, $^9Be_{parent}$, value of 4.1 ppm (Beus, 1962), measured values from stream sand (Table 3), and meteoric $^{10}Be$ delivery rates, $^{10Be}F_{met}$, of 1.68 x 10⁶ atoms cm⁻² y⁻¹ to 1.72 x 10⁶ atoms cm⁻² y⁻¹ (Reusser et al., 2010; blue, square), 1.9 x 10⁶ atoms cm⁻² y⁻¹ to 2.7 x 10⁶ atoms cm⁻² y⁻¹ (Graham et al., 2003; turquoise, black X), 1.0 x 10⁶ atoms cm⁻² y⁻¹ to 1.5 x 10⁶ atoms cm⁻² y⁻¹ (Heikkilä and von Blanckenburg, 2015; orange, triangle), ~7 x 10⁵ atoms cm⁻² y⁻¹ (Masarik and Beer, 2009; Willenbring and von Blanckenburg, 2010; purple, star), and 8.5 x 10⁵ atoms cm⁻² y⁻¹ to 1.5 x 10⁶ atoms cm⁻² y⁻¹ (Graly et al., 2011; white circle with dashed black line).**

If the majority of chemical weathering occurs below the penetration depth of most cosmic rays (< 2 m), then the chemical denudation and physical mass loss are at least in part and perhaps wholly disconnected. In this case, $\varepsilon$ (TG-9 = 22.4 Mg km⁻² y⁻¹) would need to be summed with the measured chemical mass flux (10.3 Mg km⁻² y⁻¹) and together (30.7 Mg km⁻² y⁻¹), they would estimate the total mass loss from the landscape. In this case, the summed total is ~24% greater than $D_m$ at TG-9. The presence of bedrock outcrops in some of the George River basin channels suggests that regolith thickness is limited in places and in that case, $^{10}Be_i$ measurements incorporate much of the chemical mass loss from the basin. However, the few boreholes that extend to unweathered bedrock ($n$ = 3; Fig. 2; Table 1) clearly indicate that regolith is deeper in some parts of the catchment. With the paucity of available data, we cannot determine how much of the dissolved load is coming from below the penetration depth of cosmic-ray neutrons but it could be significant.

Thus, despite the fact that $\varepsilon$ and $D_m$ at TG-9 are similar (22.4 Mg km⁻² y⁻¹ and 23.4 Mg km⁻² y⁻¹, respectively), $^{10}Be_{met}/^9Be_{reac}$-based denudation rates appear to have little correlation with landscape scale metrics suggesting that they do not reflect the rate of geomorphic processes controlling mass loss over time. $D_m$ is highest in basins with known histories of intensive land-use disturbance and high erosivity (Figs. 2, 8), a relationship that exists regardless of what value is used for $^9Be_{parent}$. In contrast, $\varepsilon$ is well-correlated to elevation and thus temperature and precipitation across the George River basin.

## 6 Conclusions

The $^{10}Be_i$-based erosion rates we present in this study are the first measured for any river system for Tasmania. In contrast to erosion across the Great Dividing Range on mainland Australia, where erosion rates and mean basin slope are closely linked, erosion in the George River basin has a strong relationship with mean basin elevation, and thus with mean annual precipitation and temperature, both of which are strongly correlated with elevation. The mean $^{10}Be_i$ erosion rate in the George River basin, $24.1 \pm 1.4$ Mg km$^{-2}$ y$^{-1}$, reflects erosion in tributaries to the George River where precipitation is greatest and temperatures are lowest; little sediment is generated in trunk channel valley bottoms. These findings support the notion that precipitation imparts a significant influence on landscape development in low-slope, low-elevation landscapes, which are often located in post-orogenic, passive margin settings. We also suspect that low but positive mean annual temperatures with frequent excursions below zero drives the mechanical breakdown of rock, thereby increasing sediment production in high-elevation basins through frost cracking. Although hillslope erosion associated with mining, agricultural, and forestry land-use practices occurred in the George River basin during the 19$^{th}$ and 20$^{th}$ centuries, $^{10}Be_i$ based erosion rates in the basin appear to reflect pre-disturbance rates of landscape change. Such rates are useful as part of Tasmania's current efforts to re-establish healthy and sustainable ecological conditions in its many estuarine environments, particularly those in northeast Tasmania where estuary tributaries have similar geological and topographic characteristics to those found in the George River basin. $^{10}Be_m/^9Be_{reac}$ denudation rates generally replicate $^{10}Be_i$-based erosion within a factor of three but show no correlation with landscape-scale metrics. Calculated $^{10}Be_m/^9Be_{reac}$ denudation rates are highly sensitive to the concentration of native beryllium in bedrock ($^9Be_{parent}$) and appear to be affected by intensive topsoil disturbance by mining, forestry, and agricultural land use. Data from the George River basin suggest that $^{10}Be_m/^9Be_{reac}$ denudation rates will be most meaningful in small, lithologically-homogeneous basins with limited amounts of topsoil disturbance and where the value of $^9Be_{parent}$ is well constrained by sampling and measurement of local bedrock.

## Acknowledgements

We acknowledge the Palawa peoples of lutruwita, the traditional custodians of the lands on which this work was completed.

## Data Availability

All maps were created by EWP; data within maps (e.g. DEMs, geology, etc.) is properly cited. All data used in this study and all data needed to reproduce our findings and the equations used to calculate $^{10}Be_i$ erosion rates and $^{10}Be_m/^9Be_{reac}$ denudation rates are presented in Tables 1–6. Mean annual precipitation and geological borehole data were gathered from online databases supported by the Australian Bureau of Meteorology (Rainfall: http://www.bom.gov.au/climate/data/; borehole: http://www.bom.gov.au/water/groundwater/explorer/index.shtml). Mean annual temperature data come from the State of Tasmania Air Temperature Logger Recording Database (© 2018 State of Tasmania), accessed through personal communication. Water quality data for the water intake station in St. Helens was provided by TasWater (pers. comm. John Fawcett). Maps throughout this book were created using ArcGIS® software by Esri.

**Author Contribution**

The conceptual analysis of the data presented in this paper comes from LAV's Undergraduate Honors Thesis (2020) at Eastern Michigan University. EWP, PRB, and AHS contributed to post-thesis manuscript revisions, data analysis, and figure

drafting. Samples and the $^{10}Be_i$ data presented here were collected and facilitated by PRB and ECL in 2008. $^{9}Be$ and $^{10}Be_m$ data were first presented in Sophie E. Greene's Master's Thesis (2016) at the University of Vermont; SEG declined a request to participate in the writing and publication of this paper. AHS completed chemical weathering calculations. AJH verified Lawrence Livermore National Laboratory's measurement of beryllium at the Center for Accelerator Mass Spectrometry in 2009. This work was performed in part under the auspices of the U.S. Department of Energy by Lawrence Livermore

National Laboratory under Contract DE-AC52- 07NA27344. This is LLNL-JRNL-825534.

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
