# Peer review of "Comparison of basin-scale *in situ* and meteoric 10Be erosion and denudation rates in felsic lithologies across an elevation gradient at the George River, northeast Tasmania, Australia"

_Geochronology, 2021_

## Author Comment (AC2)

**Comparison of basin-scale in situ and meteoric 10Be erosion and denudation rates across a rainfall, slope, and elevation gradient in felsic lithologies at George River, northeast Tasmania, Australia**

VanLandingham et al. submitted to *Geochronology*
Author Responses to Reviewer #2

**Reviewer #2**

This manuscript by VanLandigham et al. reports in-situ and meteoric beryllium-10 ($^{10}$Be) based erosion and denudation rates for a small catchment located in Tasmania. The study is presented as aiming at two main goals: evaluate the long-term sediment delivery to the estuarine zone, which has implications for policy regarding land use and coastal ecosystem preservation; compare the estimates of erosion / denudation yielded by the two "varieties" of $^{10}$Be.

Although I find the overall manuscript nicely written and well structured, I have concerns about (1) unclear terminology; (2) interpretation of meteoric $^{10}$Be based erosion rates - see detail provided below. In addition, I must say that I noticed is some "dissonance" between the way the scientific question is set up in the introduction (revolving much around the issue of increased delivery of sediment to coastal ecosystems under land use change in modern times), and what is discussed in the rest of the paper (controls on long-term erosion rates, and more methodological aspects about the comparison between in situ and meteoric $^{10}$Be). As a consequence, the manuscript would benefit from a clarification of its goals.
**Author Response:** We thank the Reviewer for their thoughtful comments on our manuscript. We agree that there is dissonance in the Introduction and we revised Section 1.1 "The Importance of Erosion of George River" to focus on understanding the drivers of erosion here and less on how erosion rates might be used to help ecological restoration efforts of Georges Bay.

1) Terminology
The first concern I have about terminology is about the fact that the authors keep on calling the $^{10}Be_i$-derived "epsilon" an "erosion rate" (*e.g.* Appendix A and throughout the manuscript), in apparent opposition to $^{10}Be/^9Be$-derived $D_m$ called a "denudation rate" (*e.g.* Appendix A). Both are denudation rates (sum of physical and chemical removal of matter), really, except if a significant fraction of the chemical weathering occurs at depth < 2 m, typically. I understand that the difference between erosion and denudation rates might be very small if the chemical weathering rate is negligible, but this is discussed nowhere in this paper. In addition, in tectonically stable landscapes like the one under study here, and at erosion rates of ~10-20 mm kyr$^{-1}$, it's very possible that chemical weathering is a significant component of total denudation. Anyway, this might be a purely terminological issue, but one that relates to fundamental understanding of the proxies, such that I think this needs to be fixed before the manuscript can even be considered for publication.
**Author Response:** The Reviewer is correct that (1) both $\varepsilon$ and $Dm$ are denudation if there is no loss of mass to chemical weathering and that (2) chemical weathering can be significant in settings such as George River. However, we do not know *a priori* that chemical weathering is negligible and precisely why we compare an erosion metric to a denudation metric in this study.

It is true that we presume there to be negligible chemical weathering in George River, based on the topography, geological setting, and bedrock, and therefore we also consider George River to be a location where the two independent measures of $\varepsilon$ and $Dm$ can be compared. We revised our Introduction to make it clear how we define erosion and how we define denudation in this study. Given that $^{10}Be_i$ is used and has been used to measure erosion since Lal's (1991) seminal paper, in which $\varepsilon$ is defined as the erosion rate, we continue its use here, and because $^{10}Be_i$ is measured from quartz minerals, which are highly resistant to chemical weathering, it may not always reflect total denudation. For this reason, we choose to distinguish $\varepsilon$ in this study from $^{10}Be_m/^{9}Be_{reac}$ based measures of total mass loss (von Blanckenburg et al., 2012; i.e. denudation), and we choose to use $Dm$ as our abbreviation for this measure to differentiate it from $^{10}Be_m$-based measures of erosion, $E$ (Willenbring and von Blanckenburg, 2010)

With regards to chemical weathering, we have, as the reviewer suggested, obtained chemical and physical measurements of river water and flow at a drinking-water intake station for the town of St. Helens near the outlet of George River basin as well as discharge measurements; this site is comparable to TG-9. The chemical and suspended load data extend over 6 years, and the flow/discharge data extend over 60 years. This appears to be the only site sampled in the basin. Considering the measurements of major anions and cations as well as total suspended solids, the chemical export and suspended solids export rates appear similar to the erosion rate determined using $^{10}Be$ measured in the sediment exiting the basin at TG-9. When revising the manuscript, we will include details of these data we obtained, the flow records, and the procedure we used to estimate chemical and suspended sediment export rates for the basin.

Second, it appears surprising to me that the $^{10}Be/^{9}Be$ ratio is here called "$^{10}Be_m/^{9}Be_{reac}$", whereas in most recent studies about this ratio the term "$^{10}Be/^{9}Be_{reac}$" was used. I think I understood why the authors have done so: in this study $^{10}Be_m$ is measured by digestion of the bulk sample (l. 203), rather than on the chemical leachate of the "reac" fraction on which $^{9}Be$ is measured (l. 206-207). This is fine as long as $^{10}Be_i$ - which is also released during bulk digestion - is negligible compared to $^{10}Be_m$, which seems to be the case (Table 2 shows that the latter us typically two orders of magnitude smaller than the former). If I am right, I now feel like it would have been nicer if the authors have explained that themselves in their manuscript, instead of leaving the job to the reader / reviewer. If I am wrong, this terminology is simply very misleading.
**Author Response:** We revised the methods to make it clear that $^{10}Be_i$ is incorporated into the $^{10}Be_m$ measurement, but that the amount of $^{10}Be_i$ is two orders of magnitude smaller than the overall $^{10}Be_m$ measurement, thus being negligible.

Finally - and this is a concern of lesser importance - "$Q$" is an unusual notation for the flux of meteoric $^{10}Be$, at least in the most recent literature. I also note that it is not really defined in the text: l. 211 is the first of this term, with no definition. On a more general note, having the equations just at the end of the manuscript makes the reading and the evaluation very difficult. To come back to this $Q$-notation, this is particularly misleading since in the recent $^{10}Be/^{9}Be$ literature, $Q$ was used to denote discharge, which might be an important parameter for estimating denudation rates for $^{10}Be/^{9}Be$ ratio in cases where Be retentivity onto particles is not complete (see below).
**Author Response:** We had chosen to use $Q$ to denote meteoric $^{10}Be$ delivery to Earth's surface because this was the notation being used in older literature (i.e. Brown et al., 1988); however, we

recognize that Monaghan et al. (1986) used $F$ as the notation for $^{10}Be_m$ delivery to Earth's surface and since this is what other researchers, including authors on this paper, have used in recent work, we changed all instances of $Q$ to $^{10Be}F_{met}$.

2) Interpretation of $^{10}Be_m$- and $^{10}Be/^9Be$-derived rates

The main point made by the authors from their comparison of $^{10}Be_i$- and $^{10}Be_m$-derived rates is the 5-6 times greater rates (leaving aside the fact that one might reflect total denudation and the other erosion rates, which in itself would deserve some discussion in the manuscript) obtained from the latter method (Figure 8). To me, this is simply a grain size effect. Indeed, and as acknowledged by the authors, 10Be_i tends to weakly depend on grain size itself (l. 181-182), $^{10}Be_m$ is strongly grain-size dependent (e.g. Wittmann et al., 2012). Hence any erosion rate inferred from $^0Be_m$ will be affected by grain size effects. I understand from section 3 that both varieties of $^{10}Be$ were measured on the 250-850 µm fraction, which is much smaller than the average grain size of sediment delivered to the alluvial plain of the George River (l. 179-180).

Now, I admit that one could expect much greater $^{10}Be_m$ concentration in the analyzed, relatively fine fraction (meteoric Be being enriched in fine fractions offering large mineral surface areas) than in the "representative" sediment generated in the catchment, thereby likely leading to an understimate of erosion rates by $^{10}Be_m$ (see equation in Appendix A) compared to what could be deduced from $^{10}Be_i$ - even if the latter includes some fraction of chemical weathering. The opposite observation is made in Fig. 8. But in my opinion this apparent contradiction shows even more how these estimates need to be discussed in the frame of the limitation / inherent assumptions of each proxy, an aspect of the discussion that is critically missing from this manuscript. I note that the equation used to calculate $D_m$ in Appendix A accounts for such grain size effects through the term $^9Be_{min}/^9Be_{reac}$ and leads to estimates of *denudation rates* (hence more directly comparable to 10Be_i-based estimates) that, although with some significant scatter, lie uniformly around the 1:1 line (between 1:3 and 2:1) in Fig. 8, rather than showing a systematic overestimate.

**Author Response:** We agree with the Reviewer that the differences between $\varepsilon$ and $E$ are due to grain size bias in our $^{10}Be_m$ measurements. As described to Reviewer 1 above, we now justify our disregard for geological meaning in our calculated values of $E$ and remove interpretation of $E$ from the Discussion altogether. We are not concerned about the grain size effect on $^{10}Be_i$ data, as noted in the manuscript, because $^{10}Be_i$ in low-elevation, temperate settings is rarely affected by grain size bias (van Dongen et al., 2019) and evidence for deep landslides that could otherwise dilute $^{10}Be_i$ in stream sand is not present in the field area.

Another issue with the use of the $^{10}Be/^9Be$ ratio here is the potential bias induced by loss of Be to solution. The equation for $D_m$ in Appendix A here does not account for such poor retentivity. And in contradiction to what the authors say, a pH in the range 4.0-5.5 (l. 119-120) entails significant loss of Be to solution (*e.g.*, Aldahan et al, 1999). This loss can in turn lead to a strong bias in both $E$ and $D_m$ estimates, particularly in situation where the ratio between discharge and erosion rates ($Q/E$, "Q" here being understood as water discharge, see my comment above about terminology) is high. I think this field setting, which is reasonably wet and tectonically quiescent, is one where this ratio is expected to be high. Taking the 10Be_i-derived estimate of erosion rate of ~20 mm kyr$^{-1}$ (Table 3) and the precipitation of ~ 1000 mm kyr$^{-1}$ (Table 2), and

assuming a evapotranspiration factor of 0.5, I get a $Q/E$ ratio of around $10^4$ L/kg, which at pH 5 corresponds to an overestimation of $E$ from $^{10}Be_m$ of ~100% and of $D_m$ from $^{10}Be_m/^9Be_{reac}$ by 10 to 100% depending on the fraction of $^9Be$ in the "min" fraction (see von Blanckenburg et al., 2012). Now, these back-of-the-envelope calculations might well be wrong, and surely can be refined, but clearly this potential issue clearly has to be discussed in more detail in the manuscript before a comparison with another proxy can be made.

**Author Response:** We thank the Reviewer for directing us to the Aldahan et al. (1999) study. This comment and the previous Reviewer's suggestion to refer to $^{10}Be$ partition coefficients in Yiou et al. (1999) led us to try to better understand streamwater pH throughout our field area. To this end, we find that streamwater pH in the main stem of George River at St. Helens and in the Ransom Creek tributary is >5 and has been since measurements began in the 1980s. This supports our interpretation that pH conditions in George River basin are not likely to induce $^{10}Be_m$ loss to solution.

*** Other comments ***
- l. 43: "northern tropics" is a bit misleading (to me, the "northern tropic" is the Tropic of Cancer, which is not what the authors are talking about here, I guess).
**Author Response:** We mean the northern reaches of Australia's Great Dividing Range, which is north of the Tropic of Capricorn. We edited the text to better reflect this geography and not mislead the reader.

- l. 319: "to not have not had" -> problem with this sentence.
**Author Response:** Revised to fix the wording problem.

- l. 376-383: Not sure this paragraph (about controls on long-term rates) belongs to this section (which is supposed to be about trunk stream vs. tributary sediment supply.
**Author Response:** The first paragraph is a discussion of where sediment that is passing through trunk channel sites originates, from the tributaries, from below the tributaries, or from the entire catchment? The first paragraph describes our interpretation that sediment at trunk channel sites originates in tributaries, and it describes how we derive a value for $\varepsilon$ for the whole George River basin. The second paragraph describes how this whole-catchment value for $\varepsilon$ compares to whole-catchment values of $\varepsilon$ across the rest of the Great Dividing Range.

- Table 2: Strictly speaking, "$^{10}Be_{met}$" is not defined anywhere - I think the authors mean "$^{10}Be_m$", which is used throughout.
**Author Response:** Fixed.

- The end of Figure caption 2 is missing.
**Author Response:** Text-box issue. This is now fixed.

- Fig. 5B and pie chartes in Fig. 7: I must admit I did not understand exactly what purpose the calculation and presentation of this "hillslope erosivity" serve in the manuscript.
**Author Response:** We mention and use Kidd et al.'s (2014, 2015) "hillslope erosivity" metric because we did not want to neglect their work and the maps of soil erosivity for Tasmania they produced. Despite being based on a calculated value via a multivariate equation, Kidd et al.'s

"erosivity" designations (i.e. Extreme → Very Low) are qualitative in nature, which makes it difficult to summarize "erosivity" at a catchment scale or to compare our numerical catchment-wide erosion rates to Kidd et al.'s "erosivity." By determining that "erosivity" is ultimately significantly related to slope (Fig. 5B), we are able to acknowledge Kidd et al.'s work and use hillslope angle as a proxy for erosivity. We've reworded this last paragraph of the Methods to explain this better.

---

## Author Response (AR1)

09 January 2022

Dear Dr. Wittmann-Oelze,

On behalf of my co-authors I am pleased to inform you of our study on the use of $^{10}$Be to measure landscape change in Tasmania, now resubmitted with major revisions and now titled, "Comparison of basin-scale in situ and meteoric $^{10}$Be erosion and denudation rates in felsic lithologies across an elevation gradient at the George River, northeast Tasmania, Australia."

We appreciate the time extension allowed to us as it has given us time to enjoy the holiday seasons with our families and finish off academic terms, and it also provided us with the time to fully consider all of the thoughtful, constructive, and supportive critiques made by yourself and the two reviewers on our original manuscript. As you noted in your remarks, you were only able to see our Responses to the two Reviews but not the revised manuscript before you could give your decision; in light of your comments in addition to the Reviewers' comments, even more significant changes have been made to our manuscript, which we believe fully address all relevant concerns. To this end, we not only respond to your comments, and those made by Handling Editor, Greg Balco, in the pages which follow this cover letter. We also attach our revised Responses to Reviewers' comments.

As requested, we have revised and made significant major revisions to our manuscript, most notably:

1. The Introduction section has been streamlined to focus on the newness of the data presented in this study. Whereas we previously focused on coastal ecosystems, we now focus on the fact that we present the first erosion rates from the island of Tasmania and the southernmost erosion rates for Australia's eastern passive margin and Great Dividing Range.

2. We remove all mention of meteoric $^{10}$Be-based erosion rates because they are subject to grain-size bias and all reviewers called this out. The new manuscript only includes $^{10}$Be$_i$ erosion rates and $^{10}$Be$_m$/$^9$Be$_{reac}$ denudation rates, both given in units of Mg km$^{-2}$ y$^{-1}$. We also significantly revised our use of the terms "erosion" and "denudation" for this study. We were able to acquire water quality data and total suspended solids for a water intake station near the town of St. Helens, which we use to better interpret erosion and denudation rates at site TG-9. We use these data to calculate dissolved load and suspended sediment loads for the TG-9 site.

3. We change the focus of our interpretation away from mean annual precipitation and instead focus on the relationship between erosion rates and mean basin elevation. Previously, our precipitation data came from the WorldClim dataset, which reported a narrower range of precipitation across the field area than what was measured from rainfall gauging stations *and* was self-correlated to elevation in the way rainfall was interpolated. By focusing on the erosion-elevation relationship, we demonstrate that

measured values of rainfall and temperature in the field area are also correlated with elevation. Thus, the crux of our new interpretation is that erosion is primarily related to elevation.

Overall, we agree – as was noted by Reviewers – that our original manuscript suffered from overcomplicating the narrative of our study by trying to include too many data that were, in the end, unnecessary for the simplest and we believe clearest interpretation of our data. The major revisions we made to our manuscript, in response to Reviewers' and Editors' comments, have made our study more focused, easier to understand, less reliant on interpolated/modelled/abstract data, and thus clearer. We recognize that there are imperfect aspects to this study, but we discuss these limitations to the best of our ability and believe that our efforts will continue to be of use to geomorphologists using either in situ or meteoric $^{10}$Be in their own studies and to those interested in rates of landscape change over time.

Kind regards,

Dr. Eric Portenga (corresponding author)

Leah A. VanLandingham
Edward C. Lefroy
Amanda H. Schmidt
Paul R. Bierman
Alan J. Hidy

(Author responses to Editors and additional responses to Reviewers are attached below)

Comments from Associate Editor and Principal Editor on MS geochron-2021-23

Dear Eric et al.,

As you know, I have received two reviews for your MS and I´ve carefully checked your responses. Unfortunately, due to the system setup of the journal, I do not get the track-changed version of your replies at this stage, only after I have made my decision. For the record- this is nothing I can change, but it´s simply due to the system, but it makes, in my view, my life as an Associate Editor harder. The reason is that I cannot in detail see your changes, and some of them are hard to judge without the connection to the text.

One of these issues where seeing the track-changed version would clearly help me is your treatment of terminology, an issue that both reviewers commented about. I agree that back in Devendra´s time, the term erosion rate was used for in situ-derived rates. But I think that the community has clearly advanced on this in the meantime. I don´t mean that one should per se call in situ-derived rates "denudation rates", because weathering rates might be very very low, but without seeing the numbers and changes, this is difficult for me to judge.

**Author Response:** We recognize that some researchers are now referring to in situ $^{10}$Be erosion rates as denudation rates by assuming that there is no chemical mass loss beneath the depth of neutron penetration, but the majority of new work presenting $^{10}$Be$_i$ data still use the term erosion rates and many authors never test the assumption that there is no chemical loss of mass. During our revision time, we were able to acquire water quality data from a water intake station at the town of St. Helens (located near our TG-9 sample site). Using these data, we determine a dissolved load and a suspended load for the whole of the George River; unfortunately, we do not have data anywhere else in the catchment. Nevertheless, we use these data to help us make a more-informed interpretation of our in situ and meteoric $^{10}$Be data at TG-9 and we include a new section of our Discussion that explores what all of the new data mean with regard to the erosion/denudation issue. We conclude that there is likely a component of chemical mass loss in addition to the $^{10}$Be$_i$ erosion rates we present; in other words, the $^{10}$Be$_i$ data are not the full story of mass loss for the George River.

We believe this topic deserves a much fuller discussion: What do $^{10}$Be$_i$ data actually measure, erosion or denudation? When or where is each term more appropriate to use? What data are required in order to use one term versus the other? What are the limitations of this system? What does the $^{10}$Be$_m$/$^9$Be$_{reac}$ data system mean? How does it compare to $^{10}$Be$_i$? We do not think, however, that this study of the geomorphology of a small river basin in Australia is the place for this discussion. Thus, while we acknowledge the inconsistent use of the two terms in the past and discuss the difficulty in knowing what the two varieties of $^{10}$Be measure, we still refer to $^{10}$Be$_i$ data as erosion and $^{10}$Be$_m$ data as denudation, though both are given in similar units (Mg km$^{-2}$ y$^{-1}$) for ease of comparison.

I however have a hard time believing that we are dealing with a landscape where chemical weathering is totally absent. (I read from your response that there is some water chemistry data available, but even a multi-decadal record in TDS still has time scale issues when comparing to millennial-scale cosmogenic rates). So, even if it´s only a super small portion of total denudation,

the in situ-derived rates should be called denudation rates and are thus directly comparable with 10Be/9Be-derived denudation rates (btw, I´ve calculated the reac/min fraction (eq. 9 in my 2015 paper) using your data, and the values are around 0.25-0.3, which is a range in Be-related weathering degrees found by others, i.e. indicating 25-30% of a weathering degree, meaning that weathering cannot be absent). If it wasn´t for this comparison, I would not care too much, but I stress here that it would make the MS much easier to digest if the terminology was adjusted.

**Author Response:** As noted above we were able to find chemical weathering data for one site and we use that to guide our interpretation of what the $^{10}Be_i$ and $^{10}Be_m$ data mean, at least for TG-9, the lowermost sample in our field study. We agree that there is a chemical weathering component missing from $^{10}Be_i$ data, at least for this site and we fully acknowledge this in our Discussion. It is very interesting that you calculate a 25-35% degree of chemical weathering from our data because we note in our new Discussion that the $^{10}Be_i$ erosion rate at TG-9 might underestimate meteoric denudation by ~40%. Using Equation 9 in Wittmann et al. (2015), we calculate a 40% weathering degree at TG-9. This is now noted in the revised manuscript; however, we refrain from running similar measures at other sites since we do not have independent chemical weathering data to cross-check the $^9Be_{reac}/^9Be_{min}$ fraction at those sites.

Also on this matter, I would suggest that you present either in mm/kyr or in t/km2*yr, at least in the text (I would not opt for "Mg/km2*yr", for reasons of common acceptance.

**Author Response:** Since readers across the world will be interested in this study, we hesitate to give mass loss in "t". We know this to be metric tons, but it could be easily confused with imperial tons. We prefer to use Megagrams (Mg) since this is standard SI notation. However, we now present both $\varepsilon$ and $Dm$ in the same units of measurement, Mg km$^{-2}$ y$^{-1}$.

Also, it´s in my view totally unnecessary to provide Dinsitu AND in situ-derived sediment fluxes in the Figures, as these essential only differ by density). Tables could provide both values. Note that for Dinsitu, one also needs a density assumption, so one could easily convert these to t/km2*yr, or, if lithology is uniform, as claimed, one could use the same density to convert 10Be/9Be-derived denudation rates from t/km2*yr to mm/kyr. Given that you only have one major lithology, this would be a fair assumption.

**Author Response:** We agree completely. In our revised manuscript, we only refer to in situ erosion rates in units of Mg km$^{-2}$ y$^{-1}$, which bypasses requiring density to convert to L/T. We make this calculation only once in the Discussion in order to graph our new data alongside erosion rates from Codilean et al. (2021) across the rest of the Australian east coast.

Regarding erosion rates from single $^{10}Be$ concentrations, I agree with both reviewers, and welcome the removal of their interpretation from the Discussions, as their presentation did not help to clarify controls among the variables. I hope that Fig. 6 and 7 will benefit from this removal and adjustment of terminology, as these are particularly hard to digest.

**Author Response:** We wholeheartedly agree that former Figures 6 and 7 were difficult to digest or understand their meaning. To this end, we've opted for a significant reduction in illustrative data reporting and we focus our results simply on the following three figures:

1. The significant correlation between $\varepsilon$ and mean basin elevation ($R^2 = 0.91$, $p < 0.01$)
2. The comparison of $\varepsilon$ and $Dm$ against a 1:1 line
3. The moderate relationship between $Dm$ and the percent of each tributary basin that is characterized as "High" to "Extreme" erosivity.

The remainder of regression analyses are presented by $R^2$ and p values in the main body of the test. This decision has helped us streamline our results and focus our figures on the relationships we believe to be those that really matter in our interpretation of our results.

Regarding the comparison between denudation/erosion rates, I think it would help to add the "Appendix" equations into the main text, at least the two main ones (Lal, von Blanckenburg). I´m not sure where the "Appendix" equations will end up, hence if they are not instantaneously visible for the reader, this would help an audience that is not so familiar with the topic (assuming that the general audience of GChron is not familiar with this).

**Author Response:** We agree. Now that we only include $\varepsilon$ and $Dm$ in our revised manuscript, we were able to reduce the content of the former Appendix table to focus on the Lal and von Blanckenburg equations, only. We keep the equations in a table since there are so many shared variables, but that table is now part of the main body of the text.

I´m happy to see that you take the effort and compare in situ and 10Be/9Be denudation rates. I am a bit puzzled however by the overall scientific outcome of this comparison. Dinsitu show a clear trend with precipitation (but please see Greg Balco´s comment on this at the end here, in support of the concerns raised by Reviewer #1.), whereas 10Be/9Be denudation rates are all over the place if one takes the entire dataset. Why is that so?

**Author Response:** These are really good questions. We would go as far to say that the scientific meaning of this comparison (and similar comparisons in the literature) is intricately tied to the discussion of what is erosion versus what is denudation, and whether $^{10}Be_i$ and $^{10}Be_m/^{9}Be_{reac}$ metrics measure the same processes, which they are purported to do in the literature under certain circumstances. As noted above, we do not believe this paper is the correct place to provide such answers to these philosophical debates, which is why our revised manuscript is much less declarative than before. It is worth noting as well, that in our revisions, and our decision to rely on measured rainfall data, rather than interpolated rainfall from the WorldClim datasets (thus addressing GB's and R1's concerns), we recalculated $^{10Be}F_{met}$, which led to changes in resultant values of $Dm$. In doing so, we find even less consistency between the two datasets, with none of the sample sites falling on the 1:1 line; however, we note that $Dm$ values are all within a factor of 2 of $\varepsilon$. As noted elsewhere, these new $Dm$ values are, at face value, related to erosivity and are highest in catchments with known intensive mining and forestry histories. Nevertheless, we maintain that our results from both datasets supports the traditional use of $^{10}Be_i$ and the new use of $^{10}Be_m/^{9}Be_{reac}$ to quantify landscape dynamics in other field areas.

Regarding the methodology, please clarify in a response if the used rainfall derived F10Bemet could play a role in obscuring a trend between denudation rates and precipitation. I understand that you favor Graly et al. (2011) values, but that makes me a bit worried if there could be some intrinsic dependence. Btw, are you using a local rainfall-relationship or a global one? I don´t think that is specified in the paper.

**Author Response:** This is now clarified. GB's and R1's concerns about the elevation-based interpolation of rainfall in the WorldClim model made us reconsider our use of that global precipitation dataset, and we ultimately omit its use in this study altogether in favor of the strong correlation between locally-measured rainfall and elevation. Similarly, we use a locally-measured temperature dataset to show a similar strong correlation between mean annual temperature and elevation. In all, our revised interpretations are focused on the strong correlation between $\varepsilon$ and mean basin elevation. As we mention in the manuscript, we prefer the Graly model because it allows us to derive a unique value for $^{10Be}F_{met}$ for individual basins whereas the GCMs do not allow the same resolution. Furthermore, we now clarify in the paper that the values we end up using from the Graly model are consistent with those from any other model we could have used. All of this is now specified in the paper.

Further, you use the 2.5 ppm value suggested by the von Blanckenburg paper (2012) for 9Beparent. Unfortunately, we have seen that this is probably only valid for larger basins, hence I would really welcome some measured values, even if it´s one sample per lithological unit….The basalts will clearly deviate from that value, but also granites can show some heterogeneity (see Dannhaus paper) and in such small basins (your tributaries), this might really be an issue. In order to check out the potential magnitudes, the percent areal coverage of each lithological unit per basin might give a clue? At least discuss please in which direction the associated changes would go for the 10Be/9Bederived denudation rates, if you either over- or underestimate 9Beparent (but, in order to resolve the missing trend, there must be some systematic pattern). Another way to assess the accuracy of 10Be/9Bederived denudation rates would be to do the Q/E calculation suggested by reviewer #2 (note that even at "high" pH values of 5, the bias might be between 10-100% on D´s). At least that would give you an idea where the problem might be. Lastly, there might the geological reasons for the absence of this trend with precip (if it´s at all there?), which might be e.g. overland flow? However, I´m overall not too worried, since the overall numbers are not so far apart between in situ and 10Be/9Be-derived denudation, but still a bit puzzled.

**Author Response:** We agree that using a global bedrock average value for $^9Be_{parent}$ of our field site is not ideal; however, we did not sample bedrock at the time samples for this study were collected and processed. It is one of the limitations of this study and we make this clearer in our revised Methods section.

In conclusion, I think this paper needs another round of revisions. Given that one reviewer opted for "major", the other for "reject", I will send it out again. I´m looking forward to a new version.

All the best,
Hella

Below I give some more technical / structural advise that could improve your paper:

Line 101: Just to be accurate here: At least the studies I was involved in, we use the "10Bereac/9Bereac" ratio. We do not perform bulk extraction of meteoric 10Be.

**Author Response:** We make note of this in our revised manuscript.

Line 187: "No native Be" was detected. It´s hard to believe that there was "no" native Be found, because this is a matter of which instrument you use. We measure 9Be in seawater ranging down to 1 ppt and lower using ICP-MS. Please re-phrase such that "relative to added 9Be carrier amounts, no significant Be was found" or something like that.

**Author Response:** This wording is changed as suggested.

Line 240: This definition of sediment fluxes derived from Dinsitu should come in the paragraph above, where these are mentioned.

**Author Response:** We no longer refer to sediment fluxes in our revised manuscript.

Line 242: How were the "average modelled" rates derived? Not clear to me. If these are from a simple mass balance. i.e. summing up the respective yields and then dividing by summed area, then please say so.

**Author Response:** This was an area of confusion that persisted through many revisions of the manuscript. We significantly revised this section of the Results for clarity and present a clearer and more-streamlined assessment of mass dynamics between tributaries and trunk channel sites.

Fig. 4 The "historically burnt areas" are really hard to make out in the Figure. Would suggest that you use some other color / make larger.

**Author Response:** Done

In addition, editor Greg Balco had the following comments:

Reviewer 1 points out that there is not an obvious physical means by which the relatively small variability in precipitation could cause a relatively large effect on the erosion rate. This is true, and there are some additional aspects of this part of the paper that seem to need attention. First, the smoothed data set used to calculate precipitation for all the basins (Worldclim) appears to have been generated by using elevation and distance from the coast as smoothing parameters, so precipitation is not an independent variable, it is already defined to be correlated with elevation. Thus, for example, the R^2 of 0.93 in Fig. 3C appears to just be reproducing how the smooth data set was created. In other words, precipitation and elevation appear to be effectively the same independent variabie in these regressions. Second, their Figure 3 shows that the smoothed data set is doing an extremely poor job of reproducing rain gauge data -- the true variability from rain gauges is twice as big as predicted by the smooth data, so the question brought up by the reviewer about the range of precipitation is already based on an incorrect range. Overall, the aspect of the paper dealing with whether or not precipitation is the main control on erosion needs rethinking. Clearly a precipitation control is possible, but I am not sure how you would exclude the possibility that elevation is actually the controlling variable, either through, for example, a relationship between elevation and local relief or perhaps between elevation and temperature, perhaps involving freeze-thaw processes. A possible approach to this would be to take Reviewer 1's observation that there is a physical limit on how much variation in the erosion rate could be accommodated by the available range in precipitation, and evaluate whether or not that is adequate to explain the observed variation, or if an additional factor related to elevation would be required. In any case, this aspect of the discussion needs more careful attention.

**Author Response:** We agree and in response have omitted the use of the WorldClim model in our study altogether, instead opting to focus our interpretation on the $\varepsilon$ relationship with mean basin elevation as being primary. We show significant correlations between rainfall and temperature data measured in the field area with elevation (new Figure 3), and discuss that elevation-dependent climate related processes are the processes that drive rock weathering and erosion in the George River, but that ultimately it is elevation-controlled climate processes that are the controlling variable.

Reviewer #1

General comments
It is my pleasure to review the manuscript by VanLandingham titled "Comparison of basin-scale in situ and meteoric $^{10}$Be erosion and denudation rates across a rainfall, slope, and elevation gradient at George River, northeast Tasmania, Australia". Quantification of millennial-scale background erosion rate is crucial to understand landscape evolution over time and to assess human-induced land degradation. Although the driver of long-term erosion is commonly attributed to tectonic uplift/topographic relief over large scales, such pattern may be less clear on local scale due to small variability in these factors and distinct variability in other factors. Here the authors studied the background erosion rates in the George River on the island-state of Tasmania. The major goals of this study include two parts: 1) to find the controlling factors of millennial-scale denudation rates in the study region and 2) to compare between denudation rates derived from a well-established method (in situ 10Be) and a relatively new method (meteoric 10Be/9Be). The first part is a piece of standard work and the highlight should be the second part.

In brief, the authors found that in situ 10Be-based erosion rates are positively correlated with precipitation (R2=0.82) and only poorly correlated with slope (R2=0.17), which is (surprisingly) different from the pattern derived from dataset of mainland Australia (slope control of erosion, Fig. 9b). The authors also showed that the denudation rates based on 10Be_i and 10Be_m/9Be_reac agree within a factor of 2 (except TG-7, Fig. 8), supporting the meteoric 10Be/9Be applications in basins with minor geological heterogeneity and little human-induced disturbance.

In general, I think it is a nice case study regarding inter-method comparison (in situ 10Be vs. meteoric 10Be) and this study on determining rates of catchment-scale denudation processes meets the scope of GChron. Nevertheless, the interpretation on precipitation control requires more lines of evidence especially quantitative constraints and more details need to be added regarding calculation of 10Be_m/9Be_reac-based denudation rates. I provide specific comments and technical corrections below and hope that these comments can help to improve the manuscript.

**Author Response:** We thank the Reviewer for their thoughtful considerations on our manuscript and address them individually below.

Specific comments
1. Control of precipitation vs. other factors. The correlation between 10Be_i-based denudation rates and precipitation rates looks sound. However, a key quantitative link is missing here. Please note that the variability of precipitation among all sampling basins is quite small as 1.4-fold (0.97-1.26 m/yr), compared to ~5-fold variability in denudation rates (4.8-24.5 m/kyr).

**Initial Author Response:** We now acknowledge this in the first paragraph of the Discussion.

**Additional Author Responses:** Upon Major Revisions, this acknowledgement remains in the Discussion, though no longer in the first paragraph. We also note that we now only make a correlation between $^{10}$Be$_i$ erosion and mean basin elevation because the rainfall datasets we were

using (WorldClim) are interpolated globally and self-correlated to elevation. Comparing $^{10}Be_i$ erosion rates only to elevation ensures that we are not drawing conclusions based on a relationship with rainfall, the measure of which is self-correlated with elevation. We still discuss what it means that erosion in our field area is correlated with elevation and that rainfall and a new inclusion and discussion of temperature mean for the mechanical breakdown and erosion of rock in our study.

Erosion rates (E) are commonly assumed to scale with precipitation rates (P) as E ∝ P^m (D'Arcy and Whittaker, 2014), and m is commonly assumed to be 0.5 (using m/n of 0.5 and n of 1) or may be a bit higher as e.g. 1.2 (using m/n of 0.5 and n of 2.4) based on global data fitting (Harel et al., 2016). However, in both cases the large variability in erosion rates cannot be explained by the small variability in precipitation rates. If such scaling is applicable in this study area (if not, please justify), it means that the majority of erosion rate variability should be explained by factors other than precipitation. The authors provided several other alternative explanations (around Lines 338-345), which I appreciated, but then they rejected these scenarios later. From my perspective, it seems that the denudation rates are controlled by certain processes related to elevation (6-fold variability, Fig. 6). Although glacial processes (elevation-related) may not play a major role in such low-elevation regions as the authors mentioned, what about other processes? For example, discharge variability may also play a role in river incision (Lague, 2014). I think the WorldClim global dataset includes similar parameters (precipitation seasonality?) that can be extracted for analysis. In short, the authors should provide more alternative scenarios to explain the variability of denudation rates, which may not be mainly controlled by the small precipitation gradient.

**Initial Author Response:** D'Arcy and Whittaker (2014) demonstrate that the normalized channel steepness index – often used as a proxy for erosion in tectonically-active regions – is linked to precipitation in the sense that uplift in the Andes creates a topographic barrier, which is reduced via precipitation-driven stream incision. The relationships that D'Arcy and Whittaker model may be appropriate for other tectonically-active, high-elevation regions; however, this is not the setting for George River in Australia. As such, we hesitate to fit our observations into a landscape evolution model derived for a significantly different topography. As the Reviewer suggests, we argue that erosion rates are controlled by some factor related to elevation, and as noted, we review the likely possibilities (e.g. periglacial weathering, mass movements, etc.) and ultimately use our observed relationships between longitude, rainfall, and elevation and erosion rates to suggest that in this specific region of the world, erosion rates are primarily driven by precipitation. The coastal setting of George River and Tasmania's temperate climate are not highly seasonal, and Tasmania lies outside the track of tropical cyclones which seasonally make landfall in far northern Australia.

**Additional Author Responses:** We stand by our initial response to this comment that it would be inappropriate to apply erosion-precipitation scaling from the Andes to the George River. We just now reiterate that we no longer rely on precipitation, or any climate variable, in our study that comes from gridded, interpolated global datasets as these data are self-correlated to elevation. We now only rely on rainfall and temperature data that has been measured at our field area and their relationship to elevation.

2. Terminology (epsilon, E and D_m; erosion vs. denudation). It is quite confusing to the audience (or at least to me) when reading rate estimates of different meanings, from different calculation methods, with different units (mm/kyr vs. Mg/km2/yr) and different from the terminology used by previous studies. First, I think both 10Be_i and 10Be_m/9Be_reac methods derive denudation rates, i.e. removal of whole rock by physical erosion and chemical weathering. Second, I think the unit should be unified in the text (either mm/kyr or t/km2/yr) for reading purpose. Third and more importantly, I do not think 10Be_m based E is needed in the discussion. Meteoric 10Be concentration alone is very sensitive to grain-size effect (Singleton et al., 2016; Wittmann et al., 2012). When the analyzed sample is dominated by coarse materials (250–850 microns of bedload rather than suspended load), meteoric 10Be concentrations will be low as expected (less adsorption capacity and/or quartz dilution) and thus the calculated erosion rate will be biased towards higher values as shown here (e.g. Fig. 8). Hence, I would suggest to simply remove all the content related to 10Be_m based E (also in figures) as it has not been discussed in detail anyway and does not contribute to the key conclusions. If the authors insist, including such estimates in the supplement would be more than enough.

**Initial Author Response:** First, in this study, as with past studies, we find it very important to maintain a distinction between erosion and denudation and which is measured by each $^{10}Be_i$ or $^{10}Be_m$. Here, and in our previous work, we acknowledge that $^{10}Be_i$, which is measured on weathering resistant quartz grains, is an apt measure of erosion (the total physical mass loss from a landscape). Given that some mass in any landscape may be lost to chemical weathering a different method is needed, and this is where the von Blanckenburg et al. (2012) measure of total mass loss – i.e. denudation – is derived.

Presenting erosion rates in units of length/time (i.e. mm/kyr) is consistent with nearly the entire literature of $^{10}Be_i$ studies; if there is no mass loss to chemical weathering, the product of erosion rate and the density of rock should yield an adequate measure of denudation as well mass/area/time (i.e. $Mg/km^2/yr$), and should be equal to the result of methods that directly measure denudation. Similarly, presenting denudation rates in units of mass/area/time (i.e. $Mg/km^2/yr$) is consistent with the units of denudation presented in von Blanckenburg et al.'s equation derivations. In our previous work (Portenga et al., 2019, GSA Bulletin), we ask the question, "When should erosion and denudation be the same?" and the field area of this study at George River is ideal to answer this question: homogenous lithology, relatively thin soils, and pH conditions that do not promote desorption of $^{10}Be_m$ from sediment grain coatings. We do, following revisions and Reviewers' suggestions, make it more clear that pH conditions for George River, in soils and in stream water, are such that desorption of $^{10}Be_m$ is unlikely. Understandably, many previous studies measuring landscape dynamics with $^{10}Be_i$ use the term "denudation" because the assumption of little mass loss to chemical weathering is made, or the amount of chemical weathering is measured independently.

Lastly, given that both reviewers take issue with our inclusion of the geomorphological meaning of *E* presented in this study, we now highlight in the Discussion that our calculated values of *E* are likely due to unresolved grain size bias (citing suggested Wittmann and Singleton papers), and therefore are inaccurate measures of landscape dynamics at George River. We choose to retain calculated values of *E* in our data tables and in our Figures to visually highlight this inaccuracy. We focus all remaining discussion on the comparison of $D_m$ and $\varepsilon$.

**Additional Author Responses:** We, the authors of this study, had some lengthy discussion during revisions about the terms "erosion" and "denudation" and what, really, the $^{10}Be_i$ or $^{10}Be_m/^9Be_{reac}$ metrics of landscape change measure and under what conditions. The reality is that the terms "erosion" and "denudation" are sometimes used interchangeably and inexactly in the literature, despite many efforts to codify these terms. We also recognize that there are some physical conditions under which $^{10}Be_i$ data do measure physical and chemical loss of mass from landscapes (often referred to as denudation) and some conditions under which $^{10}Be_i$ data miss out on chemical mass loss; we also recognize that the $^{10}Be_m/^9Be_{reac}$ data have always been cast as denudation rates and maintain that use. Ultimately, we believe that the "erosion/denudation" discussion, and determining what actually $^{10}Be_i$ or $^{10}Be_m/^9Be_{reac}$ measure, is something we as a geomorphology community need to discuss elsewhere — that this manuscript is not the place for that discussion. To this end, we soften our declaration of what is and is not erosion or denudation and simply state early on in our revised manuscript that $^{10}Be_i$ data have traditionally been referred to as erosion rates, that $^{10}Be_m/^9Be_{reac}$ data have traditionally been referred to as denudation rates, and that continue to use these terms for the purposes of this manuscript.

We do now refer to all $^{10}Be_i$ erosion rates in this study in units of Mg km$^{-2}$ y$^{-1}$ to be consistent with Lal's (1991) original unit nomenclature for $^{10}Be_i$-based erosion rates, and to be able to compare our $^{10}Be_i$ and $^{10}Be_m/^9Be_{reac}$ datasets as well as chemical and suspended sediment fluxes.

As suggested by both Reviewers, we completely remove all $^{10}Be_m$-based erosion rates from this study as the $^{10}Be_m$ data were not normalized to $^9Be_{reac}$, and therefore likely suffering from grain-size bias, which we are unable to account for. We now only include $^{10}Be_i$-based erosion rates ($\varepsilon$) and $^{10}Be_m/^9Be_{reac}$-based denudation rates ($D_m$).

3. Choice of $^{10}Be$ depositional flux. First, it is better not to use Q as 10Be_m delivery rate. I think it may cause confusion as it means water discharge for many geomorphologists, and it is inconsistent with the original framework (von Blanckenburg et al., 2012) (cited by the authors) or the co-authors' previous paper (Portenga et al., 2019). Second, it is appreciated that the authors mentioned several different approaches to determine10Be_m delivery rate. However, the authors then decided to only use Graly et al. (2011)'s approach. Graly et al. (2011)'s equation is based on fitting of modern precipitation 10Be dataset and might cause flux overestimation in some cases when applied to millennial timescale (Deng et al., 2020). Since there is no 10Be_m delivery rate measured in the studied basin (e.g. using dated soil profiles as Reusser et al. (2010)), I would recommend to also calculate denudation rates using 10Be delivery rates from GCM that indeed integrate over millennial timescale (Heikkilä and von Blanckenburg, 2015). This approach can also provide latitude- and longitude- specific 10Be fluxes. As such, readers can get more comprehensive information on the utility of both methods in this specific region by comparing resulting 10Be_m/9Be_reac based denudation rates with those from 10Be_i. If the authors still decide to only use 10Be delivery rates from Graly et al. (2011)'s equation, the denudation rate results using GCM-based 10Be delivery rates should at least be included in the supplement.

**Initial Author Response:** We thank the Reviewer for suggesting that we consider using values of $^{10}$Be_m delivery from other studies and deriving denudation rates from those values. First, we now refer to Q as $^{10Be}F_{met}$ to be consistent with recent works. Second, we've revised our methods to acknowledge the many different ways that $^{10Be}F_{met}$ can be measured or modelled. The CGMs that the Reviewer directed us to do not have the spatial resolution that is required to differentiate values of $^{10Be}F_{met}$ for individual study basins. We prefer to use Graly et al.'s (2011) $^{10Be}F_{met}$ estimations because it allows us to present a specific value of $^{10Be}F_{met}$ for each studied basin. We revised our Methods section to better present the possible values of $^{10Be}F_{met}$ from each model, and we show that our estimated values of $^{10Be}F_{met}$ based on Graly's work is similar to $^{10Be}F_{met}$ predicted from GCMs. While it would be an interesting exercise to measure Dm from each $^{10Be}F_{met}$ value, we think this would distract from the narrative of this study; because of the similarity of all $^{10Be}F_{met}$ values, which we already present, the spatial patterns and relationships to basin metrics shown in Figure 6 and Figure 7 would not change. We therefore do not believe it is necessary to calculate denudation rates based on each of the GCMs to interpret the spatial variability of meteoric $^{10}$Be-based measurements of erosion and denudation.

**Additional Author Responses:** In our major revisions, we no longer rely on precipitation data from the WorldClim model due to its self-correlation with elevation. Rather, we recalculated mean annual precipitation rates for each sample basin from the observed correlation between measured rainfall and elevation, derived from Australian Bureau of Meteorology gauging stations. In doing so, the calculated mean annual precipitation rates, which we use to calculate values of $^{10Be}F_{met}$ changed, as did the resulting measures of $^{10}$Be$_m$/$^{9}$Be$_{reac}$ denudation rates. The meaning of these newly-calculated rates is now fully discussed in the revised manuscript.

4. Long-term trend in denudation rates (millennial-scale vs. decadal-scale). The authors mentioned in several places that the sediment input increased due to land use prior to 1990s and later decreased (?) afterwards, and the sediment input nowadays should be generally higher than millennial-scale denudation rates. At least this is my impression after reading the text. So I am wondering if there is any gauging data (e.g. sediment yield) in the studied catchments so that comparison between rate estimates that integrate over different timescales can be made and thus support the authors' claim. Although I am not sure about data availability, such comparison seems to be important as the authors emphasized this point as a major implication at the end of the abstract.

**Initial Author Response:** The Reviewer is correct that historical sediment yield increased and has since decreased back to pre-disturbance conditions (Knighton, 1991), but it is not accurate to say that sediment input nowadays is higher than millennial-scale conditions. Most of the excess historical sediment has either been sequestered in floodplains or delivered already to Georges Bay, which we discuss and cite appropriate literature. Unfortunately, sediment gauging data are scant and generalized when referred to in the literature at all, and to our knowledge there are no long-term records of sediment yields for George River or any of its tributaries. We are limited to brief mentions of sediment yields in Knighton (1991) and recorded variations of sediment supply shown in Figure 1, therein. In order for Knighton (1991) to quantify variations in sediment supply, sediment yields must have been recorded at some time; however, we have not been able to ascertain where such data exist. We have revised our text to note that our knowledge of predisturbance channel conditions, or grain size measurements (i.e. 30-50 mm) come from a limited number of sites in the field area.

**Additional Author Responses:** In our revisions, we were able to acquire water quality data from TasWater – the organization that manages drinking water for the region – from a water intake station at the town of St. Helens. This water quality data included total suspended sediments, which we found to be small in magnitude. This water intake station, however, did not measure bedload and we are still unable to extrapolate much bedload information from this station to the rest of the catchment. However, we do use the water quality data to derive the chemical load of the George River at this site, which we use in our Discussion of our understanding of $\varepsilon$ and $D_m$ at sample site TG-9.

5. Low D_m caused by topsoil erosion. In Lines 409-432, the authors argued that the low D_m in the headwaters are caused by significant 10Be_m-rich topsoil erosion. I do not necessarily disagree on this argument. However, I am confused why such process does not affect 10Be_i data. Both in situ and meteoric 10Be should show a decline profile with soil depth and thus are enriched at the surface, and if bioturbation plays a role and a mixing layer is established, it should also affect both nuclides.

**Author Response:** Meteoric [10]Be profiles in soil have been shown to re-establish over short-term periods following bioturbation events whereas in situ [10]Be in bioturbated soils remains homogenous within the soil mixing zone (Jungers et al., 2009), which would explain why recent and intense disturbance to topsoil affects only meteoric [10]Be and not in situ [10]Be. We make this clearer in the manuscript.

Technical corrections and minor scientific comments
Main text
Title: I am not sure if a range of precipitation rate of 0.97-1.26 m/yr can be considered as a gradient. The variability is relatively small compared to that in the eastern Australia coastal rivers (Fig. 9). How about "… denudation rates in felsic lithologies at George River…"? The studied catchment is indeed dominated by Devonian felsic intrusions and the authors emphasized in the text that the simple lithology in this catchment makes the inter-method comparison easier. I will leave the decision to the authors.

**Initial Author Response:** We appreciate the reviewer's suggestion about focusing on the lithology, and we agree that this is important for the inter-method and comparison. We recognize that the rainfall gradient observed across George River is not significant when compared to other regions, which we already note in the text. However, of all studies that compare [10]Be[i] erosion rates to basin metrics, including mean annual precipitation, there are few correlations as close as that which we observe for George River. Instead, we think it is more impressive that such a small rainfall gradient seems to influence erosion in this low-elevation, post-rift margin. Nevertheless, we revised the title of this study to remain inclusive of precipitation, but also add the inclusion of felsic lithologies as the reviewer suggests.

**Additional Author Responses:** The Reviewers and Editors all encouraged us to reconsider the overall importance of the relationship we identified between [10]Be[i] erosion rates and rainfall. In

doing so, we chose to reconsider our use of the WorldClim rainfall model, opting to stop using these data altogether because of the model's self-correlation with elevation, and instead choosing to rely only on measured rainfall and temperature data. Our revised interpretations focus on the strongest relationship we observed, between $\varepsilon$ and mean basin elevation, but we also highlight the correlations between elevation and mean annual rainfall and temperature from gauging stations and temperature loggers, respectively to make interpretations about geomorphic process.

Lines 59-60 There are too many references here. Can they be assigned to each specific topic? E.g., mining (ref), fishing (ref)…

**Initial Author Response:** The activities referred to in this list are interconnected for Tasmanian estuaries, and the studies cited in this list are comprehensive assessments of all activities leading to degradation of Tasmanian Estuaries. However, Augustineus et al. (2010) focuses mainly on mining and Nanson et al. (1994) focuses mainly on tourism. As such, we separated these studies from the rest of the list, per the reviewer's suggestion.

**Additional Author Responses:** Upon major revisions, we focus our narrative heavily on the geology and geomorphology and land use dynamics in the George River basin and significantly scale back our focus on estuarine environments. These references are no longer relevant to the more-focused narrative that is set up in the revised manuscript.

Lines 97-98 Please separate references on 10Be delivery from those on catchment applications.

**Author Response:** $^{10}Be_m$ delivery citations are now separate from application studies.

Line 100 "non-cosmogenic" should be "stable"?

**Author Response:** We use "non-cosmogenic" to refer to $^9Be$ because it is found terrestrially on Earth and to differentiate it from the $^{10}Be$, which is produced only through cosmogenic reactions. While tempted to describe $^9Be$ as "naturally occurring," this would suggest that cosmogenic production of $^{10}Be$ is not a natural process, which it is. We are unsure how else to describe $^9Be$, so we added "stable" to our description of $^9Be$ per the reviewer's suggestion.

Line 103 Harrison et al., 2021 only measured 10Be_m instead of 10Be/9Be. Hence, it should be placed at the beginning of this paragraph.

**Author Response:** We revised this part of the paragraph to better distinguish between which studies used which $^{10}Be_m$ method.

Line 108 "pH…high (>3.9…)" I do not think 3.9 can be considered as a high pH and the partition coefficient of Be can be low (You et al., 1989).

**Author Response:** We thank the Reviewer for directing us to You et al. (1989). This comment and the other Reviewer's suggestion that we reference Aldahan et al. (1999) with regards to $^{10}Be_m$ mobility in the environment. We clarify that soil and stream pH levels are all within the

realm of $^{10}Be_m$ retentivity in sediment grain coatings and that desorption of $^{10}Be_m$ in George River is unlikely.

Lines 119-120 "soil pH". Could you also provide river water pH data if available?

**Author Response:** Both Reviewers' comments encouraged us to take a second look at pH values in stream water. We now present long-term, decadal pH values for George River and Ransom Creek (the only two sites in our field area for which data were available [DPIPWE, 2021]), which is consistently >5 and mostly >6.

Line 146 "drain" should be "drains".

**Author Response:** Changed.

Line 189 It is hard to imagine that the average grain-size of alluvium sediment can be 30-50 mm with moderate precipitation rate of ~1 m/yr and gentle slope. Are there any field photos on the sampling sites (perhaps included in the supplement)? Besides, these are the materials left behind and can not represent most materials that have been transported to the sea, which should be much finer.

**Initial Author Response:** Unfortunately not. We searched for sediment yield/grain size data in both published and grey literature, and we reached out to people in Australia/Tasmania who might know whether such data exist without success. The only data we were able to find were numbers from Knighton (1991), which we cite in this study, although we now clarify that the pre-disturbance grain size data are limited in their applicability across the field area.

**Additional Author Responses:** As noted above, we now present total suspended solids measured at a water intake station in the town of St. Helens, but we remain unable to find or acquire any bedload data.

Line 206 Please give a brief description on the acid used here.

**Author Response:** Revised to be more detailed: 6N HCl.

Lin 245 Which type of regression? Linear?

**Author Response:** Linear. Sentence is now revised.

Line 247 Here TG-1=1.1 km3/yr, but in the text above TG-1=3.8 km3/yr. Please use different terms for both values.

**Initial Author Response:** In this instance TG-1 = 1.1 mm kyr$^{-1}$, which is the average of modelled erosion rates for the TG-1 subcatchment based on linear regression equations for longitude, elevation, and precipitation from Fig. 6, of the subcatchment (area upstream of TG-1 sampling site and downstream of tributary sampling sites). The 3.8 km3/yr value is the product of the measured $^{10}Be_i$ erosion rate and the total catchment area upstream of TG-1. We, however,

discuss the accuracy of the measured $^{10}Be_i$ erosion rate for TG-1 (and TG-9) in this paragraph, demonstrating that it is dominated by erosion in the tributaries and missing a significant contribution of erosion from the TG-1 subcatchment. We make revisions to this section of the Discussion to make our logic and reasoning here more clear, especially as it relates to the value we believe best-reflects the average erosion for the entire catchment area of George River.

**Additional Author Responses:** During our major revisions, we've changed our approach to subcatchments, which included a reanalysis of the mass balance between tributaries and trunk channel sites. We believe much of our original intentions here were lost in overcomplicating what ought to have been a very straightforward comparison. In our new approach we convert the erosion rates of tributaries to masses by multiplying $\varepsilon$ by basin area; we then compare the mass exiting the tributaries to the mass passing through trunk channel sites. They are similar, and the way in which this comparison is presented and discussed now is more straightforward and less complicated.

Line 303 I checked Mishra et al. (2018) and they actually claim that "the regime between ~1000 and ~2200 mm/yr is dominated by opposing relationships where higher rainfall acts to increase erosion rate, but more water also increases vegetation/tree cover, which slows erosion". As such, there is no correlation or even negative correlation between precipitation and erosion rates within the precipitation range of 0.97-1.26 m/yr (Mishra et al. (2018)'s Fig. 7). Hence, this point needs to be rephrased.

**Author Response:** We clarify in the first paragraph of the Discussion that Mishra et al.'s findings also suggest that increased rainfall leads to increased vegetation cover which can slow erosion. We maintain our argument that precipitation drives erosion in George River, which is a highly-localized area of Earth's surface and what controls erosion here may not be reflected in erosion studies at the global scale.

Lines 326-327 and Fig. 9 The close relationship does not mean 10Be_i denudation rates must be correct, especially when the variability in precipitation rates cannot explain the large variability in 10Be_i denudation rates. I think Fig. 9 shows that the 10Be_i measurements in this study should be ok as the George River data can fit in the general pattern over a large spatial scale. However, Fig. 9 also shows some evidence against the precipitation control: although precipitation/elevation may play a role in controlling erosion rates on local scale, such relationship can not be found on a larger spatial scale (east Australia). Besides, the control of mean slope seems to be clear on the same (large) scale. If this is correct, it means that the pattern found in the George River is a very local phenomenon and its applicability is very limited. One suggestion may be that the authors simply claim that their denudation rate data do fit in the large-scale pattern in east Australia and spend much less text on its controlling factors, as I think the highlight is the inter-method comparison anyway. Otherwise, the authors need to explain such inconsistency to convince readers that their conclusion is not only of local impact.

**Initial Author Response:** It is unclear what the Reviewer is suggesting here. It seems the Reviewer accepts the validity of our $^{10}Be_i$ data only because they are consistent with erosion across the remainder of the Great Dividing Range. We discuss in the paper that we observe no relationship between $^{10}Be_i$ erosion and slope for George River whereas $^{10}Be_i$ erosion and slope

are closely related across mainland Australia. Between this comment and a previous comment, it seems the Reviewer wishes to reject our primary interpretation because a modelled relationship between channel steepness and precipitation in a vastly different landscape and tectonic setting says it shouldn't exist. We maintain our primary interpretation that erosion is driven by rainfall in George River and is not related to basin slope, as it is elsewhere along the Great Dividing Range (Codilean et al., 2021). The similarities and differences of erosion in George River compared to erosion on the Australian mainland is already discussed in detail in this paper. We disagree with the Reviewer that our findings have limited applicability. George River may be small and not important for most, but our dataset presents erosion data for a part of the world that previously had none.

**Additional Author Responses:** We maintain that our $^{10}Be_i$ erosion rates are geologically meaningful, but as noted above, we scale back our original primary interpretation – that $\varepsilon$ was related most to precipitation. We focus our revised manuscript on the relationship between $\varepsilon$ and mean basin elevation, which maintains a strong and significant correlation ($R^2 = 0.91$, $p < 0.01$).

Tables & Figures
Table 1 Please clarify if slope and precipitation provided here are basin-averaged values

**Author Response:** Fixed per the Reviewer's suggestion.

Table 2 Q's unit: atoms/cm2/yr

**Author Response:** Fixed (column width was too-narrowly adjusted)

Table 3 Please add a note to explain the meaning of epsilon, E and D_m.

**Initial Author Response:** Fixed.

**Additional Author Responses:** We no longer use or refer to meteoric erosion rates ($E$) in this revised manuscript.

Fig. 1 Please provide a color bar to the precipitation map.

**Initial Author Response:** Added.

**Additional Author Responses:** Given that we no longer rely on the relationship between $\varepsilon$ and mean annual precipitation, we changed the inset for Figure 1 to be a topographic map instead of a precipitation map.

Fig. 2 Caption text is incomplete. Also, what does the white star (St. Helens) mean? City?

**Author Response:** St. Helens is the town of St. Helens and this is indicated in the main text and previous figures.

Fig. 3b The color of the text (Elevation) is different from that of the corresponding symbol.

**Initial Author Response:** There is no map symbology for elevation because the background map is a shaded relief map. We've changed the elevation color symbology to grey, however, since that is the same as the shaded relief map.

**Additional Author Responses:** Figure 3 has now changed completely from that in previous versions.

Fig. 9 Caption text: "B. Comparison" should be "C. Comparison"

**Initial Author Response:** Fixed

References
D'Arcy, M., Whittaker, A.C., 2014. Geomorphic constraints on landscape sensitivity to climate in tectonically active areas. Geomorphology 204, 366-381.
Deng, K., Wittmann, H., von Blanckenburg, F., 2020. The depositional flux of meteoric cosmogenic 10Be from modeling and observation. Earth Planet. Sci. Lett. 550, 116530.
Graly, J.A., Reusser, L.J., Bierman, P.R., 2011. Short and long-term delivery rates of meteoric 10Be to terrestrial soils. Earth Planet. Sci. Lett. 302, 329-336.
Harel, M.A., Mudd, S.M., Attal, M., 2016. Global analysis of the stream power law parameters based on worldwide 10Be denudation rates. Geomorphology 268, 184-196.
Heikkilä, U., von Blanckenburg, F., 2015. The global distribution of Holocene meteoric 10Be fluxes from atmospheric models. Distribution maps for terrestrial Earths surface applications, GFZ Data Services, GFZ Potsdam, Germany.
Lague, D., 2014. The stream power river incision model: evidence, theory and beyond. Earth Surf. Processes Landforms 39, 38-61.
Portenga, E.W., Bierman, P.R., Trodick, C.D., Jr., Greene, S.E., DeJong, B.D., Rood, D.H., Pavich, M.J., 2019. Erosion rates and sediment flux within the Potomac River basin quantified over millennial timescales using beryllium isotopes. GSA Bulletin 131, 1295-1311.
Reusser, L., Graly, J., Bierman, P., Rood, D., 2010. Calibrating a long-term meteoric 10Be accumulation rate in soil. Geophys. Res. Lett. 37.
Singleton, A.A., Schmidt, A.H., Bierman, P.R., Rood, D.H., Neilson, T.B., Greene, E.S., Bower, J.A., Perdrial, N., 2016. Effects of grain size, mineralogy, and acid-extractable grain coatings on the distribution of the fallout radionuclides 7 Be, 10 Be, 137 Cs, and 210 Pb in river sediment. Geochim. Cosmochim. Acta 197, 71-86.
von Blanckenburg, F., Bouchez, J., Wittmann, H., 2012. Earth surface erosion and weathering from the Be-10 (meteoric)/Be-9 ratio. Earth Planet. Sci. Lett. 351, 295-305.
Wittmann, H., von Blanckenburg, F., Bouchez, J., Dannhaus, N., Naumann, R., Christl, M., Gaillardet, J., 2012. The dependence of meteoric Be-10 concentrations on particle size in Amazon River bed sediment and the extraction of reactive Be-10/Be-9 ratios. Chem. Geol. 318, 126-138.
You, C.F., Lee, T., Li, Y.H., 1989. The partition of Be between soil and water. Chem. Geol. 77, 105-118.

Reviewer #2

This manuscript by VanLandigham et al. reports in-situ and meteoric beryllium-10 ($^{10}$Be) based erosion and denudation rates for a small catchment located in Tasmania. The study is presented as aiming at two main goals: evaluate the long-term sediment delivery to the estuarine zone, which has implications for policy regarding land use and coastal ecosystem preservation; compare the estimates of erosion / denudation yielded by the two "varieties" of $^{10}$Be.

Although I find the overall manuscript nicely written and well structured, I have concerns about (1) unclear terminology; (2) interpretation of meteoric $^{10}$Be based erosion rates - see detail provided below. In addition, I must say that I noticed is some "dissonance" between the way the scientific question is set up in the introduction (revolving much around the issue of increased delivery of sediment to coastal ecosystems under land use change in modern times), and what is discussed in the rest of the paper (controls on long-term erosion rates, and more methodological aspects about the comparison between in situ and meteoric $^{10}$Be). As a consequence, the manuscript would benefit from a clarification of its goals.

**Initial Author Response:** We thank the Reviewer for their thoughtful comments on our manuscript. We agree that there is dissonance in the Introduction and we revised Section 1.1 "The Importance of Erosion of George River" to focus on understanding the drivers of erosion here and less on how erosion rates might be used to help ecological restoration efforts of Georges Bay.

**Additional Author Responses:** Upon major revisions to the manuscript, we scaled back the meandering narrative that had been presented in the Introduction, choosing to focus it more on the geomorphology of the Great Dividing Range and Tasmania and less on coastal ecosystems.

1) Terminology
The first concern I have about terminology is about the fact that the authors keep on calling the $^{10}$Be$_i$-derived "epsilon" an "erosion rate" (e.g. Appendix A and throughout the manuscript), in apparent opposition to $^{10}$Be/$^9$Be-derived $D_m$ called a "denudation rate" (e.g. Appendix A). Both are denudation rates (sum of physical and chemical removal of matter), really, except if a significant fraction of the chemical weathering occurs at depth < 2 m, typically. I understand that the difference between erosion and denudation rates might be very small if the chemical weathering rate is negligible, but this is discussed nowhere in this paper. In addition, in tectonically stable landscapes like the one under study here, and at erosion rates of ~10-20 mm kyr$^{-1}$, it's very possible that chemical weathering is a significant component of total denudation. Anyway, this might be a purely terminological issue, but one that relates to fundamental understanding of the proxies, such that I think this needs to be fixed before the manuscript can even be considered for publication.

**Initial Author Response:** The Reviewer is correct that (1) both $\varepsilon$ and $Dm$ are denudation if there is no loss of mass to chemical weathering and that (2) chemical weathering can be significant in settings such as George River. However, we do not know *a priori* that chemical weathering is negligible and precisely why we compare an erosion metric to a denudation metric in this study. It is true that we presume there to be negligible chemical weathering in George River, based on

the topography, geological setting, and bedrock, and therefore we also consider George River to be a location where the two independent measures of $\varepsilon$ and $Dm$ can be compared. We revised our Introduction to make it clear how we define erosion and how we define denudation in this study. Given that $^{10}Be_i$ is used and has been used to measure erosion since Lal's (1991) seminal paper, in which $\varepsilon$ is defined as the erosion rate, we continue its use here, and because $^{10}Be_i$ is measured from quartz minerals, which are highly resistant to chemical weathering, it may not always reflect total denudation. For this reason, we choose to distinguish $\varepsilon$ in this study from $^{10}Be_m/^9Be_{reac}$ based measures of total mass loss (von Blanckenburg et al., 2012; i.e. denudation), and we choose to use $Dm$ as our abbreviation for this measure to differentiate it from $^{10}Be_m$-based measures of erosion, $E$ (Willenbring and von Blanckenburg, 2010). With regards to the extent of chemical weathering in George River, we were able to acquire water quality data from the mouth of George River in the town of St. Helens, which includes turbidity, conductivity, and dissolved ions. Using these data, we will be able to ascertain some degree of chemical weathering and address the Reviewer's primary concern in this comment.

**Additional Author Responses:** As noted above, we have made significant revisions to the nomenclature used in this study. In situ $^{10}Be$ studies have long used the terminology "erosion rates" and recent meteoric denudation work refers to "denudation." In our revised manuscript we acknowledge these histories, and we present $\varepsilon$ and $Dm$ in similar units. Both in the Introduction/Methods and Discussion of the revised manuscript, we consider what $^{10}Be_i$ erosion and meteoric denudation means when there is deeper chemical weathering and discuss the limitations of each method. We were also able to find and acquire water quality data for a water intake station in St. Helens, which allows us to better make sense of the $^{10}Be_i$ erosion rates and $^{10}Be_m/^9Be_{reac}$ denudation rates at TG-9. Unfortunately, due to the lack of water quality data elsewhere in the field area, this discussion is limited to the basin as a whole.

Second, it appears surprising to me that the $^{10}Be/^9Be$ ratio is here called "$^{10}Be_m/^9Be_{reac}$", whereas in most recent studies about this ratio the term "$^{10}Be/^9Be_{reac}$" was used. I think I understood why the authors have done so: in this study $^{10}Be_m$ is measured by digestion of the bulk sample (l. 203), rather than on the chemical leachate of the "reac" fraction on which $^9Be$ is measured (l. 206-207). This is fine as long as $^{10}Be_i$ - which is also released during bulk digestion - is negligible compared to $^{10}Be_m$, which seems to be the case (Table 2 shows that the latter us typically two orders of magnitude smaller than the former). If I am right, I now feel like it would have been nicer if the authors have explained that themselves in their manuscript, instead of leaving the job to the reader / reviewer. If I am wrong, this terminology is simply very misleading.

**Author Response:** We revised the methods to make it clear that $^{10}Be_i$ is incorporated into the $^{10}Be_m$ measurement, but that the amount of $^{10}Be_i$ is two orders of magnitude smaller than the overall $^{10}Be_m$ measurement, thus being negligible.

Finally - and this is a concern of lesser importance - "Q" is an unusual notation for the flux of meteoric $^{10}Be$, at least in the most recent literature. I also note that it is not really defined in the text: l. 211 is the first of this term, with no definition. On a more general note, having the equations just at the end of the manuscript makes the reading and the evaluation very difficult. To come back to this Q-notation, this is particularly misleading since in the recent $^{10}Be/^9Be$

literature, Q was used to denote discharge, which might be an important parameter for estimating denudation rates for $^{10}$Be/$^9$Be ratio in cases where Be retentivity onto particles is not complete (see below).

**Author Response:** We had chosen to use *Q* to denote meteoric $^{10}$Be delivery to Earth's surface because this was the notation being used in older literature (i.e. Brown et al., 1988); however, we recognize that Monaghan et al. (1986) used *F* as the notation for $^{10}$Be$_m$ delivery to Earth's surface and since this is what other researchers, including authors on this paper, have used in recent work, we changed all instances of *Q* to $^{10Be}F_{met}$.

2) Interpretation of $^{10}$Be$_m$- and $^{10}$Be/$^9$Be-derived rates
The main point made by the authors from their comparison of $^{10}$Be$_i$- and $^{10}$Be$_m$-derived rates is the 5-6 times greater rates (leaving aside the fact that one might reflect total denudation and the other erosion rates, which in itself would deserve some discussion in the manuscript) obtained from the latter method (Figure 8). To me, this is simply a grain size effect. Indeed, and as acknowledged by the authors, 10Be_i tends to weakly depend on grain size itself (l. 181-182), $^{10}$Be$_m$ is strongly grain-size dependent (e.g. Wittmann et al., 2012). Hence any erosion rate inferred from $^0$Be$_m$ will be affected by grain size effects. I understand from section 3 that both varieties of $^{10}$Be were measured on the 250-850 µm fraction, which is much smaller than the average grain size of sediment delivered to the alluvial plain of the George River (l. 179-180).

Now, I admit that one could expect much greater $^{10}$Be$_m$ concentration in the analyzed, relatively fine fraction (meteoric Be being enriched in fine fractions offering large mineral surface areas) than in the "representative" sediment generated in the catchment, thereby likely leading to an understimate of erosion rates by $^{10}$Be$_m$ (see equation in Appendix A) compared to what could be deduced from $^{10}$Be$_i$ - even if the latter includes some fraction of chemical weathering. The opposite observation is made in Fig. 8. But in my opinion this apparent contradiction shows even more how these estimates need to be discussed in the frame of the limitation / inherent assumptions of each proxy, an aspect of the discussion that is critically missing from this manuscript. I note that the equation used to calculate D$_m$ in Appendix A accounts for such grain size effects through the term $^9$Be$_{min}$/$^9$Be$_{reac}$ and leads to estimates of denudation rates (hence more directly comparable to 10Be_i-based estimates) that, although with some significant scatter, lie uniformly around the 1:1 line (between 1:3 and 2:1) in Fig. 8, rather than showing a systematic overestimate.

**Initial Author Response:** We agree with the Reviewer that the differences between *ε* and *E* are due to grain size bias in our $^{10}$Be$_m$ measurements. As described to Reviewer 1 above, we now justify our disregard for geological meaning in our calculated values of *E* and remove interpretation of *E* from the Discussion altogether. We are not concerned about the grain size effect on $^{10}$Be$_i$ data, as noted in the manuscript, because $^{10}$Be$_i$ in low-elevation, temperate settings is rarely affected by grain size bias (van Dongen et al., 2019) and evidence for deep landslides that could otherwise dilute $^{10}$Be$_i$ in stream sand is not present in the field area.

**Additional Author Responses:** Per suggestions by Reviewers and Editors, we remove meteoric erosion rates (*E*) from this revised manuscript altogether.

Another issue with the use of the $^{10}$Be/$^9$Be ratio here is the potential bias induced by loss of Be to solution. The equation for $D_m$ in Appendix A here does not account for such poor retentivity. And in contradiction to what the authors say, a pH in the range 4.0-5.5 (l. 119-120) entails significant loss of Be to solution (e.g., Aldahan et al, 1999). This loss can in turn lead to a strong bias in both E and $D_m$ estimates, particularly in situation where the ratio between discharge and erosion rates (Q/E, "Q" here being understood as water discharge, see my comment above about terminology) is high. I think this field setting, which is reasonably wet and tectonically quiescent, is one where this ratio is expected to be high. Taking the 10Be$_i$-derived estimate of erosion rate of ~20 mm kyr$^{-1}$ (Table 3) and the precipitation of ~ 1000 mm kyr$^{-1}$ (Table 2), and assuming a evapotranspiration factor of 0.5, I get a Q/E ratio of around $10^4$ L/kg, which at pH 5 corresponds to an overestimation of E from $^{10}$Be$_m$ of ~100% and of $D_m$ from $^{10}$Be$_m$/$^9$Be$_{reac}$ by 10 to 100% depending on the fraction of $^9$Be in the "min" fraction (see von Blanckenburg et al., 2012). Now, these back-of-the-envelope calculations might well be wrong, and surely can be refined, but clearly this potential issue clearly has to be discussed in more detail in the manuscript before a comparison with another proxy can be made.

**Author Response:** We thank the Reviewer for directing us to the Aldahan et al. (1999) study. This comment and the previous Reviewer's suggestion to refer to $^{10}$Be partition coefficients in Yiou et al. (1999) led us to try to better understand streamwater pH throughout our field area. To this end, we find that streamwater pH in the main stem of George River at St. Helens and in the Ransom Creek tributary is >5 and has been since measurements began in the 1980s. This supports our interpretation that pH conditions in George River basin are not likely to induce $^{10}$Be$_m$ loss to solution.

\*\*\* Other comments \*\*\*
- l. 43: "northern tropics" is a bit misleading (to me, the "northern tropic" is the Tropic of Cancer, which is not what the authors are talking about here, I guess).

**Author Response:** We mean the northern reaches of Australia's Great Dividing Range, which is north of the Tropic of Capricorn. We edited the text to better reflect this geography and not mislead the reader.

- l. 319: "to not have not had" -> problem with this sentence.

**Author Response:** Revised to fix the wording problem.

- l. 376-383: Not sure this paragraph (about controls on long-term rates) belongs to this section (which is supposed to be about trunk stream vs. tributary sediment supply.

**Initial Author Response:** The first paragraph is a discussion of where sediment that is passing through trunk channel sites originates, from the tributaries, from below the tributaries, or from the entire catchment? The first paragraph describes our interpretation that sediment at trunk channel sites originates in tributaries, and it describes how we derive a value for ε for the whole George River basin. The second paragraph describes how this whole-catchment value for ε compares to whole-catchment values of ε across the rest of the Great Dividing Range.

**Additional Author Responses:** The Discussion for the revised manuscript has been heavily altered, and this section now mostly focuses on the big picture look at mass loss dynamics in the whole of the George River basin, and exploring how the mean erosion rate for the field area fits within the context of the Great Dividing Range is still important. We believe our revisions and more-streamlined narrative help with flow of the Discussion.

- Table 2: Strictly speaking, "$^{10}Be_{met}$" is not defined anywhere - I think the authors mean "$^{10}Be_m$", which is used throughout.

**Author Response:** Fixed.

- The end of Figure caption 2 is missing.

**Author Response:** Text-box issue. This is now fixed.

- Fig. 5B and pie chartes in Fig. 7: I must admit I did not understand exactly what purpose the calculation and presentation of this "hillslope erosivity" serve in the manuscript.

**Initial Author Response:** We mention and use Kidd et al.'s (2014, 2015) "hillslope erosivity" metric because we did not want to neglect their work and the maps of soil erosivity for Tasmania they produced. Despite being based on a calculated value via a multivariate equation, Kidd et al.'s "erosivity" designations (i.e. Extreme → Very Low) are qualitative in nature, which makes it difficult to summarize "erosivity" at a catchment scale or to compare our numerical catchment-wide erosion rates to Kidd et al.'s "erosivity." By determining that "erosivity" is ultimately significantly related to slope (Fig. 5B), we are able to acknowledge Kidd et al.'s work and use hillslope angle as a proxy for erosivity. We've reworded this last paragraph of the Methods to explain this better.

**Additional Author Responses:** Additionally, after recalculating $^{10}Be_m/^9Be_{reac}$ denudation rates using $^{10Be}F_{met}$ values based on measured precipitation in the field area, rather than gridded datasets that are self-correlated to elevation, we find that there is some relationship between $D_m$ and the % of basins that have High-Extreme erosivity, and we use this to suggest that the meteoric $^{10}Be$ system has been affected by the intensive land-use history of the field area whereas the in situ $^{10}Be$ data appear to not have been affected.

---

## Editor Decision (ED1)

Comments from Associate Editor and Principal Editor on MS geochron-2021-23

Dear Eric et al.,

As you know, I have received two reviews for your MS and I´ve carefully checked your responses. Unfortunately, due to the system setup of the journal, I do not get the track-changed version of your replies at this stage, only *after* I have made my decision. For the record- this is nothing I can change, but it´s simply due to the system, but it makes, in my view, my life as an Associate Editor harder. The reason is that I cannot in detail see your changes, and some of them are hard to judge without the connection to the text.

One of these issues where seeing the track-changed version would clearly help me is your treatment of terminology, an issue that both reviewers commented about. I agree that back in Devendra´s time, the term *erosion rate* was used for *in situ*-derived rates. But I think that the community has clearly advanced on this in the meantime. I don´t mean that one should *per se* call *in situ*-derived rates "denudation rates", because weathering rates might be very very low, but without seeing the numbers and changes, this is difficult for me to judge. I however have a hard time believing that we are dealing with a landscape where chemical weathering is totally absent. (I read from your response that there is some water chemistry data available, but even a multi-decadal record in TDS still has time scale issues when comparing to millennial-scale cosmogenic rates). So, even if it´s only a super small portion of total denudation, the *in situ*-derived rates should be called denudation rates and are thus directly comparable with $^{10}Be/^9Be$-derived denudation rates (btw, I´ve calculated the reac/min fraction (eq. 9 in my 2015 paper) using your data, and the values are around 0.25-0.3, which is a range in Be-related weathering degrees found by others, i.e. indicating 25-30% of a weathering degree, meaning that weathering cannot be absent). If it wasn´t for this comparison, I would not care too much, but I stress here that it would make the MS much easier to digest if the terminology was adjusted. Also on this matter, I would suggest that you present either in mm/kyr or in t/km²*yr, at least in the text (I would not opt for "Mg/km²*yr", for reasons of common acceptance. Also, it´s in my view totally unnecessary to provide $D_{insitu}$ AND in situ-derived sediment fluxes in the Figures, as these essential only differ by density). Tables could provide both values. Note that for $D_{insitu}$, one also needs a density assumption, so one could easily convert these to t/km²*yr, or, if lithology is uniform, as claimed, one could use the same density to convert $^{10}Be/^9Be$-derived denudation rates from t/km²*yr to mm/kyr. Given that you only have one major lithology, this would be a fair assumption. Regarding erosion rates from single $^{10}Be$ concentrations, I agree with both reviewers, and welcome the removal of their interpretation from the Discussions, as their presentation did not help to clarify controls among the variables. I hope that Fig. 6 and 7 will benefit from this removal and adjustment of terminology, as these are particularly hard to digest. Regarding the comparison between denudation/erosion rates, I think it would help to add the "Appendix" equations into the main text, at least the two main ones (Lal, von Blanckenburg). I´m not sure where the "Appendix" equations will end up, hence if they are not instantaneously visible for the reader, this would help an audience that is not so familiar with the topic (assuming that the general audience of GChron is not familiar with this).

I´m happy to see that you take the effort and compare *in situ* and $^{10}Be/^9Be$ denudation rates. I am a bit puzzled however by the overall scientific outcome of this comparison. $D_{insitu}$ show a clear trend with precipitation (but please see Greg Balco´s comment on this at the end here, in support of the concerns raised by Reviewer #1.), whereas $^{10}Be/^9Be$ denudation rates are all over the place if one takes the entire dataset. Why is that so? Regarding the methodology, please clarify in a response if the used rainfall-derived $F^{10}Be_{met}$ could play a role in obscuring a trend between denudation rates and precipitation. I understand that you favor Graly et al. (2011) values, but that makes me a bit worried if there could be some intrinsic dependence. Btw, are you using a local rainfall-relationship or a global one? I don´t think that is specified in the paper. Further, you use the 2.5 ppm value suggested by the von Blanckenburg paper (2012) for $^9Be_{parent}$. Unfortunately, we have seen that this is probably only valid for larger basins, hence I would really welcome some measured values, even if it´s one sample per lithological unit….The basalts will clearly deviate from that value, but also granites can show some heterogeneity (see Dannhaus paper) and in such small basins (your tributaries), this might really be an issue. In order to

check out the potential magnitudes, the percent areal coverage of each lithological unit per basin might give a clue? At least discuss please in which direction the associated changes would go for the $^{10}$Be/$^{9}$Be-derived denudation rates, if you either over- or underestimate $^{9}$Be$_{parent}$ (but, in order to resolve the missing trend, there must be some systematic pattern). Another way to assess the accuracy of $^{10}$Be/$^{9}$Be-derived denudation rates would be to do the Q/E calculation suggested by reviewer #2 (note that even at "high" pH values of 5, the bias might be between 10-100% on D's). At least that would give you an idea where the problem might be. Lastly, there might the geological reasons for the absence of this trend with precip (if it´s at all there?), which might be e.g. overland flow? However, I´m overall not too worried, since the overall numbers are not so far apart between in situ and $^{10}$Be/$^{9}$Be-derived denudation, but still a bit puzzled.

In conclusion, I think this paper needs another round of revisions. Given that one reviewer opted for "major", the other for "reject", I will send it out again. I´m looking forward to a new version.

All the best,
Hella

**Below I give some more technical / structural advise that could improve your paper:**

Line 101: Just to be accurate here: At least the studies I was involved in, we use the "$^{10}$Be$_{reac}$/$^{9}$Be$_{reac}$" ratio. We do not perform bulk extraction of meteoric $^{10}$Be.

Line 187: "No native Be" was detected. It´s hard to believe that there was "no" native Be found, because this is a matter of which instrument you use. We measure $^{9}$Be in seawater ranging down to 1 ppt and lower using ICP-MS. Please re-phrase such that "relative to added $^{9}$Be carrier amounts, no significant Be was found" or something like that.

Line 240: This definition of sediment fluxes derived from D$_{insitu}$ should come in the paragraph above, where these are mentioned.

Line 242: How were the "average modelled" rates derived? Not clear to me. If these are from a simple mass balance. i.e. summing up the respective yields and then dividing by summed area, then please say so.

Fig. 4 The "historically burnt areas" are really hard to make out in the Figure. Would suggest that you use some other color / make larger.

**In addition, editor Greg Balco had the following comments:**

Reviewer 1 points out that there is not an obvious physical means by which the relatively small variability in precipitation could cause a relatively large effect on the erosion rate. This is true, and there are some additional aspects of this part of the paper that seem to need attention. First, the smoothed data set used to calculate precipitation for all the basins (Worldclim) appears to have been generated by using elevation and distance from the coast as smoothing parameters, so precipitation is not an independent variable, it is already defined to be correlated with elevation. Thus, for example, the R^2 of 0.93 in Fig. 3C appears to just be reproducing how the smooth data set was created. In other words, precipitation and elevation appear to be effectively the same independent variabie in these regressions. Second, their Figure 3 shows that the smoothed data set is doing an extremely poor job of reproducing rain gauge data -- the true variability from rain gauges is twice as big as predicted by the smooth data, so the question brought up by the reviewer about the range of precipitation is already based on an incorrect range. Overall, the aspect of the paper dealing with whether or not precipitation is the main control on erosion needs rethinking. Clearly a precipitation control is possible, but I am not sure how you would exclude the possibility that elevation is actually the controlling variable, either through, for example, a relationship between elevation and local relief or perhaps between elevation and temperature, perhaps involving freeze-thaw processes. A possible approach to this would be to take

Reviewer 1's observation that there is a physical limit on how much variation in the erosion rate could be accommodated by the available range in precipitation, and evaluate whether or not that is adequate to explain the observed variation, or if an additional factor related to elevation would be required. In any case, this aspect of the discussion needs more careful attention.

---

## Author Response (AR2)

Dear Dr. Wittmann-Oelze,

Thank you for a prompt return and positive comments with regards to our revised manuscript. We recognize there are a few minor issues remaining before you consider the manuscript for publication, and we address those below.

Additionally, the References list should now be in concordance with the EGU Journal guidelines for reference style.

Kind regards,

Dr. Eric Portenga (corresponding author)
Leah VanLandingham, B.Sc.
Dr. Edward Lefroy
Dr. Amanda Schmidt
Dr. Paul Bierman
Dr. Alan Hidy

*Most recent correspondence from Geochronology*:

Comments from Associate Editor Wittmann to Van Landingham et al., GChron

Dear Eric et al.,

Many thanks for the thorough revision on this manuscript. I really appreciate your efforts, which in my view have significantly improved the MS. I think you have in most cases corresponded adequately to my and the reviewers comments. There is one issue remaining, but I think that can be addressed with minor efforts. I would suggest, for time-efficient handling, that you address the (minor) comments I have below, after which the MS could be published.

With best wishes, Hella

General comments:

1) "Sensitivity analysis" of Dm under changing 9Beparent and F10Be_met. As mentioned in my and a reviewers first comments on V1, I am missing a short discussion in which directions Dm´s would go if, e.g. 9Beparent would be higher than 2.5 ppm (à Dm would decrease), or lower (basalt units), how changes in F10Be_met would affect Dm, and how retentivity would affect them (see the Q/E approach mentioned by reviewer #2). Of course, this assessment would be largely hypothetical (at least for 9Be because there simply isn´t any data on 9Be_parent for this catchment). However, previous studies have shown that a factor of 2 in 9Beparent even in granites, and something in the order of 30% for F10Bemet can be easily the case. Specifically for F10Bemet, wouldn´t the use of precipitation-derived [elevation-controlled…] meteoric flux counterbalance any elevation dependency? (Perhaps one way around this is using the basin-wide average F10Bemet value from the CGM. Having values for each catchment doesn´t benefit the discussion if their reliability is unclear. Also, this would normalize at least this effect, and the remaining effects could be better identified, perhaps…). I think looking at this carefully would put the Dm into the right frame, as a factor of 2 disagreement with e (or agreement, however you want to call it) is okay for a method that is still in its infancy. That´s why my main critique is that I would not generally call the 10Be/9Be-derived denudation rates as being sensitive to land use. We simply don´t know how the 9Be_reac plays into this, whereas 10Be_met-derived erosion rates can be clearly affected by e.g. soil loss. Also it seems like from Fig 7 that the way in which Dm and e scatter around the 1:1 line does not follow a trend with erosivity. When comparing to the landuse data, it looks like not all tributaries that have high landuse (erosivitiy) values also have Dm that are higher than e (i.e. for T6-8, Dm are actually lower than e). This clearly points in my view at some random effects (for which in my view catchment-wide variability in 9Beparent would be a good candidate). For these reasons, I would ask you to tune down these statement about Dm and land-use affected areas.

**Author Response:** With regards to the various requested sensitivity analyses.

**⁹Be_parent:** Knowing that we are unable to measure bedrock from the field area at this time, we dug further into the literature to see if we could find other studies that measure natively occurring beryllium in similar lithologies. To that end, Grew (2002; *Reviews in Mineralogy and Geochemistry*) compiled a compendium of everything we (he) knew about beryllium at the time. In the introductory chapter, Grew writes that Earth's crustal average concentration of Be is 3 ppm, that most Be-bearing minerals contain it in <10 ppm range, but it is not unheard of for Be to be found >100 ppm, in terrestrial systems Be is found at <1 ppm in (ultra)mafic lithologies, and that Be can range 10-fold within the same igneous complex. In the same compendium on beryllium, London and Evensen (2002) present a table of typical Be concentrations from felsic granites ranging from 1.6–160 ppm, but for S-type granites or those that are tin-bearing – the classification of those in our field area (Higgins, 1985) – the range of Be is from 2.3–130 ppm (n = 11, average = 18 ppm). Yet another study on tin-bearing biotite granites in Alaska (Sainsbury, 1964) illustrates a range of Be concentrations from 2–26 ppm (n = 5, average 16.6 ppm). In this study, Sainsbury cites work done by Beus (1962; *Beryllium: Evaluation of deposits during prospecting and exploratory work*), who measured beryllium concentrations of >200 granites across the former Soviet Union and China (average ~5 ppm), noted that the average concentration in the complete subset of biotite granites is 4.1 ppm. In our study, we initially used

the crustal average value for Be of 2.5 ppm, which was used in von Blanckenburg et al. (2012), who therein also cite Grew (2002); this 2.5 ppm value is clearly on the low end of estimates for beryllium content in felsic biotite granites.

We now believe our original use of 2.5 ppm was not the best-estimate (although reasonable) of beryllium content in our field area based on lithology alone, and we know that Be could, in rare cases, be as high as 130 ppm. Thus, we now make the following changes to our analytical approach: (1) We use a $^9Be_{parent}$ concentration of 4.1 ppm in our final calculations of $D_m$ because this is the average of biotite granites comprising a subset of >200 granites (Beus, 1962). (2) We discuss that we did not measure bedrock and that bedrock estimates of Be could be as low as 2.5 ppm (Grew, 2002; used by von Blanckenburg et al., 2012) and use this value to calculate $Dm$ for all samples based on this low estimate; these values of $Dm$ are those from the previous versions of this manuscript. (3) Given that Beryllium concentrations can exist >100 ppm – albeit rarely – it is unlikely (but possible) that Be in our field area is as high. Thus, we take a more conservative approach to estimating a high end of the possible range for Be in lithologies similar to those in our field area and calculate $Dm$ for all samples using a value of 18 ppm (average of 11 S-type granites and tin-bearing granites compiled in London and Evensen [2002]). We show all of these values in a redrafted Figure 7 (now figure 8) with most likely $Dm$ values, calculated from 4.1 ppm, compared to $\varepsilon$ values for the same sample sites, and we also show a likely possible range of $Dm$ values if Be concentrations were low (2.5 ppm) or high (18 ppm). The overall result is that $Dm$ values are very sensitive to $^9Be_{parent}$ concentrations in bedrock – an important finding – and that $Dm$ for individual samples in our field area could range from being 2x higher than $\varepsilon$ to being consistently <3x lower than $\varepsilon$. The higher the concentration, the lower $Dm$ values become.

$^{10Be}F_{met}$: We also carried out a sensitivity analysis on the meteoric $^{10}$Be delivery rate and how it affects measured values of $Dm$ (assuming a similar, constant value of $^9Be_{parent}$ of 4.1 ppm). In doing this, we identified another study from New Zealand that presents an additional measure of $^{10Be}F_{met}$ (Graham et al., 2003; *Geochimica et Cosmochimica Acta, 67(3)*) and now include their data in our sensitivity analysis of how different measures or estimates of $^{10Be}F_{met}$ affects our calculations of $Dm$. In our sensitivity analysis, we find that measured values of $^{10Be}F_{met}$ from New Zealand (Reusser et al., 2010a; Graham et al. 2003) yield $Dm$ values that would be consistently greater than those we present in this study using the Graly et al. (2011) precipitation-based estimates of $^{10Be}F_{met}$. The $^{10Be}F_{met}$ based on atmospheric-depth integrated rates (Masarik and Beer, 2009; Willenbring and von Blanckenburg, 2010) yield $Dm$ rates that are consistently lower than those we present in this study. Using Holocene-averaged $^{10Be}F_{met}$ rates (Heikkilä and von Blanckenburg, 2015) yields $Dm$ that are consistent with those that we present in this study using basin-specific estimates of $^{10Be}F_{met}$ from Graly et al. (2011).

Thus, barring the low end-member models for $^{10Be}F_{met}$ (Masarik & Beer; Willenbring and von Blanckenburg) and the measured values of $^{10Be}F_{met}$ for New Zealand, hundreds of km away (Reusser et al., 2010a; Graham et al., 2003), we find confidence in the way that our estimates for $^{10Be}F_{met}$ from Graly et al. (2011) and measured rainfall from Australian Bureau of Meteorology precipitation gauging stations yield $Dm$ values that are consistent and within the range of those we would have calculated had we used the Holocene-averaged $^{10Be}F_{met}$ rate (Heikkilä and von Blanckenburg) from a dataset that has a much coarser resolution. We present a new figure in the

Discussion (Fig. 11) that shows the comparison of *Dm* from Graly et al. versus *Dm* calculated using $^{10Be}F_{met}$ from GCMs and measured rates from New Zealand.

It was suggested that calculating *Dm* using a global climate model would be preferable to using elevation-dependent mean annual precipitation, but we find in our sensitivity analysis that *Dm* using Graly et al.'s $^{10Be}F_{met}$ values plot squarely between *Dm* using $^{10Be}F_{met}$ values from either global climate model, in which case the choice of source of $^{10Be}F_{met}$ values does not matter, at least for this study. These details are now included in the discussion.

*Dm* calculations are clearly sensitive to the values of $^9Be_{parent}$ used, but not necessarily the choice of $^{10Be}F_{met}$ values. With regards to land use, no matter the $^9Be_{parent}$ used, the result is simply a shift in the overall magnitude of *Dm* but the relationship to the % of land-use classified as "High" to "Extreme" erosivity does not change, in which case we are unable to suggest that measurements of *Dm* are not affected by land-use.

In the spirit of transparency and openness to the broader geomorphology community who might be interested in using von Blanckenburg et al.'s (2012) denudation method – one that the Associate Editor expresses is in its infancy – it is not appropriate for us tone-down our discussion on land use and *Dm*, but we have presented the information required for readers to come to their own conclusions about these findings (e.g., $R^2$ value, p value, description of "erosivity", etc.).

2) Regarding the use of "Mg/km2*yr", I can follow your argument about metric vs. imperial tons and SI units. However, I would still suggest to use the more commonly used "t(metric)/km2*yr" (simply define "t" as being metric tons" somewhere). "Mg" is just very clumsy and not often seen in use.

**Author Response:** As American authors writing to an international audience, it is important for us to be clear in the units we use, and we know that some readers will inadvertently misinterpret our use of "t" to mean imperial tons, whereas there is no ambiguity in using "Mg." The use of SI units, such as Mg, is also the required unit notation for EGU journals, including *Geochronology*, so we are just being consistent with standard Author Guidelines. We have retained Mg.

3) The Methods part is very long- Would suggest that you use subheadings to give the section more structure.

**Author Response:** Done

Line comments:

100: "mass loss from" (typo)

**Author Response:** Done

Fig. 1: Please include a lat/long grid

**Author Response:** Done

118: 10Bem is desorbed under LOW pH conditions.

**Author Response:** Done

125: I know this sounds picky, but this should be "10Bem/9Bereac"-derived denudation rates (Just to avoid confusion with "10Bem"-based erosion rates). Here and elsewhere (e.g. heading section 4.2, line 278).

**Author Response:** Done

141…,located in northeastern Tasmania,

**Author Response:** Done

Fig. 3: Caption sais that "at least one full year of recorded data…". From Table 1, I get the impression that the data coverage is much better (at least 4 years)?

**Author Response:** Yes, there are four years of data, and each has data for the entire year for at least one year (as opposed to a year where there is data available, but not a full year).

Table 4: Please change "Q" in D_m eq. to "10BeFmet".

**Author Response:** Done

172: over which time period has this mass been removed?

**Author Response:** Decades since this is the timing of disruptive land use within the field area.

195ff: Note that grain size effects may impact the 9Be_reac/9Be_min ratio when measured in river sediments due to sorting (not for soils!), because 9Be_reac is potentially enriched over 9Be_min in finer particles. Hence, the D equation is not fully grain size independent (even though grain size variations in the 10Be_reac/9Be_reac are minor).

**Author Response:** We clarify that normalization minimizes grain size dependence in stream sand.

Again 195: "10Bem and the reactive as well as silicate-bound 9Be phases (9Bereac, 9Bemin, respectively)….

**Author Response:** Done

200: That equation/approach was originally suggested by Brown et al., 1988 (ESPL).

**Author Response:** We added a citation to Brown et al. (1988) here as well.

217: What does "to a single point" mean?

**Author Response:** Because we use the CRONUS online calculator (Balco et al., 2008), which requires a single set of coordinates and a single elevation for input in order to scale in situ $^{10}$Be production, we need to find a way to summarize the elevations throughout the basin and a way to summarize the latitude. Rather than taking a mean value for elevation and latitude, we use the hypsometry of the basin to find an effective elevation, and we use the specific spatial geometry of the basin's drainage divides to determine an effective latitude, all of which is described in Portenga and Bierman (2011).

280: The depth of regolith per se does not tell anything about mobility of 10Be_m. You could use that depth to estimate the integration time scale of Dm (Willenbring & fvB 2010), but mobility itself can only be evaluated from pH, discharge, resulting Kd, dissolved v. reactive measurements, reactive 9Be vs. 10Be….So, I am not sure that the text for regolith depth (around lines 285) is needed.

**Author Response:** The depth to regolith does not tell anything about mobility of $^{10}$Be$_m$, but it does indicate whether the potential for $^{10}$Be$_m$ mobility is low or high. Deeper regolith offers more potential for $^{10}$Be$_m$ mobility below the spallogenic $^{10}$Be$_i$ production zone. We therefore believe we needed to let the reader know we considered this possibility.

Table5: If I type in all numbers that you give, I get slightly lower values for Dm- e.g. 54 instead of 60 t/km2*yr for TG5. I think that lower value was the value presented in the first version of the MS (Table 3). I note that some of the values (Version 1 vs this version) changed. Why ? Flux numbers are the same as before, even though you used now different precipitation rates? Please check all calculations again.

**Author Response:** The previous version of Table 3 had not been updated with Graly et al.'s $^{10Be}F_{met}$ values based on measured rainfall rather than WorldClim. This is now fixed.

398: frost-cracking: this reads like there is basin-wide temperature data (and not only data from point stations). Would you like to put that into a map? Seems like frost cracking plays a dominate role on e (conclusion), which is why I would present such data somewhere.

**Author Response:** Temperature data (Figure 3) are given in Table 1.

411: remove the second "of". What is "stymy" erosion?

**Author Response:** Stymy, as in to slow down; Mishra et al. (2018) suggests that increased rainfall leads to an increase in vegetation cover, which serves to slow erosion down.

414: Has % of bedrock outcrops been mapped in the George River basin, or is there data from landslide occurrence?

**Author Response:** Not that we were able to find.

424: The part of the sentence on absent long-lasting dilution effects needs a justification. The whole section (420-430) would benefit in my view from a topic sentence around the question whether e/ in situ cosmogenic nuclide concentrations are at steady state or not in the basin, and then present the arguments. Actually, if you discuss whether e are at steady-state, why not do the same for Dm? Clearly, 10Be_m concentrations are more sensitive to e.g. recent soil loss/general anthropogenic disturbances (see e.g. Belmont et al., 2014, ESPL, as an example for the available literature body on this topic). So, for 10Be_m concentrations this could be discussed. In this respect, the paper would benefit in my view from a sort of sensitivity analysis of Dm for changes in F10Bemet and 9Beparent (see my initial comment from the first round, mentioned above, too).

**Author Response:** We added a figure to the Field Area showing that our samples come from streams with concave-up profiles (Fig. 5), an indication that they are in steady state. We also reiterate this in the Discussion where requested by the Associate Editor. We also agree with the Associate Editor here, that $^{10}Be_m$ is sensitive to soil disturbance, which is why we present a figure showing the relationship between $^{10}Be_m/^9Be_{reac}$ denudation rates and erosivity (former Figure 8, new Fig. 9).

429: I don´t understand the part about "normalized along with bedload characterisitcs". Are your referring to the decrease in mean particle size ?

**Author Response:** This was unclear and we've revised the sentence for clarity.

443-446: That is a long sentence….almost "German". Please split. Is "Conservation and Protective Native Land Cover" a government project? (Why capitalized?)

**Author Response:** Revised to break that long sentence down and de-capitalized the land use designations.

Section 5.2: First, the heading isn´t very informative. Second, you may wanna consider half a sentence on saying that you now calculate sediment loads (Mg/yr) (i.e. area-normalized).

**Author Response:** We believe the header sufficiently describes the content of this section. We did add an introductory sentence qualifying how we derive the mass loads from $\varepsilon$ and basin area.

Line 454: would suggest to replace "mass loss" with "mass produced".

**Author Response:** Done

463-465: You use the word "similar/similarity" three times in the same sentence. Please consider re-phrasing. Actually, the similarity with mainland Australia given their common geologic history not so surprising, but still a finding worth mentioning.

**Author Response:** Done

Fig. 9: I really like that Figure. A small issue for improvement, still: Could you indicate the trunk stream sample also in B/C, please?

**Author Response:** Done

483: This is in general a valid statement. But, it really depends where in the soil the clay rich horizons are, where 10Bem will be mostly located. So, I guess, this could be improved by relating to the local soil type and characteristics.

**Author Response:** Unfortunately, that specific of detail for soil depths is not available, which is why our comments here were more generic in nature.

511: It´s not clear to me where the bedload estimate is coming from. (Not from Table 6, it appears, where only suspended load is mentioned?)

**Author Response:** Bedload = $\varepsilon$ at TG-9 minus the measured dissolved load minus the measured suspended load. This is described in the previous sentence.

---

## Editor Decision (ED2)

Comments from Associate Editor Wittmann to Van Landingham et al., GChron

Dear Eric et al.,

Many thanks for the thorough revision on this manuscript. I really appreciate your efforts, which in my view have significantly improved the MS. I think you have in most cases corresponded adequately to my and the reviewers comments. There is one issue remaining, but I think that can be addressed with minor efforts. I would suggest, for time-efficient handling, that you address the (minor) comments I have below, after which the MS could be published.

With best wishes, Hella

**General comments:**
1) "Sensitivity analysis" of Dm under changing 9Beparent and F10Be_met. As mentioned in my and a reviewers first comments on V1, I am missing a short discussion in which directions Dm´s would go if, e.g. 9Beparent would be higher than 2.5 ppm ($\rightarrow$ Dm would decrease), or lower (basalt units), how changes in F10Be_met would affect Dm, and how retentivity would affect them (see the Q/E approach mentioned by reviewer #2). Of course, this assessment would be largely hypothetical (at least for 9Be because there simply isn´t any data on 9Be_parent for this catchment). However, previous studies have shown that a factor of 2 in 9Beparent even in granites, and something in the order of 30% for F10Bemet can be easily the case. Specifically for F10Bemet, wouldn´t the use of precipitation-derived [elevation-controlled…] meteoric flux counterbalance any elevation dependency? (Perhaps one way around this is using the basin-wide average F10Bemet value from the CGM. Having values for each catchment doesn´t benefit the discussion if their reliability is unclear. Also, this would normalize at least this effect, and the remaining effects could be better identified, perhaps…). I think looking at this carefully would put the Dm into the right frame, as a factor of 2 disagreement with $\varepsilon$ (or agreement, however you want to call it) is okay for a method that is still in its infancy. That´s why my main critique is that I would not generally call the 10Be/9Be-derived denudation rates as being sensitive to land use. We simply don´t know how the 9Be_reac plays into this, whereas 10Be_met-derived erosion rates can be clearly affected by e.g. soil loss. Also it seems like from Fig 7 that the way in which Dm and $\varepsilon$ scatter around the 1:1 line does not follow a trend with erosivity. When comparing to the landuse data, it looks like not all tributaries that have high landuse (erosivitiy) values also have Dm that are higher than $\varepsilon$ (i.e. for T6-8, Dm are actually lower than $\varepsilon$). This clearly points in my view at some random effects (for which in my view catchment-wide variability in 9Beparent would be a good candidate). For these reasons, I would ask you to tune down these statement about Dm and land-use affected areas.

2) Regarding the use of "Mg/km2*yr", I can follow your argument about metric vs. imperial tons and SI units. However, I would still suggest to use the more commonly used "t(metric)/km2*yr" (simply define "t" as being metric tons" somewhere). "Mg" is just very clumsy and not often seen in use.

3) The Methods part is very long- Would suggest that you use subheadings to give the section more structure.

**Line comments:**

100: "mass loss from" (typo)

Fig. 1: Please include a lat/long grid

118: 10Bem is desorbed under LOW pH conditions.

125: I know this sounds picky, but this should be "10Bem/9Bereac"-derived denudation rates (Just to avoid confusion with "10Bem"-based *erosion* rates). Here and elsewhere (e.g. heading section 4.2, line 278).

141…,located in northeastern Tasmania,

Fig. 3: Caption sais that "at least one full year of recorded data…". From Table 1, I get the impression that the data coverage is much better (at least 4 years)?

Table 4: Please change "Q" in D_m eq. to "10BeFmet".

172: over which time period has this mass been removed?

195ff: Note that grain size effects may impact the 9Be_reac/9Be_min ratio when measured in river sediments due to sorting (not for soils!), because 9Be_reac is potentially enriched over 9Be_min in finer particles. Hence, the D equation is not fully grain size independent (even though grain size variations in the 10Be_reac/9Be_reac are minor).

Again 195: "10Bem and the reactive as well as silicate-bound 9Be phases (9Bereac, 9Bemin, respectively)….

200: That equation/approach was originally suggested by Brown et al., 1988 (ESPL).

217: What does "to a single point" mean?

280: The depth of regolith per se does not tell anything about mobility of 10Be_m. You could use that depth to estimate the integration time scale of Dm (Willenbring & fvB 2010), but mobility itself can only be evaluated from pH, discharge, resulting Kd, dissolved v. reactive measurements, reactive 9Be vs. 10Be….So, I am not sure that the text for regolith depth (around lines 285) is needed.

Table5: If I type in all numbers that you give, I get slightly lower values for Dm- e.g. 54 instead of 60 t/km2*yr for TG5. I think that lower value was the value presented in the first version of the MS (Table 3). I note that some of the values (Version 1 vs this version) changed. Why ? Flux numbers are the same as before, even though you used now different precipitation rates? Please check all calculations again.

398: frost-cracking: this reads like there is basin-wide temperature data (and not only data from point stations). Would you like to put that into a map? Seems like frost cracking plays a dominate role on ε (conclusion), which is why I would present such data somewhere.

411: remove the second "of". What is "stymy" erosion?

414: Has % of bedrock outcrops been mapped in the George River basin, or is there data from landslide occurrence?

424: The part of the sentence on absent long-lasting dilution effects needs a justification. The whole section (420-430) would benefit in my view from a topic sentence around the question whether ε/ in situ cosmogenic nuclide concentrations are at steady state or not in the basin, and then present the arguments.

Actually, if you discuss whether ε are at steady-state, why not do the same for Dm? Clearly, 10Be_m concentrations are more sensitive to e.g. recent soil loss/general anthropogenic disturbances (see e.g. Belmont et al., 2014, ESPL, as an example for the available literature body on this topic). So, for 10Be_m concentrations this could be discussed. In this respect, the paper would benefit in my view from a sort of sensitivity analysis of Dm

for changes in F10Bemet and 9Beparent (see my initial comment from the first round, mentioned above, too).

429: I don´t understand the part about "normalized along with bedload characterisitcs". Are your referring to the decrease in mean particle size ?

443-446: That is a long sentence....almost "German".  Please split. Is "Conservation and Protective Native Land Cover" a government project? (Why capitalized?)

Section 5.2: First, the heading isn´t very informative. Second, you may wanna consider half a sentence on saying that you now calculate sediment loads (Mg/yr) (i.e. area-normalized). Line 454: would suggest to replace "mass loss" with "mass produced".

463-465: You use the word "similar/similarity" three times in the same sentence. Please consider re-phrasing. Actually, the similarity with mainland Australia given their common geologic history not so surprising, but still a finding worth mentioning.

Fig. 9: I really like that Figure. A small issue for improvement, still: Could you indicate the trunk stream sample also in B/C, please?

483: This is in general a valid statement. But, it really depends where in the soil the clay rich horizons are, where 10Bem will be mostly located. So, I guess, this could be improved by relating to the local soil type and characteristics.

511: It´s not clear to me where the bedload estimate is coming from. (Not from Table 6, it appears, where only suspended load is mentioned?)

---

## Author Response (AR3)

Dear Dr. Balco and Dr. Wittmann-Oelze

Thank you for your decision to accept our study for publication in *Geochronology*. I have copied your final comments below and responded to them.

Kindest regards,
Dr. Eric W. Portenga
Leah A. VanLandingham
Dr. Edward C. Lefroy
Dr. Amanda H. Schmidt
Dr. Paul R. Bierman
Dr. Alan J. Hidy

**Comments to the author**:
Thanks very much to all the authors of this paper for significantly revising the paper through multiple rounds of review; I think the paper is significantly improved over the initially submitted version. This paper is accepted for publication subject to several minor technical corrections as follows. Please make these corrections on the text and figures submitted for copy-editing.

1. As noted in the AE comments, I believe the land acknowledgement should be placed in a separate section headed 'Acknowledgements.'

**Author Response:** Changed as suggested.

2. "Stymie" is typically spelled with an "ie," at least in the US, but we can defer that to the copy-editing professionals if you want.

**Author Response:** Corrected spelling as suggested.

3. I have one minor technical concern about the maps in Figures 2 and 4. These maps are drawn in an odd projection. At first I thought they were drawn in a completely uncorrected geographic projection such that degrees of longitude (the x-coordinate) and degrees of latitude (the y-coordinate) have equal length. However, this is not exactly the case; I actually printed out Figure 2 to measure and found a degree of longitude is about 90% of a degree of latitude in the map coordinates. In reality, at 41° latitude, a degree of longitude is about 75% of the length of a degree of latitude, so this choice of projection distorts the map by stretching features in an east-west direction.

If basin areas or perimeters were calculated in this projection, they would be incorrect. However, basin areas are listed in Table 2 in km2, so one would tend to assume that they were calculated correctly using an equal-area projection and not the projection used in the figures. Thus, there is no evidence that there is anything quantitatively wrong with using this projection for the figures.

Although it makes the basins look a bit squashed, it helps graphically in fitting multiple views of the map into a compact figure.

What it does mean that a scale bar, as is given in all the maps, is correct in only one compass direction and wrong in others. I also measured out the scale bar and found that it was correct in relation to the latitude ticks, i.e., in the north-south direction. What this means, of course, is that it is not correct in the east-west direction. Thus, either multiple scale bars should be given, or the existing scale bars should be removed. As the lat/long ticks at the edge of the maps redundantly provide scale information, I suggest removing the scale bars entirely.

Thus, please remove the scale bars. It would also be helpful to readers if the projection was stated in the caption.

**Author Response:** It turns out the latitude tick marks were incorrectly placed, but the Editor is correct that these maps were initially produced in uncorrected geographic coordinates. The maps and figures were redrafted to exhibit the field area in UTM 55S, which is now clarified in figure captions for Figures 2 and 4. The scale bars have been removed to avoid redundancy. The basin areas were previously calculated using an equal-area projection; they have been verified and unchanged from previous versions of the manuscript.

In reproducing the maps in Figure 2 and Figure 4, we used a World Topographic Map basemap for which Esri owns the copyright. There seem to be many nuanced ways of attributing Esri's basemaps, and I believe I have followed their requirements (copyright on the maps themselves, a citation to the online dataset, and an acknowledgement in the Data Availability section). From a laypersons viewpoint, it seems a lot for a basemap, so please advise if there is an alternative option to cite this dataset.

Thanks for your attention to this, and thank you for submitting your work to Geochronology.

-- greg

**Comments to the author**:
Thank you for your prompt and complete revision. Please consider including a proper acknowledgement text. All the best, Hella

**Author Response:** We thank you for your constructive comments throughout this process.